# Efficient Graph Matching
# for Correlated Stochastic Block Models

**Shuwen Chai**
Northwestern University
Evanston, IL 60208
shuwenchai2027@u.northwestern.edu

**Miklós Z. Rácz**
Northwestern University
Evanston, IL 60208
miklos.racz@northwestern.edu

## Abstract

We study learning problems on correlated stochastic block models with two balanced communities. Our main result gives the first efficient algorithm for graph matching in this setting. In the most interesting regime where the average degree is logarithmic in the number of vertices, this algorithm correctly matches all but a vanishing fraction of vertices with high probability, whenever the edge correlation parameter $s$ satisfies $s^2 > \alpha \approx 0.338$, where $\alpha$ is Otter's tree-counting constant. Moreover, we extend this to an efficient algorithm for exact graph matching whenever this is information-theoretically possible, positively resolving an open problem of Rácz and Sridhar (NeurIPS 2021). Our algorithm generalizes the recent breakthrough work of Mao, Wu, Xu, and Yu (STOC 2023), which is based on centered subgraph counts of a large family of trees termed chandeliers. A major technical challenge that we overcome is dealing with the additional estimation errors that are necessarily present due to the fact that, in relevant parameter regimes, the latent community partition cannot be exactly recovered from a single graph. As an application of our results, we give an efficient algorithm for exact community recovery using multiple correlated graphs in parameter regimes where it is information-theoretically impossible to do so using just a single graph.

## 1 Introduction

The proliferation of network data has highlighted the ubiquity and importance of *graph matching* in machine learning, with applications in a variety of domains, including social networks [51, 56], computational biology [61], and computer vision [13, 36]. While the graph matching task—recovering the latent node alignment between two networks—is known to be NP-hard to solve or even approximate in general [38, 52], in practice it is often possible to solve it well, such as in the works cited above. This has motivated an exciting recent line of work studying average-case graph matching [14, 15, 65, 16, 23, 27, 24, 49, 5, 21, 22, 39, 40, 41, 17, 20, 18], focusing on correlated Erdős–Rényi random graphs [56]. These papers culminated in recent breakthrough works which developed efficient graph matching algorithms in the constant noise regime [40, 41].

However, real-world networks are not modeled well by Erdős–Rényi random graphs, which in turn has motivated a growing line of recent work studying graph matching beyond Erdős–Rényi [8, 32, 11, 54, 58, 70, 57, 25, 63, 59, 19, 67, 66, 60, 10]. In particular, an important problem in this vein is to study graph matching in *correlated stochastic block models (correlated SBMs)* [35, 54, 34] (see Section 2 for definitions), since community structure is prevalent in many networks and the community recovery problem is a fundamental inference task that is often a starting point for deeper analyses. Recent work of Rácz and Sridhar [57] determined the fundamental information-theoretic limits for exact graph matching in correlated SBMs; however, the underlying algorithm used to achieve this limit is inefficient (that is, not polynomial time). Rácz and Sridhar [57] posed the open problem of finding an *efficient* algorithm for (exact) graph matching whenever this is information-theoretically feasible.

38th Conference on Neural Information Processing Systems (NeurIPS 2024).

Our main contribution positively resolves this open problem of Rácz and Sridhar [57], giving the first efficient algorithm for graph matching for correlated SBMs with two balanced communities, under a condition on the correlation strength that is conjectured to be necessary. Specifically, we give an efficient algorithm that, in the most interesting regime where the average degree is logarithmic in the number of vertices, achieves almost exact recovery of the latent matching, whenever the edge correlation parameter $s$ satisfies $s^2 > \alpha \approx 0.338$, where $\alpha$ is Otter's tree-counting constant. Moreover, we extend this to an efficient algorithm for exact graph matching whenever this is information-theoretically possible. See Section 3 and Theorem 1 for details.

In addition, our results on graph matching directly imply novel efficient algorithms and results for community recovery. Specifically, combining—in a black-box fashion—our (exact) graph matching algorithm with existing community recovery algorithms, we give an efficient algorithm for exact community recovery using multiple correlated graphs in parameter regimes where it is information-theoretically impossible to do so using just a single graph. See Section 3 and Theorem 2 for details.

Our algorithm generalizes the recent breakthrough work of Mao, Wu, Xu, and Yu [41], which is based on centered subgraph counts of a large family of trees termed chandeliers, to the setting of correlated SBMs. A major technical challenge that we overcome is dealing with the additional estimation errors that are necessarily present due to the fact that, in relevant parameter regimes, the latent community partition cannot be exactly recovered from a single graph, and thus the edge-indicator variables in the centered subgraph counts cannot be precisely centered. Our technical contributions highlight the interplay between graph matching and community recovery in ways that are complementary to the recent work of Gaudio, Rácz, and Sridhar [25].

## 2 Models and problems

In this section we describe the setting of the paper by introducing the stochastic block model (SBM), correlated SBMs, and the community recovery and graph matching tasks.

The **stochastic block model (SBM)** is the canonical probabilistic generative model for a network with latent community structure. The SBM was first introduced by Holland, Laskey, and Leinhardt [29] and has been widely studied over the past decades [1]. In general, an SBM may consist of a number of communities, with distinct vertices connected randomly with a probability that depends on their community memberships.

In this work, we focus on the simplest setting of the balanced two-community SBM. Given $n \in \mathbb{Z}_+$ and $p, q \in [0, 1]$, we construct $G \sim \mathrm{SBM}(n, p, q)$ as follows. The graph $G$ has $n$ vertices, with vertex labels given by $V = [n] := \{1, 2, \ldots, n\}$. Let $\boldsymbol{\sigma}_* = \{\boldsymbol{\sigma}_*(i)\}_{i=1}^n$ be the vector of community labels, where each entry $\boldsymbol{\sigma}_*(i) \in \{-1, +1\}$ is drawn independently and uniformly at random. Then, given the community labels $\boldsymbol{\sigma}_*$, for any pair of vertices $i \neq j \in [n]$, edge $(i, j)$ is in $G$ with probability $p\mathbf{1}_{\{\boldsymbol{\sigma}_*(i)=\boldsymbol{\sigma}_*(j)\}} + q\mathbf{1}_{\{\boldsymbol{\sigma}_*(i)\neq\boldsymbol{\sigma}_*(j)\}}$. That is, two different vertices are connected with probability $p$ if they are from the same community and connected with probability $q$ otherwise.

**Correlated SBMs** are multiple SBMs where the corresponding edge variables are correlated [35, 54, 34]. Specifically, we construct two correlated SBMs $(G_1, G_2) \sim \mathrm{CSBM}(n, p, q, s)$ using a natural subsampling procedure as follows. Let $G \sim \mathrm{SBM}(n, p, q)$ be a parent graph with community labels $\boldsymbol{\sigma}_*$. Next, given $G$, we construct $G_1$ by random sampling of the edges: each edge of $G$ is included in $G_1$ with probability $s$, independently of everything else, and non-edges of $G$ remain non-edges of $G_1$. We then do the edge sampling independently again to obtain $G_2'$ in the same way. The child graphs $G_1$ and $G_2'$ inherit both the vertex labels (given by $[n]$) and the community labels $\boldsymbol{\sigma}_*$ from the parent graph $G$. Finally, let $\pi_*$ be a uniformly random permutation of $[n] := \{1, 2, \ldots, n\}$ and generate $G_2$ by relabeling the vertices of $G_2'$ according to $\pi_*$ (e.g., vertex $i$ in $G_2'$ is relabeled as $\pi_*(i)$ in $G_2$). This last step reflects the fact that in practice often the correspondence between the two vertex sets is unknown. We denote the adjacency matrices of $G_1$ and $G_2$ as $A$ and $B$, respectively, and note that the community labels of the two graphs are $\boldsymbol{\sigma}_*^A := \boldsymbol{\sigma}_*$ and $\boldsymbol{\sigma}_*^B := \boldsymbol{\sigma}_* \circ \pi_*^{-1}$, respectively. See Figure 1 for an illustration.

Marginally, $G_1$ and $G_2$ are identically distributed SBMs: we have $G_1, G_2 \sim \mathrm{SBM}(n, ps, qs)$. Moreover, $G_1$ and $G_2$ are *correlated*. Specifically, for every pair of distinct vertices $\{i, j\}$, the edge-indicator random variables $A_{i,j}$ and $B_{\pi_*(i),\pi_*(j)}$ are correlated Bernoulli random variables. A simple calculation shows that if $\boldsymbol{\sigma}_*(i) = \boldsymbol{\sigma}_*(j)$, then the correlation coefficient of $A_{i,j}$ and $B_{\pi_*(i),\pi_*(j)}$ is

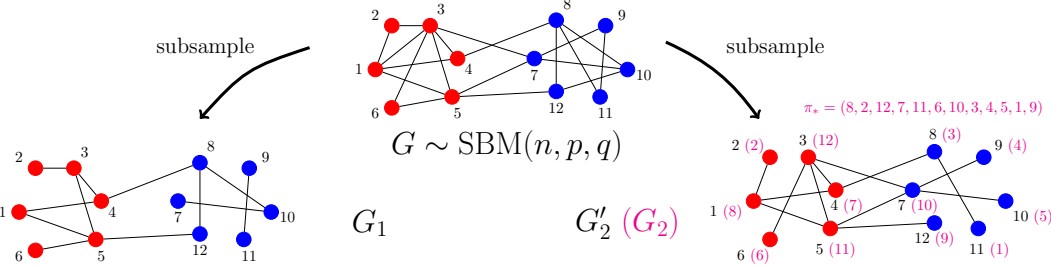

Figure 1: Schematic illustrating two-community correlated SBMs; see the text for details. (Figure reproduced from [57] with permission.)

equal to $\rho_+ := s\frac{1-p}{1-ps}$, whereas if $\boldsymbol{\sigma}_*(i) \neq \boldsymbol{\sigma}_*(j)$, then this correlation coefficient is $\rho_- := s\frac{1-q}{1-qs}$. Our focus will be on the sparse setting where $p, q = o(1)$ (as $n \to \infty$), in which case both $\rho_+$ and $\rho_-$ are asymptotically $(1 - o(1))s$, and hence we can regard $s$ as the *edge correlation parameter*.

**Community recovery.** The goal of community recovery is to recover the latent community labels $\boldsymbol{\sigma}_*$ given some (graph) data, such as an SBM $G$ or correlated SBMs $(G_1, G_2)$. There are various notions of community recovery, depending on how close an estimate is to the ground truth $\boldsymbol{\sigma}_*$. In this work, we focus on *exact community recovery*, defined as follows: an estimator $\widehat{\boldsymbol{\sigma}}$ achieves exact community recovery if $\lim_{n\to\infty} \mathbb{P}(|\frac{1}{n}\sum_{i=1}^n \widehat{\boldsymbol{\sigma}}(i)\boldsymbol{\sigma}_*(i)| = 1) = 1$. The absolute value is present in the previous expression since we can only hope to recover the community labels up to a global sign flip; in other words, our goal is to recover the partition of the graph into two communities. A slightly weaker notion, which also appears throughout our work, is *almost exact community recovery*, which holds if $\lim_{n\to\infty} \mathbb{P}(|\frac{1}{n}\sum_{i=1}^n \widehat{\boldsymbol{\sigma}}(i)\boldsymbol{\sigma}_*(i)| = 1 - o(1)) = 1$; in other words, this notion tolerates a vanishing fraction of errors. Further weaker notions include partial recovery and weak recovery; since these are not the focus here, we refer to [1] for details.

Different parameter regimes give rise to different challenges and different notions of recovery become most relevant. In the constant average degree regime, that is, when $p = \frac{a}{n}$ and $q = \frac{b}{n}$ for some constants $a$ and $b$, it is impossible to recovery the communities exactly. Prior works [46, 48, 43] have characterized the information-theoretic threshold and developed efficient algorithms for partial recovery in this regime. On the other hand, if the vertices have polynomially growing degrees, that is, when $p = n^{-a+o(1)}$ and $q = n^{-b+o(1)}$ for some constants $a, b \in [0, 1)$, then community recovery is easy as long as $\liminf_{n\to\infty} |p_n/q_n - 1| > 0$ (see [47]).

In this work, we focus on the "bottleneck regime" of logarithmic average degree, which is the bare minimum for the graph to be connected, and which is when exact community recovery is most interesting. In most of the paper we assume that $p = a\frac{\log n}{n}$ and $q = b\frac{\log n}{n}$ for some positive constants $a, b$. For an SBM with two balanced communities and these parameters, there is a sharp information-theoretic threshold for exact community recovery, which is given by $D_+(a, b) = 1$, where $D_+(a, b) := (\sqrt{a} - \sqrt{b})^2/2$ (see [47, 2, 4, 1]). This quantity is known as the *Chernoff–Hellinger divergence* in the general $k$-community SBM setting with linear size communities [4] and simplifies to the above form in our setting. In other words, when $D_+(a, b) > 1$, there exists an estimator $\widehat{\boldsymbol{\sigma}}$ that is computable in polynomial-time and which achieves exact community recovery with high probability. On the other hand, when $D_+(a, b) < 1$, exact community recovery is impossible, in the sense that for all estimators $\widehat{\boldsymbol{\sigma}}$ we have that $\lim_{n\to\infty} \mathbb{P}(|\frac{1}{n}\sum_{i=1}^n \widehat{\boldsymbol{\sigma}}(i)\boldsymbol{\sigma}_*(i)| = 1) = 0$. Moreover, when it is possible to achieve exact community recovery on a single graph, several polynomial-time algorithms have been studied by previous works (e.g., [47, 2]).

**Community recovery and graph matching.** What are the information-theoretic limits for exact community recovery given two correlated SBMs $(G_1, G_2) \sim \text{CSBM}(n, a\frac{\log n}{n}, b\frac{\log n}{n}, s)$? This question was initiated and partially solved by Rácz and Sridhar [57], and subsequently fully solved by Gaudio, Rácz, and Sridhar [25]. Without yet going into the details, these works highlight the importance of *graph matching*, that is, the task of recovering the latent matching $\pi_*$ given the two correlated graphs $(G_1, G_2)$. In brief, when $\pi_*$ can be perfectly recovered from $(G_1, G_2)$, then one can take the union graph $G_1 \vee_{\pi_*} G_2$ of $G_1$ and $G_2$, which is also an SBM, but with a larger edge density,

which makes community recovery easier. This shows that exact community recovery is possible from $(G_1, G_2)$ even in parameter regimes where this is impossible from just a single graph $G_1$ (see [57]).

**Graph matching.** Motivated by the above discussion, we now discuss average-case graph matching and notions of recovery. In general, suppose that $(G_1, G_2)$ are two correlated random graphs with $n$ vertices each and that $\pi_*$ is the underlying latent vertex matching. The goal of graph matching is to output an estimator $\widehat{\pi} = \widehat{\pi}(G_1, G_2)$ that is close to $\pi_*$. There are various notions of recovery depending on how close $\widehat{\pi}$ is to $\pi_*$; the two most relevant notions are the following. We say that an estimator $\widehat{\pi}$ achieves *exact graph matching* if $\lim_{n \to \infty} \mathbb{P}(\widehat{\pi} = \pi_*) = 1$. We say that an estimator achieves *almost exact graph matching* if with high probability there exists a subset $I \subseteq [n]$ with $|I| = (1 - o(1))n$ such that $\widehat{\pi}|_I = \pi_*|_I$, where $\pi|_I$ denotes the restriction of $\pi$ to $I$. In words, almost exact graph matching allows the estimator to make a vanishing fraction of errors.

The graph matching problem has been widely studied, with applications to computer vision [13, 36], computational biology [61], and social networks [56]. In particular, de-anonymizing social networks is possible with graph matching algorithms, which implies that anonymity is not equivalent to privacy [51]. That said, studying the limits of graph matching algorithms—including potential information-computation gaps, as we shall discuss—can help guide data regulators on when to take more actions with regards to data protection, in addition to anonymity.

As discussed in the introductory paragraphs, there has been a large body of recent work on average-case graph matching, both studying correlated Erdős–Rényi random graphs [56, 14, 15, 65, 16, 23, 27, 24, 49, 5, 21, 22, 39, 40, 41, 17, 20, 18] and more general models of correlated random graphs [8, 32, 11, 54, 58, 70, 57, 25, 63, 59, 19, 67, 66, 60, 10]. In particular, Rácz and Sridhar [57] determined the fundamental information-theoretic limits for exact graph matching in correlated SBMs: in the logarithmic average degree regime discussed above, this threshold is $s^2 \frac{a+b}{2} = 1$. However, this result is information-theoretic, and the authors posed the open problem of finding an *efficient* algorithm for exact graph matching, whenever this is information-theoretically possible.

Our main contribution positively resolves this open problem, giving the first efficient algorithm for graph matching for correlated SBMs with two balanced communities. Our algorithm generalizes the recent breakthrough work of Mao, Wu, Xu, and Yu [41] that developed an efficient graph matching algorithm for correlated Erdős–Rényi graphs. As an application, our results imply novel efficient algorithms and results for community recovery. We now turn to describing our results.

## 3  Main results: Graph matching

Our main theorem for graph matching on correlated SBMs is that there exists a polynomial-time algorithm that can achieve exact matching if $s^2 > \alpha$, where $\alpha$ is Otter's tree counting constant[1] [55].

**Theorem 1.** *Fix constants $a \neq b > 0$ and $s \in [0, 1]$. Let $(G_1, G_2) \sim \mathrm{CSBM}(n, a\frac{\log n}{n}, b\frac{\log n}{n}, s)$. For any $\varepsilon > 0$, if $s^2 \geq \alpha + \varepsilon$, then the following holds.*

(a) *(**Almost exact matching**) There exists a polynomial-time algorithm that outputs a subset $I \in [n]$ and a mapping $\widehat{\pi} : I \to [n]$ such that $\widehat{\pi} = \pi_*|_I$ and $|I| = (1 - o(1))n$ with high probability.*

(b) *(**Exact matching**) If, in addition, $s^2(a + b)/2 > 1$, then there exists a polynomial-time algorithm that ouputs a mapping $\widehat{\pi}$ such that $\lim_{n \to \infty} \mathbb{P}(\widehat{\pi} = \pi_*) = 1$.*

Several remarks are now in order about the tightness of the main result, an overview of the chandelier counting algorithm when $a = b$, and the main challenge of our analysis.

**Tightness.** This result is tight whenever $s^2 > \alpha$, because $s^2(a+b)/2 = 1$ is the information-theoretic threshold of exact graph matching given two correlated SBMs $(G_1, G_2)$ [57]. When $s^2 < \alpha$, it is conjectured that an information-computation gap exists for the correlated Erdős–Rényi graphs [41]. Specifically, by assuming that $a = b$, it is information-theoretically possible to match correlated Erdős–Rényi graphs exactly if $s^2 a > 1$ [14, 15, 65]. However, it is believed hard to find a polynomial-time algorithm to do this. In our model with SBMs, which is an extension from Erdős–Rényi graphs, it is also likely hard to find a polynomial-time algorithm when $s^2 < \alpha$.

---

[1]This constant captures the base of the exponential growth of unlabeled rooted trees: the total number of unlabeled rooted trees with $N$ vertices is $(\alpha + o(1))^{-N}$.

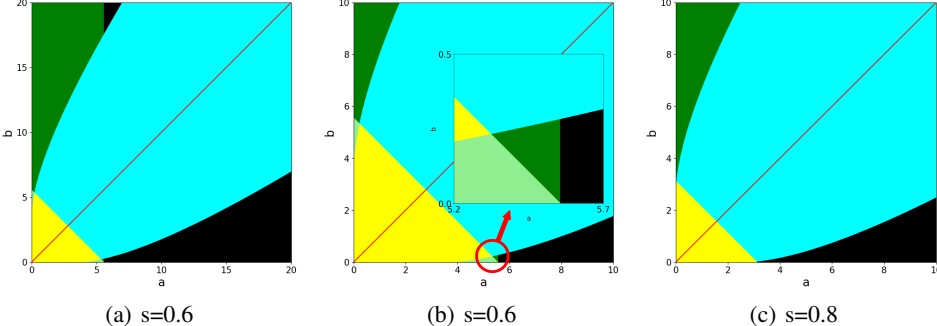

(a) s=0.6        (b) s=0.6        (c) s=0.8

Figure 2: Phase diagram for graph matching on $(G_1, G_2) \sim \text{CSBM}(n, \frac{a \log n}{n}, \frac{b \log n}{n}, s)$. The red diagonal line depicts $a = b$, which is an Erdős–Rényi graph. *Black regions*: exact graph matching is possible and can be done efficiently for each community separately by applying the graph matching algorithm for correlated Erdős–Rényi graphs; *Green regions*: exact graph matching is possible and can be done efficiently; *Light green regions*: exact graph matching is impossible, but almost exact graph matching is possible and can be done efficiently; *Cyan regions*: exact graph matching is possible and can be done efficiently by first recovering the community labels almost exactly; *Yellow regions*: exact graph matching is impossible but almost exact graph matching can be done efficiently by first recovering the community labels almost exactly.

**Signed chandelier counts.** Our theorem extends from the main theorem in Mao et al. [41], which proposed a polynomial-time algorithm that matches the correlated Erdős–Rényi graphs exactly. The algorithm has two main steps: First, construct signature vectors $\boldsymbol{s}_i$ and $\boldsymbol{t}_j$ for vertices $i \in [n]$ in $G_1$ and $j \in [n]$ in $G_2$ by the *signed subgraph counts* of a specially designed graph class—termed *Chandeliers*—and calculate the weighted inner product of pairs of signature vectors $\langle \boldsymbol{s}_i, \boldsymbol{t}_j \rangle$ and match vertices if the inner product value is large enough; Second, use a seeded graph matching algorithm to boost the almost exact graph matching algorithm to exact graph matching. It is natural to adapt this algorithm from correlated Erdős–Rényi graphs to correlated SBMs but the details present non-trivial challenges, as we explain below.

**Main challenge.** The main challenge on correlated SBMs is that signed subgraph counts is no longer a free lunch. Signed subgraph counts is counting the subgraphs on a centralized adjacency matrix, which is first proposed by Bubeck et al. [9] and later commonly used to control the variance of counting statistics. The success of the chandelier counting method relies on the sufficient separation of the two inner product distributions of true and false vertex correspondence. We want to find a way that keeps doing the adjacency matrices centralization possible. Recall that we explore the graph matching motivated by community recovery. Interestingly, the solution to this centralization problem is now the other way around—using a rough community label estimate to help the graph matching. Our main technical contributions are first showing that when there are no error occurs in the community label estimate, the signed chandelier counting can be generalized to correlated SBMs and then show that when the exact community recovery is not possible, the errors introduced in the community label estimation, which is polynomial in $n$, are actually tolerable for the whole algorithm.

The analysis falls in two cases. If $sD_+(a, b) > 1$, then we can achieve exact community recovery on each of the graphs by applying the community recovery algorithm from [47]. In addition, if $s^2 \frac{a}{2} > 1$,[2] then it suffices to look at each community individually. This is easy and follows in a black-box fashion from [41] (See Section 7). However, on the other side, $s^2 \frac{a}{2} < 1$, one still needs to use information the community information. Therefore, we need to go through the whole algorithm analysis again in this case. Note that the analysis would work for both regimes with no constraint on $s^2 \frac{a}{2} < 1$. We plot the black-box regime in *black* and the non-black-box regime in *green* in Figure 2.

The second case is even trickier. If $sD_+(a, b) < 1$, by the same algorithm, we can only obtain almost exact correct community labels $\widehat{\boldsymbol{\sigma}}_A$ and $\widehat{\boldsymbol{\sigma}}_B$ on graph $G_1$ and $G_2$, respectively. We perform adjacency matrix centralization based on $\widehat{\boldsymbol{\sigma}}_A$ and $\widehat{\boldsymbol{\sigma}}_B$ and show that the error introduced in this step is negligible

---

[2]For a SBM$(n, \frac{a \log n}{n}, \frac{b \log n}{n})$, each community, conditioned on its size $N \approx n/2$, is an Erdős–Rényi graph $\mathcal{G}(N, \frac{a \log n}{n}) \approx \mathcal{G}(N, \frac{a \log N}{2N})$.

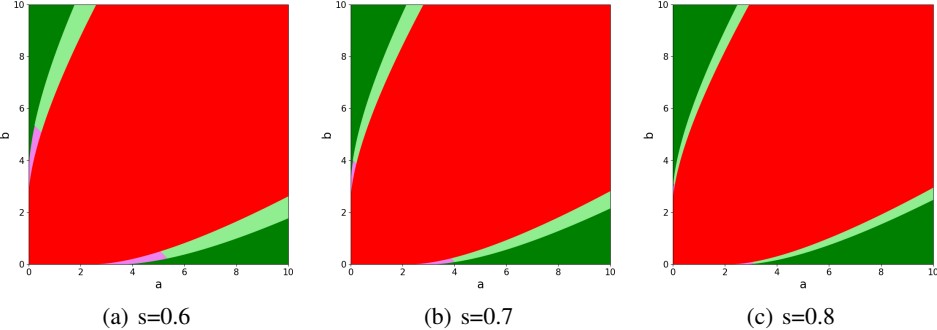

(a) s=0.6   (b) s=0.7   (c) s=0.8

Figure 3: Phase diagram for exact community recovery with fixed $s$ on correlated SBMs. *Green regions*: exact community recovery is possible from $G_1$ alone and can be done efficiently; *Lightgreen regions*: exact community recovery is possible from $(G_1, G_2)$ but impossible from $G_1$ alone, exact graph matching can be done efficiently and therefore exact community recovery can be done efficiently; *Violet regions*: exact community recovery is impossible from $G_1$ alone, impossible from $(G_1, G_2)$ if $s^2(\frac{a+b}{2}) + s(1-s)D_+(a,b) < 1$ and possible if $s^2(\frac{a+b}{2}) + s(1-s)D_+(a,b) > 1$ [25]. It is unknown whether there exists an efficient algorithm for exact community recovery in this regime.

in the sense that the inner-product scores remains sufficiently distinguishable between true pairs $(j = \pi_*(i))$ and fake pairs $(j \neq \pi_*(i))$.

# 4   Application: Community recovery

Once matching up the vertices on two correlated graphs, we can combine the information of them onto a union graph and then immediately have an application on community recovery. Our result for community detection is that there exists a polynomial-time algorithm for exact community recovery on correlated SBMs when the squared edge correlation parameter satisfies $s^2 > \alpha$.

**Theorem 2.** *Fix constants $a \neq b > 0$ and $s \in [0, 1]$. Let $(G_1, G_2) \sim \mathrm{CSBM}(n, a\frac{\log n}{n}, b\frac{\log n}{n}, s)$. For any $\varepsilon > 0$, if*

$$s^2 \geq \alpha + \varepsilon, \quad s^2(\frac{a+b}{2}) > 1, \quad \text{and} \quad (1 - (1-s)^2)D_+(a,b) > 1,$$

*then, there exists an estimator $\widehat{\boldsymbol{\sigma}} = \widehat{\boldsymbol{\sigma}}(G_1, G_2)$ that can be computed in polynomial-time such that $\lim_{n \to \infty} \mathbb{P}(|\frac{1}{n}\sum_{i=1}^n \widehat{\boldsymbol{\sigma}}(i)\boldsymbol{\sigma}_*(i)| = 1) = 1$.*

Theorem 2 is a direct application of our Theorem 1. The proof mainly follows the Theorem 3.3 in [57], which gives exact community recovery on the union graph of $G_1$ and $G_2$ regarding to the permutation $\widehat{\pi}$, $G_1 \vee_{\widehat{\pi}} G_2$. The key difference is that we substitute the maximum a posterior estimator used in the first step with the $\widehat{\pi}(G_1, G_2)$ output by the algorithm used to prove Theorem 1. Figure 3 is a summary of the phase diagram for community recovery determined by this work along with previous works [57, 25], focusing on the exact community recovery and efficiency.

**Remark 1.** *Consider a more general correlated SBMs with $K$ correlated graphs $(G_1, G_2, \ldots, G_K) \sim \mathrm{CSBM}(n, a\frac{\log n}{n}, b\frac{\log n}{n}, s, K)$. Theorem 1 also implies an efficient algorithm for exact community recovery above the exact graph matching threshold for $K$ correlated SBMs when $D_+(a,b) > \frac{1}{1-(1-s)^K}$ (Theorem 3.6 in [57]).*

# 5   Algorithm and the overview of the proofs

In this section, we define a few key concepts, give a brief overview of the algorithm, and briefly discuss the proof of Theorem 1 and challenges. See Appendix C for a full version.

**Chandelier.** An $(L, M, K, R, D)$-chandelier is a rooted tree with $L$ branches, each of which consists of a path with $M$ edges ($M$-wire), followed by a rooted tree with $K$ edges ($K$-bulb); the $K$-bulbs are

non-isomorphic to each other, each of them has at most $R$ automorphisms, and the maximum degree is at most $D$. For each chandelier $H$, let $\mathcal{K}(H)$ denote the set of bulbs of $H$.

For a rooted tree $T$, let $\mathrm{aut}(H)$ denote the number of rooted automorphisms of $T$ throughout this paper. We abbreviate rooted automorphism as automorphism when it is clear that we are applying it to a chandelier. The number of automorphisms of $H$ is determined by the automorphisms of its bulbs.

Let $\mathcal{T}$ denote the family of non-isomorphic $(L, M, K, R, D)$–chandelier. The family size of chandelier is $|\mathcal{T}| = \binom{|\mathcal{J}|}{L}$, where $\mathcal{J} \equiv \mathcal{J}(K, R, D)$ denotes the collection of unlabeled rooted trees with $K$ edges, at most $R$ automorphisms, and maximum degree $D$. Otter [55] showed that the number of unlabeled rooted trees with $K$ edges (and no constraint on the automorphisms and vertex degrees) is $|\mathcal{J}(K, \infty, \infty)| = (\alpha + o(1))^{-K}$, where $\alpha \approx 0.338$. We show that under proper choices of $R$ and $D$, we have $|\mathcal{J}(K, R, D)| = (\alpha + o(1))^{-K}$ in Section C.

**Algorithm overview.** Given $(G_1, G_2) \sim \mathrm{CSBM}(n, p, q, s)$. Our algorithm contains mainly three steps. Firstly, we apply the Algorithm 3 (discussed in Section D.3) by Mossel, Neeman, and Sly [47] to obtain almost exact community label estimates for each single graph. Secondly, we calculate the signed chandelier counting [41] based similarity score to give an almost exact graph matching (Algorithm 1 and Algorithm 4). Lastly, we boost the almost exact matching to exact matching by extending the seeded graph matching algorithm [41] on Erdős–Rényi graphs to SBMs (Algorithm 2).

**Subgraph counts.** For an arbitrary weighted adjacency matrix $M$ of some adjacency matrix $A$, vertex $i \in [n]$, and a rooted graph $H$, we define the *weighted subgraph counts* on $M$ as

$$W_{i,H}(M) := \sum_{S(i) \cong H} M_S, \text{ where } M_S := \prod_{e \in E(S)} M_e,$$

and $S(i)$ enumerates subgraphs of the complete graph $K_n$, rooted at $i$, that are isomorphic to $H$.

When $M$ is the adjacency matrix itself, $W_{i,H}(M)$ is the usual subgraph count, representing the number of subgraphs rooted at $i$ in $M$ that are isomorphic to $H$. When $M$ is the centralized adjacency matrix $\overline{A} := A - \mathbb{E}[A]$, we call $W_{i,H}(M)$ a signed subgraph count following [9]. However, we do not have access to $\mathbb{E}[A]$ in many cases. Specifically for SBM, we can estimate $\mathbb{E}[A]$ through estimating the community labels. We define the *approximately centralized adjacency matrix* regarding to community label estimate $\widehat{\boldsymbol{\sigma}}$, denoted as $\overline{A}^{\widehat{\boldsymbol{\sigma}}}$, entry-wise as $\overline{A}^{\widehat{\boldsymbol{\sigma}}}_{i,j} = A_{i,j} - p\mathbf{1}_{\widehat{\boldsymbol{\sigma}}[i]=\widehat{\boldsymbol{\sigma}}[j]} - q\mathbf{1}_{\widehat{\boldsymbol{\sigma}}[i]\neq\widehat{\boldsymbol{\sigma}}[j]}$.

Given a family $\mathcal{H}$ of non-isomorphic rooted graphs, we define the *subgraph count signature* of vertex $i$ as $W_i^{\mathcal{H}}(M) := (W_{i,H}(M))_{H \in \mathcal{H}}$.

**Similarity score.** Given a pair of correlated SBMs $(G_1, G_2)$, we define the similarity score between vertex $i$ on graph $G_1$ and vertex $j$ on graph $G_2$ as a weighted inner product between two signatures:

$$\Phi_{ij} := \langle W_i^{\mathcal{T}}(\overline{A}), W_j^{\mathcal{T}}(\overline{B}) \rangle := \sum_{H \in \mathcal{T}} \mathrm{aut}(H) W_{i,H}(\overline{A}) W_{j,H}(\overline{B}).$$

When we do not have access to $\overline{A}$ and $\overline{B}$, we use the approximately centralized adjacency matrices $\overline{A}^{\widehat{\boldsymbol{\sigma}}}$ and $\overline{B}^{\widehat{\boldsymbol{\sigma}}}$. We define the similarity score in a slightly different notation:

$$\Phi_{ij}^{\widehat{\boldsymbol{\sigma}}} := \langle W_i^{\mathcal{T}}(\overline{A}^{\widehat{\boldsymbol{\sigma}}_A}), W_j^{\mathcal{T}}(\overline{B}^{\widehat{\boldsymbol{\sigma}}_B}) \rangle = \sum_{H \in \mathcal{T}} \mathrm{aut}(H) W_{i,H}(\overline{A}^{\widehat{\boldsymbol{\sigma}}_A}) W_{j,H}(\overline{B}^{\widehat{\boldsymbol{\sigma}}_B}).$$

**Almost exact graph matching.** The first part in the analysis is to show that by calculating this similarity score, with an appropriate thresholding strategy, we can match up $(1 - o(1))n$ vertices correctly (Theorem 3). The high-level idea is to show that the similarity score distributions are well-separated between true pairs and fake pairs. We expect the similarity score having the following properties, under event $\mathcal{H} := \{\frac{n}{2} - n^{\frac{3}{4}} \le |V^+|, |V^-| \le \frac{n}{2} + n^{\frac{3}{4}}\}$ (to be discussed in Section D.2):

- For true pairs $j = \pi(i)$ :

$$\mathbb{E}[\Phi_{i\pi_*(i)}^{\widehat{\boldsymbol{\sigma}}}\mathbf{1}_{\mathcal{H}}] > 0, \quad \mathrm{Var}[\Phi_{ij}^{\widehat{\boldsymbol{\sigma}}}\mathbf{1}_{\mathcal{H}}] = o\left(\mathbb{E}[\Phi_{i\pi_*(i)}^{\widehat{\boldsymbol{\sigma}}}\mathbf{1}_{\mathcal{H}}]^2\right), \qquad (1)$$

- For fake pairs $j \neq \pi(i)$ :

$$\mathbb{E}[\Phi_{ij}^{\widehat{\boldsymbol{\sigma}}}\mathbf{1}_{\mathcal{H}}] = o\left(\mathbb{E}[\Phi_{i\pi_*(i)}^{\widehat{\boldsymbol{\sigma}}}\mathbf{1}_{\mathcal{H}}]\right), \quad \mathrm{Var}[\Phi_{ij}^{\widehat{\boldsymbol{\sigma}}}\mathbf{1}_{\mathcal{H}}] = o\left(\frac{\mathbb{E}[\Phi_{i\pi_*(i)}^{\widehat{\boldsymbol{\sigma}}}\mathbf{1}_{\mathcal{H}}]^2}{n^2}\right). \qquad (2)$$

Precisely forming bounds for these moments constitutes the main bulk of the paper. We provide results from Proposition 1 to Proposition 6. Followed by these moment bounds, we have Theorem 3.

**Theorem 3.** *Fix $a \neq b > 0$ and $s \in [0,1]$. Let $p = a\frac{\log n}{n}, q = b\frac{\log n}{n}$ and $(G_1, G_2) \sim$ CSBM$(n, p, q, s)$. For any $\varepsilon > 0$, suppose $s^2 \geq \alpha + \varepsilon$. There exists positive constants $C_1, C_2, C_3, C_4, C_5 > 0$ such that the following holds. Pick $K, M, L, N, D$ as*

$$L = \frac{C_1}{\varepsilon}, \quad K = C_2 \log n, \quad M = \frac{C_3 K}{\log(ns(p \wedge q))}, \quad R = \exp(C_4 K), \quad D = C_5 \frac{\log n}{(\log \log n)^2}. \tag{3}$$

*Pick an arbitrary $c \in (0,1)$ and set $\mu = |\mathcal{T}| n^N \rho^N \sigma_{\text{eff}}^{2N}$, where $\sigma_{\text{eff}}^2 := (\frac{\sigma_+^2 + \sigma_-^2}{2})$. Then, Algorithm 1 outputs a set $I$ with size $(1 - o(1))n$ and a mapping $\widehat{\pi}$ such that $\widehat{\pi}|_I = \widehat{\pi}_*|_I$ with high probability.*

**Proof challenge.** In regime $sD_+(a,b) > 1$, the probability of existing one vertex being classified incorrectly is vanishing for Algorithm 3. Therefore, with high probability, $\overline{A}^{\widehat{\sigma}_A} = \overline{A}$. If $sD_+(a,b) < 1$, then the recovered $\widehat{\sigma}$ contains errors (polynomial in $n$), which will cause some edges being centralized incorrectly and thereby affect the moments calculation.

This phenomenon poses a challenge to the algorithm analysis. We highlight some key points in the context of second moment calculation. For simplicity, we ignore event $\mathcal{H}$ here. From definition,

$$\text{Var}[\Phi_{ij}^{\widehat{\sigma}}] = \sum_{H, I \in \mathcal{T}} \text{aut}(H)\text{aut}(I) \sum_{S_1(i), S_2(j) \cong H} \sum_{T_1(i), T_2(j) \cong I} \left( \mathbb{E}_{\sigma_*} \left[ \mathbb{E}[\overline{A}_{S_1}^{\widehat{\sigma}_A} \overline{B}_{S_2}^{\widehat{\sigma}_B} \overline{A}_{T_1}^{\widehat{\sigma}_A} \overline{B}_{T_2}^{\widehat{\sigma}_B} \mid \sigma_*] \right] \right.$$

$$\left. - \mathbb{E}_{\sigma_*} \left[ \mathbb{E}[\overline{A}_{S_1}^{\widehat{\sigma}_A} \overline{B}_{S_2}^{\widehat{\sigma}_B} \mid \sigma_*] \right] \mathbb{E}_{\sigma_*} \left[ \mathbb{E}[\overline{A}_{T_1}^{\widehat{\sigma}_A} \overline{B}_{T_2}^{\widehat{\sigma}_B} \mid \sigma_*] \right] \right).$$

Let us define the union graph $U := S_1 \cup S_2 \cup T_1 \cup T_1{}^3$. If $sD_+(a,b) > 1$, we view $\overline{A}^{\widehat{\sigma}}$ and $\overline{B}^{\widehat{\sigma}}$ as $\overline{A}$ and $\overline{B}$, respectively. $\mathbb{E}[\overline{A}_{S_1} \overline{B}_{S_2} \overline{A}_{T_1} \overline{A}_{T_2} \mid \sigma_*] \neq 0$ only if there exists no edge $e \in E(U)$ such that it occurs only once among $(S_1, S_2, T_1, T_2)$. This is because different edges are independent and centralized conditioned on $\sigma_*$.

However, in regime $sD_+(a,b) < 1$, $\mathbb{E}[\overline{A}_{S_1}^{\widehat{\sigma}_A} \overline{B}_{S_2}^{\widehat{\sigma}_B} \overline{A}_{T_1}^{\widehat{\sigma}_A} \overline{B}_{T_2}^{\widehat{\sigma}_B} \mid \sigma_*] \neq 0$ even when edges are not occurring multiple times as we have the expectation of $\overline{A}_e^{\widehat{\sigma}_A}$ can be non-zero when conditioning on $\sigma_*$ and an estimate $\widehat{\sigma}_A$ that disagrees with $\sigma_*$ on the edge type (i.e., for $e = (u,v), \sigma_*(u)\sigma_*(v) \neq \widehat{\sigma}_A(u)\widehat{\sigma}_A(v)$). This not only causes this cross-moments calculation being more complicated, but significantly increasing the possibility of the combinations between $S_1, S_2, T_1$, and $T_2$.

The most important high-level idea to properly bound the moments is: the cross-moment conditioning on a specific $\widehat{\sigma} = (\widehat{\sigma}_A, \widehat{\sigma}_B)$ would be non-trivial if and only if *all edges occurring only once are centralized incorrectly*. Assuming there are $z$ edges occurring once, we show that the probability that $\widehat{\sigma}$ satisfying this property is no greater than $n^{-\frac{z(sD_+(a,b) - \varepsilon \log(a/b)/2)}{D}}$ for any $\varepsilon > 0$, by using the definition of $\mathcal{T}$ and Lemma 8 later in Section K and Section L. It turns out that we require $n^{-\frac{z(sD_+(a,b) - \varepsilon \log(a/b)/2)}{D}}$ to be $o(\frac{1}{\log^C n})$ for some positive constant $C$ so that (1) and (2) are satisfied.

**Efficient algorithm.** Calculating $\Phi_{ij}^{\widehat{\sigma}}$ exactly (Algorithm 1) takes quasi-polynomial time as searching for all $S \cong H$ has time complexity $n^{\Theta(N)}$. Algorithm 4 in Section F computes an approximated score in polynomial time. Specifically, we follow the color-coding-based similarity score approximation from [41]. The basic idea is coloring the stochastic block model using $N + 1$ colors uniformly at random. Then, we only do signed counts on vertex sets that are colorful with $N + 1$ distinct colors. We show that this is an unbiased estimator of $\Phi_{ij}^{\widehat{\sigma}}$ and only potentially increase the variance by an additional constant factor in Section F. The result is stated as in Theorem 4.

**Theorem 4.** *Theorem 3 continues to hold with the color-coding sampled estimation $\widetilde{\Phi}_{ij}^{\widehat{\sigma}}$ in place of $\Phi_{ij}^{\widehat{\sigma}}$. Moreover, by Algorithm 4, $\{\widetilde{\Phi}_{ij}^{\widehat{\sigma}}\}_{i,j \in [n]}$ can be computed in $O(n^C)$ for some constant $C \equiv C(\varepsilon)$ depending only on $\varepsilon$, where $\varepsilon$ is from (3).*

---

${}^3$Note that $S_1$ and $T_1$ are rooted at $i$, while $S_2$ and $T_2$ are rooted at $j$. For $e = (u,v) \in E(S_1)$, we say it occurs in $T_1$ if $e \in E(T_1)$ and it occurs in $S_2$ (resp. $T_2$) if $(\sigma_*(u), \sigma_*(v)) \in E(S_2)$ (resp. $T_2$).

**Exact graph matching.** The final step of the algorithm is boosting the almost exact matching to a exact matching. The key idea is exploring the number of common neighbors.

Denote $N_\pi(i, j)$ as the number of common neighbors of $i$ and $j$ under correspondence $\pi$. The high-level idea is that if $i$ and $j$ form a true correspondence, then for a correct partial matching $\widetilde{\pi}$ on $(1 - o(1))n$ vertices, with high probability, $N_{\widetilde{\pi}}(i, j) \gtrsim \frac{p^2 + q^2}{2} s^2 (n + 2n^{\frac{3}{4}})$ under the nice event $\mathcal{H}$. Therefore, we match up $i$ and $j$ if they have more common neighbors than this threshold. Define $h(x) = x \log x - x + 1$. We give the following guarantee.

**Theorem 5.** *Fix $a \neq b > 0$ and $s \in [0, 1]$. Let $p = a\frac{\log n}{n}, q = b\frac{\log n}{n}$ and $(G_1, G_2) \sim$ CSBM$(n, p, q, s)$. Suppose $s^2(\frac{a+b}{2}) \geq 1 + \varepsilon$ and $s^2 \geq \alpha + \varepsilon$, for some $\varepsilon > 0$. Let $\gamma$ be the unique solution in $(1, \infty)$ to $h(\gamma) = \frac{3 \log n}{(n-2)pqs^2}$. Then, the seeded matching Algorithm 2 with input $\widehat{\pi}$ and an index set $I \subset [n], |I| = (1 - o(1))n \geq (1 - \varepsilon/16)n$ such that $\widehat{\pi}|_I = \pi_*$ outputs an exact matching $\widetilde{\pi} = \pi_*$ in $O(n^3(p + q)^2)$ time with probability $1 - o(1)$.*

**Putting these pieces together implies Theorem 1.** From Theorem 4, we know that for $(G_1, G_2) \sim$ CSBM$(n, a\frac{\log n}{n}, b\frac{\log n}{n}, s), a \neq b$, if $s^2 \geq \alpha + \varepsilon$ for some $\varepsilon > 0$ then there we can match $(1 - o(1))n$ vertices correctly and efficiently with high probability. Take the returned $\widehat{\pi}$ from Algorithm 4 as input of Algorithm 2, then Theorem 5 guarantees that the final output $\widetilde{\pi} = \pi_*$ with high probability.

# 6 Related work

**Graph Matching.** Correlated Erdős–Rényi graphs were first studied in [56] for social network de-anonymization. The information-theoretic threshold for partial matching was determined by [24, 27] and the information-theoretic threshold for exact graph matching was determined by [14, 15, 65].

Mao et al. [40] proposed the first efficient algorithm that achieves exact graph matching for correlated Erdős–Rényi graphs with average degree $(1 + \varepsilon) \log n \leq nq \leq n^{\frac{1}{\Theta(\log \log n)}}$ and constant noise. This algorithm only requires a constant edge correlation (sufficiently close to 1) rather than converging to 1, which represents a perfect correlation. Mao et al. [41] followed up with an improved efficient algorithm that achieves exact graph matching for any correlation $\rho$ satisfying $\rho^2 > \alpha$ when $nq(q + \rho(1 - q)) \geq (1 + \varepsilon) \log n$. It is conjectured that for random graphs of logarithmic average degree, $\rho^2 = \alpha$ is the computational threshold [41]. Muratori and Semerjian [50] added a small constant constraint on the maximum vertex degree of a chandelier to improve the runtime, at the expense of having a slightly larger constant $\widehat{\alpha}$ as the minimum squared correlation requirement.

In the denser regime where $p = n^{-a+o(1)}, a \in (0, 1]$, Ding and Du [17] established a sharp information-theoretic threshold for matching a positive fraction of vertices. Ding and Li [20] also developed an efficient algorithm for exact graph matching whenever the edge correlation is non-vanishing, which goes beyond the Otter's tree counting constant.

Several recent works also go beyond correlated Erdős–Rényi graphs. Wang et al. [64] studied the exact graph matching with additional attribute information on vanishing edge correlation.

Closely related to our work, Yang et al. [67] adopted the binary tree counting algorithm [40] to give an efficient graph matching algorithm for correlated SBMs. However, [67] makes several significant assumptions (which we do not). For one, the algorithm in [67] assumes that the community labels are known. This is a strong assumption which may be unrealistic in practice; moreover, this precludes using graph matching as a tool for improved community recovery. In contrast, we do not assume that community labels are known; in fact, a significant part of our technical work is devoted to dealing with the errors arising from estimating the community labels. Moreover, our graph matching algorithm can be directly applied to improve community recovery, as discussed in Theorem 2 and Section 3. In addition, [67] makes strong assumptions on the parameters, assuming that (1) the average degree is at least $(\log n)^{1.1}$, (2) the SBM has at least 3 communities, and (3) the correlation parameter satisfies $s > 1 - \varepsilon_0$ for some unspecified (small) $\varepsilon_0$. In contrast, our results hold in the most interesting regime of logarithmic average degree and the most natural setting of two balanced communities; moreover, our assumption on $s$ is also weaker.

**Community recovery with side information** Beyond correlated SBMs, there are some other models utilizing side information, from multiple networks [28, 62, 33, 30, 68, 71], additional covariates

[7], or both [44, 37]. Multi-layer SBM is first mentioned in [29], which is generated as following: first, generate the community labels for all vertices and fix them for all layers; second, form edges on each layer based on the community labels. Typically, different layers in a multi-layer SBM are conditionally independent given the shared community labels. In addition, several works [44, 37] also encode community membership correlated covariates onto each node. Aside from the multi-layer SBM, Braun and Sugiyama [7] recently studied community detection on a novel variation of SBM whose edges are attached with vectorial covariates.

# 7 Discussion and future work

Our main contribution in this paper is to give the first efficient algorithm for exact graph matching for correlated SBMs with two balanced communities, as well as a rigorous proof of its correctness (Theorem 1). We also discuss novel applications to community recovery (Theorem 2). At the same time, our work raises many interesting questions for future research, which we discuss here.

**Optimal runtime.** While our graph matching algorithm is efficient, it would be desirable to understand the optimal running time that can be achieved. Mao, Rudelson, and Tikhomirov [40] gave an efficient algorithm for matching correlated Erdős–Rényi graphs with runtime $n^{2+o(1)}$; the main drawback is that this algorithm requires the correlation parameter to satisfy $s > 1 - \varepsilon_0$ for some unspecified (small) $\varepsilon_0 > 0$. Nonetheless, it would be interesting to generalize this algorithm to correlated SBMs and the techniques developed in our work may be useful to do so. In very recent (and concurrent) work, Muratori and Semerjian [50] gave faster algorithms for matching correlated Erdős–Rényi graphs by introducing a constraint on the maximum degree of a chandelier, at the expense of strengthening the condition $s^2 > \alpha$ to $s^2 > \widehat{\alpha}$ for some $\widehat{\alpha} > \alpha$. Exploring the connections between our work and theirs, and generalizing their ideas to correlated SBMs, are of interest.

**Information-computation gap.** An important assumption throughout this work is that the correlation parameter satisfies $s^2 > \alpha$. We believe that this is inherently necessary and that there is no efficient algorithm (in the logarithmic average degree regime) when $s^2 < \alpha$. At the same time, exact graph matching is information-theoretically possible whenever $s^2(a+b)/2 > 1$, so there is a conjectured information-computation gap. This mirrors the conjecture in [41] for Erdős–Rényi graphs; see also [18] for the low-degree hardness results on the correlation detection and [10] for the very recent low-degree hardness results on testing a pair of correlated stochastic block models against a pair of independent Erdős–Rényi graphs.

**Efficient exact community recovery when exact graph matching is not possible.** Gaudio, Rácz, and Sridhar [25] determined the information-theoretic threshold for exact community recovery on correlated SBMs, in particular showing that there is a regime when this is possible even though (1) this is impossible with a single graph and also (2) exact graph matching is impossible. It remains unknown whether this can be done efficiently in this regime. We believe that this is possible, and our work is an important starting point for this question, yet additional ideas are needed to understand the subtle interplay between graph matching and community recovery in this regime.

**Sparser and denser regimes.** Our work focuses on the most interesting regime where the average degree is logarithmic in $n$; it is worth understanding other regimes too. In particular, the chandelier counting algorithm by Mao et al. [41] gives almost exact graph matching whenever the average degree diverges. In our Theorem 1 we require that the average degree diverges logarithmically for the corresponding result, so that the error rate for community recovery estimate is polynomially small in $n$. It would be interesting to overcome this technical barrier and extend the analysis to this sparser regime. Denser regimes are easier to understand. A close inspection of our analysis shows that it also works when the average degree diverges as a (small) polynomial in $n$; in even denser regimes, the community partition can be recovered exactly and efficiently whenever $\liminf_{n \to \infty} |p_n/q_n - 1| > 0$ (see [47]) and then the graph matching algorithm in [41] can be applied in a black-box fashion.

**General block models.** We focused here on the simplest case of SBMs with two balanced communities. It is of great interest to develop efficient graph matching algorithms in the general block model with $k$ communities, whenever this is possible. Recently, Yang and Chung [66] determined the information-theoretic threshold for exact graph matching in the $k$-community symmetric SBM, extending the results of Rácz and Sridhar [57]. We conjecture that substituting the community recovery algorithm used in our work with the degree-profiling algorithm by Abbe and Sandon [4] gives an efficient algorithm for graph matching in this more general setting, assuming again that $s^2 > \alpha$.

## Acknowledgements

We thank Julia Gaudio, Anirudh Sridhar, Yihong Wu, and Jiaming Xu for helpful discussions. We also thank anonymous reviewers for constructive feedback. S.C. was supported in part by the Institute for Data, Econometrics, Algorithms, and Learning (IDEAL), funded through the National Science Foundation TRIPODS Phase II program (NSF grant ECCS 2216970).

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

# A   Organization

The rest of this paper is organized as follows. In Section B, we give some notations used throughout. In Section C, we define the similarity score of a pair of vertices and give the formal proof of Theorem 1 based on Theorem 3 (Almost exact graph matching), Theorem 4 (Efficient algorithm for almost exact graph matching), and Theorem 5 (Exact graph matching by seeded graph matching). In Section D, we talk about some preliminaries: tail bounds, nice events, a tree node assigning sub-problem, an automorphism inequality for trees, and the calculation for cross-moments. These results will be repeatedly used in the following sections. In Section E, Section F, and Section G, we present the proofs for Theorem 3, Theorem 4, and Theorem 5, respectively. Section F and Section G are self-contained, while Section E contains several propositions whose proofs are deferred and which make up the remainder of the paper.

In Section E, we introduce six additional Propositions to show that under two different cases— namely, $sD_+(a,b) \geq 1$ and $sD_+(a.b) < 1$—the mean and variance of the similarity score are properly controlled. Specifically, if $sD_+(a,b) \geq 1$, then Proposition 1 gives the mean calculation of the similarity score, while Proposition 2 and Proposition 3 are about the variance calculation of the similarity score for true pairs and fake pairs of vertices. If $sD_+(a,b) < 1$, then Proposition 4 gives the mean calculation of the similarity score, while Proposition 5 and Proposition 6 are about the variance calculation of the similarity score for true pairs and fake pairs of vertices. The proofs of Proposition 1, Proposition 2, Proposition 3, Proposition 4, Proposition 5, and Proposition 6 are presented in Section H, Section I, Section J, Section K, Section L, and Section M, respectively.

# B   Notation

For any graph $G = (V, E)$, we denote $E(G)$ as the edge set and $V(G)$ as the vertex set. We let $e(G) := |E(G)|$ denote the number of edges of graph $G$ and $v(G) := |V(G)|$ denote the number of vertices in graph $G$. We define the excess of graph $G$ as $e(G) - v(G)$, the difference between the number of edges and the number of vertices of $G$.

Consider an arbitrary graph where vertices are equipped with two possible community labels $\{+1, -1\}$, we denote $V^+$ as the set of vertices with community label $+1$, $V^-$ as the set of vertices with community label $-1$, $\mathcal{N}(v)$ as the set vertices that are neighbors of $v$.

Let $\pi$ be a permutation on $[n]$, $(G_1, G_2) \sim \mathrm{CSBM}(n, p, q, s)$. Let $A$ (resp. $B$) be the adjacency matrix of $G_1$ (resp. $G_2$). Let $\overline{A} := A - \mathbb{E}[A]$ (resp. $\overline{B}$) be the centralized adjacency matrix. We further define the approximately centralized adjacency matrix with respect to community label estimate $\widehat{\boldsymbol{\sigma}}$ as $\overline{A}^{\widehat{\boldsymbol{\sigma}}_A} := A - E_A$, where $E_A$ is an $n \times n$ matrix whose $(i,j)$-th entry is $p$ if $\widehat{\boldsymbol{\sigma}}(i) = \widehat{\boldsymbol{\sigma}}(j)$ and $q$ otherwise.[4]

We denote $G_1 \vee_\pi G_2$ as the union graph with respect to $\pi$, such that $(i, j) \in E(G_1 \vee_\pi G_2)$ if and only if $(i, j) \in E(G_1)$ or $(\pi(i), \pi(j)) \in E(G_2)$. We denote $G_1 \wedge_\pi G_2$ as the intersection graph with respect to $\pi$, such that $(i, j) \in E(G_1 \wedge_\pi G_2)$ if and only if $(i, j) \in E(G_1)$ and $(\pi(i), \pi(j)) \in E(G_2)$.

We denote the variance of in-community edges $\sigma_+^2 := sp(1 - sp)$ and the variance of cross-community edges as $\sigma_-^2 := sq(1 - sq)$. Denote the correlation for in-community edges and cross-community edges as $\rho_+$ and $\rho_-$, respectively. In addition, we define $\rho := \frac{\rho_+ + \rho_-}{2}$. Consider the average degree being logarithmic in the number of vertices, then for some constants $a, b > 0$, $p = \frac{a \log n}{n}, q = \frac{b \log n}{n}$, $\rho = (1 + \Theta(\frac{\log n}{n}))\rho_+ = (1 + \Theta(\frac{\log n}{n}))\rho_- = (1 + \Theta(\frac{\log n}{n}))s$.[5]

Throughout the paper, we use standard asymptotic notation $O(\cdot), \Omega(\cdot), \Theta(\cdot), o(\cdot), \omega(\cdot)$. Any limitation is for $n \to \infty$ without special explanations. For real numbers $x, y$, we define $x \vee y := \max\{x, y\}$ and $x \wedge y := \min\{x, y\}$. In this paper, $\log$ is natural logarithmic function (with base $e$). Further notation is introduced in the following section which details the algorithms.

---

[4]See Section K for illustrations and more discussions on approximately centralized adjacency matrices.

[5]For arbitrarily small constant $\varepsilon > 0$, there exists $n$ large enough, such that $\rho^2 \geq \alpha + \varepsilon$ if and only if $s^2 \geq \alpha + \varepsilon$.

## C  Proof overview and proof of Theorem 1

### C.1  Chandelier

A line of works convert the graph matching problem from quadratic assumption to linear assignment by creating a signature vector $\boldsymbol{s}_i$ for each vertex $i \in [n]$, followed by calculating the similarity score $\Phi_{ij} = \langle \boldsymbol{s}_i^{(1)}, \boldsymbol{s}_j^{(2)} \rangle$ of all possible pairs of signatures on two graphs. Recently, Mao et al. [41] proposed a special tree family $\mathcal{T}$, chandelier, that shows a result of efficient graph matching under constant correlation.

**Definition 1** (($L, M, K, R$)–chandelier[41])**.** *An $(L, M, K, R)$-chandelier is a rooted tree with $L$ branches, each of which consists of a path with $M$ edges ($M$-wire), followed by a rooted tree with $K$ edges ($K$-bulb); the $K$-bulbs are non-isomorphic to each other and each of them has at most $R$ automorphisms.*

In this paper, we give an alternative definition of chandelier with five tuple. The first four parameters remain the same as $(L, M, K, R)$–chandelier. The last parameter $D$ stands for the maximum degree of vertices on this chandelier. We explain the necessity of controlling $D$ in the proof challenge.

**Definition 2** (($L, M, K, R, D$)–chandelier)**.** *An $(L, M, K, R, D)$-chandelier is a rooted tree with $L$ branches, each of which consists of a path with $M$ edges ($M$-wire), followed by a rooted tree with $K$ edges ($K$-bulb); the $K$-bulbs are non-isomorphic to each other, each of them has at most $R$ automorphisms, and the degree of each vertex is at most $D$.*

For each chandelier $H$, let $\mathcal{K}(H)$ denote the set of bulbs of $H$. For a rooted tree $T$, let $\mathrm{aut}(H)$ denote the number of rooted automorphisms of $T$ throughout this paper. We abbreviate rooted automorphism as automorphism when it is clear that we are applying it to a chandelier. The number of automorphisms of $H$ is determined by the automorphisms of its bulbs. Because all bulbs are non-isomorphic to each other,

$$\mathrm{aut}(H) = \prod_{\mathcal{B} \in \mathcal{K}(H)} \mathrm{aut}(\mathcal{B}). \tag{4}$$

Let $\mathcal{T}$ denote the family of non-isomorphic $(L, M, K, R, D)$–chandelier. The family size of chandelier is $|\mathcal{T}| = \binom{|\mathcal{J}|}{L}$, where $\mathcal{J} \equiv \mathcal{J}(K, R, D)$ denotes the collection of unlabeled rooted trees with $K$ edges, at most $R$ automorphisms, and maximum degree $D$.

Otter [55] showed that the number of unlabeled rooted trees with $K$ edges (and no constraint on the automorphisms and vertex degrees) is $|\mathcal{J}(K, \infty, \infty)| = (\alpha + o(1))^{-K}$, where $\alpha \approx 0.338$. We show that under proper choices of $R$ and $D$, we have $|\mathcal{J}(K, R, D)| = (\alpha + o(1))^{-K}$ through the following two Lemmas.

**Lemma 1.** *Let $K$ be the number of vertices on a unlabeled rooted tree, $C' > \frac{1}{\log(1/\alpha)} \approx 0.9227$, $D \geq C' \log K$. As $K \to \infty$,*

$$\frac{|\mathcal{J}(K, \infty, D)|}{|\mathcal{J}(K, \infty, \infty)|} = 1 - o(1). \tag{5}$$

*Proof.* Otter [55] characterized that $\frac{|\mathcal{J}(K,\infty,D)|}{|\mathcal{J}(K,\infty,\infty)|} \asymp \frac{\alpha_D^{-n}}{\alpha^{-n}}$, where $\alpha_D$ is the radius of convergence for the generating function of the number of unlabeled rooted trees whose maximum vertex degree less than or equal to $D$. Goh and Schumutz [26] (Theorem 7) showed the following property: as $D \to \infty$, for some constant $C > 0$, $\alpha_D = \alpha + C\alpha^D + o(\alpha^D)$. Immediately we can see that $\frac{|\mathcal{J}(K,\infty,D)|}{|\mathcal{J}(K,\infty,\infty)|} = (1 + O(\alpha^D))^{-K} = 1 - o(1)$ if $K\alpha^D \to 0$. Let $C' > \frac{1}{\log(1/\alpha)}$, choosing $D \geq C' \log K$ satisfies $K\alpha^D \to 0$. $\qquad\square$

**Lemma 2.** *Let $K$ be the number of vertices on a unlabeled rooted tree, $C$ be a constant and choose $R = \exp(CK)$. For sufficiently large $C$ and $K \to \infty$,*

$$\frac{|\mathcal{J}(K, R, \infty)|}{|\mathcal{J}(K, \infty, \infty)|} = 1 - o(1). \tag{6}$$

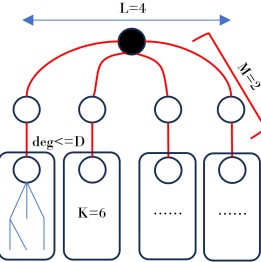

Figure 4: A chandelier.

*Proof.* Olsson and Wagner [53] (Theorem 2) showed a central limit theorem result for the number of automorphism on unlabeled rooted trees: $\frac{1}{\sqrt{K}}(\log \operatorname{aut}(H_K) - \mu K) \to \mathcal{N}(0, \sigma^2)$ as $K \to \infty$, where $H_K$ is a uniform random unlabeled rooted tree with $K$ edges and $\mu \approx 0.137, \sigma^2 \approx 0.197$. This implies that for some constant $C > \mu$ and $R = \exp(CK)$, $\operatorname{aut}(H_K) < R$ with high probability. $\qquad\square$

Putting together Lemma 1 and Lemma 2, and choosing $R$ and $D$ as specified, we have that $|\mathcal{J}(K, R, D)| = (1 - o(1))|\mathcal{J}(K, \infty, \infty)| = (\alpha + o(1))^{-K}$. Let $\beta$ denote a universal constant such that $|\mathcal{J}| \le \beta^K$. We take $\beta = \alpha^{-1}$.

### C.2  Algorithm overview

Given $(G_1, G_2) \sim \operatorname{CSBM}(n, p, q, s)$. Our algorithm contains mainly three steps. Firstly, we apply the algorithm by Mossel, Neeman, and Sly [47] to obtain almost exact community label estimates for each single graph. Secondly, we calculate the signed chandelier counting [41] based similarity score to give an almost exact graph matching. Lastly, we boost the almost exact matching to exact matching by extending the seeded graph matching algorithm [41] on Erdős–Rényi graphs to stochastic block models.

**Almost exact community recovery.** Obtaining community label estimates for both graph $G_1$ and $G_2$ is the first step of our algorithm. This is necessary for centralizing the adjacency matrices for the signed subgraph counts afterwards. We expect the following properties from the community recovery algorithm:

(a) gives almost exact recovery (down to the information-theoretic threshold, which is $s^2(a+b) = 1$ in the correlated SBMs with two balanced communities [57]);

(b) gives an error rate for each vertex of inverse-polynomial;

(c) gives error rates on different vertices that are approximately independent.

The community recovery algorithm in [47] (described as Algorithm 3) has been shown with property (a) by [47] and property (b) by [25] with an error rate of $n^{-sD_+(a,b)}$. In this paper, we show that property (c) is satisfied (See Lemma 8).

**Subgraph counts.** For an arbitrary weighted adjacency matrix $M$ of some adjacency matrix $A$, vertex $i \in [n]$, and a rooted graph $H$, we define the *weighted subgraph counts* on $M$ as

$$W_{i,H}(M) := \sum_{S(i) \cong H} M_S, \text{ where } M_S := \prod_{e \in E(S)} M_e, \tag{7}$$

and $S(i)$ enumerates subgraphs of the complete graph $K_n$, rooted at $i$, that are isomorphic to $H$.

When $M$ is the adjacency matrix itself, $W_{i,H}(M)$ is the usual subgraph count, representing the number of subgraphs rooted at $i$ in $M$ that are isomorphic to $H$. When $M$ is the centralized adjacency matrix $\overline{A} := A - \mathbb{E}[A]$, we call $W_{i,H}(M)$ a signed subgraph count following [9]. However, we do not have access to $\mathbb{E}[A]$ in many cases. Specifically for SBM, we can estimate $\mathbb{E}[A]$ through estimating the community labels. We define the *approximately centralized adjacency matrix* regarding to community label estimate $\widehat{\boldsymbol{\sigma}}$, denoted as $\overline{A}^{\widehat{\boldsymbol{\sigma}}}$, entry-wise as $\overline{A}^{\widehat{\boldsymbol{\sigma}}}_{i,j} = A_{i,j} - p\mathbf{1}_{\widehat{\boldsymbol{\sigma}}[i]=\widehat{\boldsymbol{\sigma}}[j]} - q\mathbf{1}_{\widehat{\boldsymbol{\sigma}}[i]\ne\widehat{\boldsymbol{\sigma}}[j]}$.

Using $\overline{A}^{\widehat{\boldsymbol{\sigma}}}$ in (7) yields the weighted subgraph counts for approximately centralized adjacency matrix. We also refer to this as a signed subgraph count, though errors may exist.

Given a family $\mathcal{H}$ of non-isomorphic rooted graphs, we define the *subgraph count signature* of vertex $i$ as

$$W_i^{\mathcal{H}}(M) := (W_{i,H}(M))_{H \in \mathcal{H}}. \tag{8}$$

**Similarity score.** Given a pair of correlated SBMs $(G_1, G_2)$, we define the similarity score between vertex $i$ on graph $G_1$ and vertex $j$ on graph $G_2$ as a weighted inner product between two signatures:

$$\Phi_{ij} := \langle W_i^{\mathcal{T}}(\overline{A}), W_j^{\mathcal{T}}(\overline{B}) \rangle := \sum_{H \in \mathcal{T}} \mathrm{aut}(H) W_{i,H}(\overline{A}) W_{j,H}(\overline{B}), \tag{9}$$

where $\mathcal{T}$ is the family of chandelier.

When we do not have access to the centralized adjacency matrices $\overline{A}$ and $\overline{B}$, we use community label estimates $\widehat{\boldsymbol{\sigma}}_A$ and $\widehat{\boldsymbol{\sigma}}_B$ for $G_1$ and $G_2$ correspondingly. We define the similarity score with a slightly different notation:

$$\Phi_{ij}^{\widehat{\boldsymbol{\sigma}}} := \langle W_i^{\mathcal{T}}(\overline{A}^{\widehat{\boldsymbol{\sigma}}_A}), W_j^{\mathcal{T}}(\overline{B}^{\widehat{\boldsymbol{\sigma}}_B}) \rangle = \sum_{H \in \mathcal{T}} \mathrm{aut}(H) W_{i,H}(\overline{A}^{\widehat{\boldsymbol{\sigma}}_A}) W_{j,H}(\overline{B}^{\widehat{\boldsymbol{\sigma}}_B}). \tag{10}$$

**Almost exact graph matching.** The first part in the analysis is to show that by calculating this similarity score, with an appropriate thresholding strategy, we can match up $(1 - o(1))n$ vertices correctly (Theorem 3). The high-level idea is to show that the similarity score distributions are well-separated between true pairs and fake pairs. We expect the similarity score having the following properties, under event $\mathcal{H}$:

- For true pairs $j = \pi(i)$:

$$\mathbb{E}[\Phi_{i\pi_*(i)}^{\widehat{\boldsymbol{\sigma}}} \mathbf{1}_{\mathcal{H}}] > 0, \quad \mathrm{Var}[\Phi_{ij}^{\widehat{\boldsymbol{\sigma}}} \mathbf{1}_{\mathcal{H}}] = o\left( \mathbb{E}[\Phi_{i\pi_*(i)}^{\widehat{\boldsymbol{\sigma}}} \mathbf{1}_{\mathcal{H}}]^2 \right),$$

- For fake pairs $j \neq \pi(i)$:

$$\mathbb{E}[\Phi_{ij}^{\widehat{\boldsymbol{\sigma}}} \mathbf{1}_{\mathcal{H}}] = o\left( \mathbb{E}[\Phi_{i\pi_*(i)}^{\widehat{\boldsymbol{\sigma}}} \mathbf{1}_{\mathcal{H}}] \right), \quad \mathrm{Var}[\Phi_{ij}^{\widehat{\boldsymbol{\sigma}}} \mathbf{1}_{\mathcal{H}}] = o\left( \frac{\mathbb{E}[\Phi_{i\pi_*(i)}^{\widehat{\boldsymbol{\sigma}}} \mathbf{1}_{\mathcal{H}}]^2}{n^2} \right).$$

Precisely forming bounds for these moments constitutes the main bulk of the paper. Aside from proving the desired properties of the first and second order moments, we follow the color-coding-based similarity score estimation idea from [41] to analyze an efficient algorithm. The basic idea is to color the vertices of SBMs using $N + 1$ colors uniformly at random. Then, we only do signed counts on vertex sets that are colorful with $N + 1$ colors. We show that this is an unbiased estimator and only potentially increase the variance by an additional constant factor in Section F. The result is stated formally as in Theorem 4.

---

**Algorithm 1** Almost Exact Graph Matching for CSBM

---

**Input:** Adjacency matrices $A$ and $B$ for $(G_1, G_2) \sim \mathrm{CSBM}(n, a\frac{\log n}{n}, b\frac{\log n}{n}, s)$, a constant $c$, and mean value $\mu$.
**Output:** A mapping $\widehat{\pi} : I \to [n]$.

1: Run community recovery Algorithm 3 [47] on $A$ and $B$ separately and get label vector $\widehat{\boldsymbol{\sigma}}_A, \widehat{\boldsymbol{\sigma}}_B$.
2: For each pair of vertex $i$ in $\overline{A}^{\widehat{\boldsymbol{\sigma}}_A}$ and vertex $j$ in $\overline{B}^{\widehat{\boldsymbol{\sigma}}_B}$, compute their similarity score as in [41]:

$$\Phi_{ij}^{\widehat{\boldsymbol{\sigma}}} = \langle W_i^{\mathcal{T}}(\overline{A}^{\widehat{\boldsymbol{\sigma}}_A}), W_j^{\mathcal{T}}(\overline{B}^{\widehat{\boldsymbol{\sigma}}_B}) \rangle = \sum_{H \in \mathcal{T}} \mathrm{aut}(H) W_{i,H}(\overline{A}^{\widehat{\boldsymbol{\sigma}}_A}) W_{j,H}(\overline{B}^{\widehat{\boldsymbol{\sigma}}_B}). \tag{11}$$

3: Let $\tau = c\mu$, output $I := \{i | i \in [n], \exists j \in [n], \text{s.t. } \Phi_{ij}^{\mathcal{T}} \geq \tau, \text{ and } \forall k \in [n] \setminus \{j\}, \Phi_{ik}^{\mathcal{T}} < \tau\}$.

---

**Proof challenge.** In regime $sD_+(a, b) > 1$, the probability of existing one vertex being classified incorrectly is vanishing. Therefore, with high probability, $\overline{A}^{\widehat{\boldsymbol{\sigma}}_A} = \overline{A}$. If $sD_+(a, b) < 1$, then

the recovered $\widehat{\boldsymbol{\sigma}}$ contains errors (polynomial in $n$), which will cause some edges being centralized incorrectly and thereby affect the moments calculation. For example, let $i, j \in [n]$ be two vertices on $G_1$ who has the same community label $\boldsymbol{\sigma}_*(i) = \boldsymbol{\sigma}_*(j)$. If only one of $i, j$ is labeled incorrectly by Algorithm 3, then the expectation of $\overline{A}_{i,j}^{\widehat{\boldsymbol{\sigma}}_A}$ conditioned on $\boldsymbol{\sigma}_*$ and $\widehat{\boldsymbol{\sigma}}$ is $p - q$.

This phenomenon poses a challenge to the algorithm analysis. We highlight some key points in the context of second moment calculation. For simplicity, we ignore event $\mathcal{H}$ here.

$$\mathrm{Var}[\Phi_{ij}^{\widehat{\boldsymbol{\sigma}}}] = \sum_{H,I \in \mathcal{T}} \mathrm{aut}(H)\mathrm{aut}(I) \sum_{S_1(i),S_2(j) \cong H} \sum_{T_1(i),T_2(j) \cong I} \left( \mathbb{E}_{\boldsymbol{\sigma}_*}\left[ \mathbb{E}[\overline{A}_{S_1}^{\widehat{\boldsymbol{\sigma}}_A} \overline{B}_{S_2}^{\widehat{\boldsymbol{\sigma}}_B} \overline{A}_{T_1}^{\widehat{\boldsymbol{\sigma}}_A} \overline{B}_{T_2}^{\widehat{\boldsymbol{\sigma}}_B} \mid \boldsymbol{\sigma}_* ] \right] \right.$$
$$\left. - \mathbb{E}_{\boldsymbol{\sigma}_*}\left[ \mathbb{E}[\overline{A}_{S_1}^{\widehat{\boldsymbol{\sigma}}_A} \overline{B}_{S_2}^{\widehat{\boldsymbol{\sigma}}_B} \mid \boldsymbol{\sigma}_* ] \right] \mathbb{E}_{\boldsymbol{\sigma}_*}\left[ \mathbb{E}[\overline{A}_{T_1}^{\widehat{\boldsymbol{\sigma}}_A} \overline{B}_{T_2}^{\widehat{\boldsymbol{\sigma}}_B} \mid \boldsymbol{\sigma}_* ] \right] \right).$$

Let us define the union graph $U := S_1 \cup S_2 \cup T_1 \cup T_1$[6]. If $sD_+(a,b) > 1$, we view $\overline{A}^{\widehat{\boldsymbol{\sigma}}}$ and $\overline{B}^{\widehat{\boldsymbol{\sigma}}}$ as $\overline{A}$ and $\overline{B}$, respectively. $\mathbb{E}[\overline{A}_{S_1} \overline{B}_{S_2} \overline{A}_{T_1} \overline{A}_{T_2} \mid \boldsymbol{\sigma}_*] \neq 0$ only if there exists no edge $e \in E(U)$ such that it occurs only once among $(S_1, S_2, T_1, T_2)$. This is because different edges are independent and centralized conditioned on $\boldsymbol{\sigma}_*$. Without loss of generality, we assume there exists an edge $e \in E(A)$ occur only on $S_1$, thus $\mathbb{E}[\overline{A}_e \mid \boldsymbol{\sigma}_*] = 0$ and also $\mathbb{E}[\overline{A}_{S_1} \overline{B}_{S_2} \overline{A}_{T_1} \overline{A}_{T_2} \mid \boldsymbol{\sigma}_*] = 0$.

However, in regime $sD_+(a,b) < 1$, $\mathbb{E}[\overline{A}_{S_1}^{\widehat{\boldsymbol{\sigma}}_A} \overline{B}_{S_2}^{\widehat{\boldsymbol{\sigma}}_B} \overline{A}_{T_1}^{\widehat{\boldsymbol{\sigma}}_A} \overline{B}_{T_2}^{\widehat{\boldsymbol{\sigma}}_B} \mid \boldsymbol{\sigma}_*] \neq 0$ even when edges are not occurring multiple times as we have the expectation of $\overline{A}_e^{\widehat{\boldsymbol{\sigma}}_A}$ can be non-zero when conditioning on $\boldsymbol{\sigma}_*$ and an estimate $\widehat{\boldsymbol{\sigma}}_A$ that disagrees with $\boldsymbol{\sigma}_*$ on the edge type (i.e., for $e = (u,v)$, $\boldsymbol{\sigma}_*(u)\boldsymbol{\sigma}_*(v) \neq \widehat{\boldsymbol{\sigma}}_A(u)\widehat{\boldsymbol{\sigma}}_A(v)$). This not only causes this cross-moments calculation being more complicated, but significantly increasing the possibility of the combinations between $S_1, S_2, T_1$, and $T_2$. The maximum vertex in the union graph grows from $2N$ to $4N$, squaring up the trivial bound on the number of subgraphs on the complete graph $K_n$ that is isomorphic to $U$.

The most important high-level idea to properly bound the moments is: the cross-moment conditioning on a specific $\widehat{\boldsymbol{\sigma}} = (\widehat{\boldsymbol{\sigma}}_A, \widehat{\boldsymbol{\sigma}}_B)$ would be non-trivial if and only if *all edges occurring only once are centralized incorrectly*. This is because, conditioning on $\widehat{\boldsymbol{\sigma}}$ satisfying the above property, the expectation of $\overline{A}_e$ takes either $p - q$ or $q - p$ for all $e \in E(U)$ that occurs only once. Assuming there are $z$ edges occurring once, we show that the probability that $\widehat{\boldsymbol{\sigma}}$ satisfying this property is no greater than $n^{-\frac{z(sD_+(a,b)-\varepsilon \log(a/b)/2)}{D}}$ for any $\varepsilon > 0$. Intuitively, this is saying that to incorrectly centralize $z$ edges at the same time, we expect the Algorithm 3 to label at least $\lceil \frac{z}{D} \rceil$ vertices incorrectly. It turns out that we require $n^{-\frac{z(sD_+(a,b)-\varepsilon \log(a/b)/2)}{D}}$ to be $o(\frac{1}{\log^C n})$ for some positive constant $C$ so that (1) and (2) are satisfied.

**Theorem 3.** *Fix $a \neq b > 0$ and $s \in [0,1]$. Let $p = a\frac{\log n}{n}, q = b\frac{\log n}{n}$ and $(G_1, G_2) \sim$ CSBM$(n, p, q, s)$. For any $\varepsilon > 0$, suppose $s^2 \geq \alpha + \varepsilon$. There exists positive constants $C_1, C_2, C_3, C_4, C_5 > 0$ such that the following holds. Pick $K, M, L, N, D$ as*

$$L = \frac{C_1}{\varepsilon}, \quad K = C_2 \log n, \quad M = \frac{C_3 K}{\log(ns(p \wedge q))}, \quad R = \exp(C_4 K), \quad D = C_5 \frac{\log n}{(\log \log n)^2}. \tag{12}$$

*Pick an arbitrary $c \in (0, 1)$ and set $\mu = |\mathcal{T}|n^N \rho^N \sigma_{\mathrm{eff}}^{2N}$, where $\sigma_{\mathrm{eff}}^2 := (\frac{\sigma_+^2 + \sigma_-^2}{2})$. Then, Algorithm 1 outputs a set $I$ with size $(1 - o(1))n$ and a mapping $\widehat{\widehat{\pi}}$ such that $\widehat{\widehat{\pi}}|_I = \widehat{\pi}_*|_I$ with high probability.*

Algorithm 1 takes quasi-polynomial time. Algorithm 4 in Section F computes an approximated score in polynomial time and satisfies the following Theorem.

**Theorem 4.** *Theorem 3 continues to hold with the color-coding sampled estimation $\widetilde{\Phi}_{ij}^{\widehat{\boldsymbol{\sigma}}}$ in place of $\Phi_{ij}^{\widehat{\boldsymbol{\sigma}}}$. Moreover, by Algorithm 4, $\{\widetilde{\Phi}_{ij}^{\widehat{\boldsymbol{\sigma}}}\}_{i,j \in [n]}$ can be computed in $O(n^C)$ for some constant $C \equiv C(\varepsilon)$ depending only on $\varepsilon$, where $\varepsilon$ is from (3).*

**Exact graph matching.** The final step of the algorithm is boosting the almost exact matching to a exact matching. The key idea is exploring the number of common neighbors for two unmatched vertices with regard to the current matching.

---

[6]Note that $S_1$ and $T_1$ are rooted at $i$, while $S_2$ and $T_2$ are rooted at $j$. For $e = (u,v) \in E(S_1)$, we say it occurs in $T_1$ if $e \in E(T_1)$ and it occurs in $S_2$ (resp. $T_2$) if $(\boldsymbol{\sigma}_*(u), \boldsymbol{\sigma}_*(v)) \in E(S_2)$ (resp. $T_2$).

Denote $N_\pi(i,j)$ as the number of common neighbors of $i$ and $j$ under correspondence $\pi$. In another word, $N_\pi(i,j)$ is the number of vertex $v \in I$ such that $v$ is a neighbor of $i$ in $G_1$ and $\pi(v)$ is a neighbor of $j$ in $G_2$. The high-level idea is that if $i$ and $j$ form a true correspondence, then for a correct partial matching $\widetilde{\pi}$ on $(1-o(1))n$ vertices, with high probability, $N_{\widetilde{\pi}}(i,j) \gtrsim \frac{p^2+q^2}{2}s^2(n+2n^{\frac{3}{4}})$ under the nice event $\mathcal{H}$. Therefore, we match up $i$ and $j$ if they have more common neighbors than this threshold. In addition, we can show that all the remaining vertices will be matched up with high probability. Formally, define $h(x) = x\log x - x + 1$, we summarize the algorithm as Algorithm 2 and the guarantee as Theorem 5.

**Theorem 5.** *Fix $a \neq b > 0$ and $s \in [0,1]$. Let $p = a\frac{\log n}{n}, q = b\frac{\log n}{n}$ and $(G_1, G_2) \sim$ CSBM$(n,p,q,s)$. Suppose*

$$s^2(\frac{a+b}{2}) \geq 1 + \varepsilon \quad and \quad s^2 \geq \alpha + \varepsilon,$$

*for some $\varepsilon > 0$. Let $\gamma$ be the unique solution in $(1,\infty)$ to $h(\gamma) = \frac{3\log n}{(n-2)pqs^2}$. Then, the seeded matching Algorithm 2 with input $\widehat{\pi}$ and an index set $I \subset [n], |I| = (1-o(1))n \geq (1-\varepsilon/16)n$ such that $\widehat{\pi}|_I = \pi_*$ outputs an exact matching $\widetilde{\pi} = \pi_*$ in $O(n^3(p+q)^2)$ time with probability $1-o(1)$.*

---

**Algorithm 2** Seeded Graph Matching [41]

---

**Input:** Adjacency matrices $A$ and $B$ for $(G_1, G_2) \sim$ CSBM$(n,p,q,s), p = a\frac{\log n}{n}, q = b\frac{\log n}{n}$ for some $a > 0, b > 0$. A mapping $\widehat{\pi}: I \to [n]$ with $|I| = (1-o(1))n$, parameters $p, q, s$, and $\gamma \in (1,\infty)$ such that $h(\gamma) = \frac{3\log n}{(n-2)pqs^2}$.

1: Let $J = I$, and $\widetilde{\pi} = \widehat{\pi}$.
2: **while** there exists $i \notin J$ and $j \notin \widetilde{\pi}(J)$ such that $N_{\widetilde{\pi}}(i,j) \geq \gamma\frac{p^2+q^2}{2}s^2(n+2n^{\frac{3}{4}})$ **do**
3:     Add $i$ to $J$ and let $\widetilde{\pi}(i) = j$.
4: **end while**

**Output:** $\widetilde{\pi}$.

---

### C.3 Putting things together: Proof of Theorem 1

The proof of Theorem 1 follows from Theorem 4 and Theorem 5.

*Proof of Theorem 1.* From Theorem 4, we know that for $(G_1, G_2) \sim$ CSBM$(n, a\frac{\log n}{n}, b\frac{\log n}{n}, s)$, $a \neq b$, if $s^2 \geq \alpha + \varepsilon$ for some $\varepsilon > 0$ then there we can match $(1-o(1))n$ vertices correctly and efficiently with high probability.

Take the returned $\widehat{\pi}$ from Algorithm 4 as input of Algorithm 2, then Theorem 5 guarantees that the final output $\widetilde{\pi} = \pi_*$ with probability $1-o(1)$. This completes the proof of Theorem 1. $\qquad\square$

## D Preliminaries

### D.1 Tail bounds

**Lemma 3** (Chernoff Bound, Theorem 2.1 of [31]). *Let $X \sim$ Binom$(n,p)$ be a binomial random variable. Then, for all $t \geq 0$,*

$$\mathbb{P}(X \geq \mathbb{E}X + t) \leq \exp\left(-\frac{t^2}{2(\mathbb{E}X + t/3)}\right),$$

$$\mathbb{P}(X \leq \mathbb{E}X - t) \leq \exp\left(-\frac{t^2}{2\mathbb{E}X}\right).$$

**Lemma 4** (Multiplicative Chernoff Bound, Theorem 4.4 and Theorem 4.5 of [45]). *Let $X \sim$ Binom$(n,p)$ be a binomial random variable, denote $\mu = np$ as the mean. Let $h(x) = x\log x - x + 1$. Then, for all $\gamma \in (1,\infty)$,*

$$\mathbb{P}(X \geq \gamma\mu) \leq \exp(-\mu h(\gamma)).$$

*For $\gamma \in (0, 1)$,*

$$\mathbb{P}(X \leq \gamma\mu) \leq \exp(-\mu h(\gamma)).$$

**Lemma 5.** *[47, 25] Suppose that $a < b$. Let $Y \sim \mathrm{Binom}(m_+, \frac{a\log n}{n})$ and $Z \sim \mathrm{Binom}(m_-, \frac{b\log n}{n})$ be independent. If $m_+ = (1 + o(1))\frac{n}{2}$, $m_- = (1 + o(1))\frac{n}{2}$, then,*

$$\mathbb{P}(Y < Z) = n^{-D_+(a,b)+o(1)}.$$

*For any $\varepsilon > 0$,*

$$\mathbb{P}(Y - Z \leq \varepsilon\log n) \leq n^{-(D_+(a,b)-\frac{\varepsilon\log(a/b)}{2})+o(1)}.$$

This result has been proved in Lemma 8 in [3] and Lemma 3.3 in [25].

## D.2 Nice events

When $(G_1, G_2) \sim \mathrm{CSBM}(n, \frac{a\log n}{n}, \frac{b\log n}{n}, s)$, there are some events that happen with high probability and our following analysis intuitively relies on the happening of these nice events.

- **(Balanced Communities)** We denote $\mathcal{H} := \{\frac{n}{2} - n^{\frac{3}{4}} \leq |V^+|, |V^-| \leq \frac{n}{2} + n^{\frac{3}{4}}\}$. We observe that $|V^+| \sim \mathrm{Binom}(n, \frac{1}{2})$ and $|V^-| = n - |V^+|$. By Chernoff bound,

$$\mathbb{P}(\mathcal{H}^c) = \mathbb{P}(|V^+| \geq \frac{n}{2} + n^{\frac{3}{4}}) - \mathbb{P}(|V^+| \leq \frac{n}{2} - n^{\frac{3}{4}}) \leq \frac{1}{e^{(1-o(1))\sqrt{n}}}.$$

- **(Reasonable Large Neighborhood)** Let $\gamma = \max(a, b)$, we also denote that

$$\mathcal{F} = \{\forall v \in [n], |\mathcal{N}(v)| \leq 100\max(1, \gamma)\log^3 n\}.$$

**Lemma 6.** *As $n \to \infty$, we have*

$$\mathbb{P}(\mathcal{F}) \geq 1 - n^{-O(\log^2 n)}.$$

*Proof.* Let $X \sim \mathrm{Binom}(n, \gamma\frac{\log n}{n})$. Fix $i \in [n]$, conditioned on any $\boldsymbol{\sigma}_*$, $|\mathcal{N}(i)|$ is stochastically dominated by $X$. Therefore,

$$\mathbb{P}(|\mathcal{N}(i)| \geq \gamma\log^3 n \mid \boldsymbol{\sigma}_*) \leq \mathbb{P}(X \geq \gamma\log^3 n) \leq \exp\left(-\frac{(\gamma\log^3 n)^2}{2\gamma\log n + 2/3\gamma\log^3 n}\right)$$

$$\leq \exp(-\gamma\log^3 n) = n^{-\gamma\log^2 n},$$

where the second line uses Bernstein's inequality and the third line holds for any $n$ such that $\log^2 n > 6$. From an union bound, we have

$$\mathbb{P}(\mathcal{F}) = \mathbb{E}[\mathbb{P}(\mathcal{F}|\boldsymbol{\sigma}_*)] \geq 1 - n^{-O(\log^2 n)}. \qquad \square$$

## D.3 Community recovery

We make a slight change on the choice of the partition number of the community detection algorithm proposed by Mossel, Neeman, and Sly [47]. This algorithm gives almost exact recovery with $n^{1-sD_+(a,b)+\varepsilon|\log(a/b)|}$ vertices labeled incorrectly. After community recovery, we need to match the two communities in $G_1$ and $G_2$ by applying the community matching Algorithm 3.

Consider $G \sim \mathrm{SBM}(n, \frac{a\log n}{n}, \frac{b\log n}{n})$. Define $\gamma = \max\{a, b\}$. For any vertex $v \in [n]$, we define the signed neighbor counts of $v$ in $G$ as

$$\mathrm{maj}_G(v) = \boldsymbol{\sigma}_*(v)\sum_{u\in\mathcal{N}(v)}\boldsymbol{\sigma}_*(u).$$

For any $\varepsilon > 0$, define a set of vertices:

$$I_\varepsilon(G) := \{v \in [n] : \mathrm{maj}_G(v) \leq \varepsilon\log n \text{ or } |\mathcal{N}(v)| \geq \gamma\log^3 n\}.$$

Previous results (Lemma 5.1 [25], Proposition 4.3 [47]) have shown that Algorithm 3's correctness on $[n] \setminus I_\varepsilon(G)$ with a different choice of $m$ and the lower bound of $|\mathcal{N}(v)|$ in the bad vertices set $I_\varepsilon(G)$. In our work, we first demonstrates that Algorithm 3 on input $(G, a, b, \varepsilon)$ correctly classifies all vertices in $[n] \setminus I_\varepsilon(G)$.

---

**Algorithm 3** Almost-exact Community Recovery [47]

---

**Input:** Adjacency matrix $A$ on $n$ vertices; parameters $a, b, \varepsilon > 0$.
**Output:** A community label estimate $\widehat{\boldsymbol{\sigma}} \in \{-1, +1\}^n$.

1: Choose a positive integer $m$ satisfying $(\log(\varepsilon m(2 \max(a, b) \log^2 n)^{-1}) - 1)\varepsilon/2 > 1$. Initialize two empty sets, $W_+$ and $W_-$.
2: Using the spectral method of [3] to find a community partition of $[n]$, denoted as $(U_+, U_-)$.
3: Partition $[n]$ into $\{U_1, \ldots, U_m\}$ uniformly at random.
4: **for** $i \in [m]$ **do**
5:     Using the spectral method of [3] to find a community partition $(U_{i,+}, U_{i,-})$ of $G\{[n] \setminus U_i\}$. If $|U_{i,+} \Delta U_+| \geq n/2$, then swap $U_{i,+}$ and $U_{i,-}$.
6:     For $v \in U_i$, insert $v$ into $W_+$ or $W_-$ according to its neighborhood majority (resp., minority) in $U_{i,+} \cup U_{i,-}$ if $a > b$ (resp. $a < b$).
7: **end for**
8: For $i \in W_+$, set $\widehat{\boldsymbol{\sigma}}(i) = 1$, and for $i \in W_-$, set $\widehat{\boldsymbol{\sigma}}(i) = -1$. Return $\widehat{\boldsymbol{\sigma}}$.

---

**Lemma 7.** *Algorithm 3 on input $(n, a, b, s)$ classifies all vertices in $[n] \setminus I_\varepsilon(G)$ correctly with high probability.*

The proof directly follows from the proof of Proposition 5.1 in [25], with two remarks. First, although the maximum size of neighbors $|\mathcal{N}(i)|$ we consider here is enlarged from $100 \max\{1, \gamma\} \log n$ to $\gamma \log^3 n$, we adjust the condition of $m$ accordingly such that the tail bound holds. Secondly, we need to justify that with high probability, all partitions done by the spectral method are still almost exactly correct under the new choice of $m$, which is no longer a constant independent of $n$. Theorem 3.2 of [3] showed that the vanilla spectral method achieved optimal error rate, in the sense that $\mathbb{E}[\frac{1}{n} \sum_{i=1}^{n} 1_{\{\boldsymbol{\sigma}_*(i) \neq \widehat{\boldsymbol{\sigma}}(i)\}}] \leq n^{-(1+o(1))D_+(a,b)}$. This implies that with probability $1 - n^{-(1+o(1))D_+(a,b)}$, the spectral method labels all but $o(n)$ vertices correctly. Furthermore, for all $i \in [n]$, $U_{i,+}$ matches with $V^+ \setminus U_i$ and $U_{i,-}$ matches with $V^- \setminus U_i$ on all but $o(n)$ vertices after step (5) with high probability.

In this work, we further determine the probability of a set of vertices being in the set $I_\varepsilon(G)$, which is a generalization of the result on the $\mathbb{P}(v \in I_\varepsilon)$ for an arbitrary $v \in [n]$ (Lemma 5.3 of [25]).

**Lemma 8.** *Given a random graph $G \sim \mathrm{SBM}(n, \frac{sa \log n}{n}, \frac{sb \log n}{n})$ and a fixed subgraph induced by vertex set $S \in [n]$.*

*If $|S| = O(\log n)$, then for any $\varepsilon > 0, \delta > 0$,*

$$\mathbb{P}(\{\forall i \in S, i \in I_\varepsilon\} \cap \mathcal{H}) = O(n^{-|S|(sD_+(a,b)-\varepsilon(1+\delta)|\log(a/b)|)} + n^{-\varepsilon\delta(1-o(1))\log n}).$$

*If $|S| = o(\log n)$, then for any $\varepsilon > 0, \delta > 0$,*

$$\mathbb{P}(\{\forall i \in S, i \in I_\varepsilon\} \cap \mathcal{H}) = O(n^{-|S|(sD_+(a,b)-\varepsilon(1+\delta)|\log(a/b)|)}).$$

*Proof.* The main idea is considering the intersection of the interested event $\{\forall i \in S, i \in I_\varepsilon\} \cap \mathcal{H}$ with $\mathcal{G} = \{\forall i \in S, |\mathcal{N}_S(i)| \leq \varepsilon\delta \log n\}$, where $\mathcal{N}_S(v)$ denotes the set of neighbors of $v$ restricted on the vertex set $S$.

$$\mathbb{P}(\{\forall i \in S, i \in I_\varepsilon\} \cap \mathcal{H}) \leq \mathbb{P}(\{\forall i \in S, i \in I_\varepsilon\} \cap \mathcal{H} \cap \mathcal{F} \cap \mathcal{G}) + \mathbb{P}(\mathcal{G}^c) + \mathbb{P}(\mathcal{F}^c). \quad (13)$$

Firstly, we give the upper bound of $\mathbb{P}(\mathcal{G}^c \mid \boldsymbol{\sigma}_*)1_{\mathcal{H}}$. Assume that $|S| \geq \varepsilon\delta \log n$ without loss of generality. By using an union bound,

$$\mathbb{P}(\mathcal{G}) \leq |S| \times \mathbb{E}[\mathbb{P}(|\mathcal{N}_S(i)| > \varepsilon\delta \log n \mid \boldsymbol{\sigma}_*)]$$

$$= |S| \times \sum_{k=\delta \log n}^{|S|} \binom{|S|}{\varepsilon\delta \log n} (\frac{a \log n}{n})^{\varepsilon\delta \log n} (1 - \frac{b \log n}{n})^{|S|-\varepsilon\delta \log n}$$

$$= n^{\frac{\log \log n}{\log n}} (\frac{Ca \log^2 n}{n})^{\varepsilon\delta \log n} = O(n^{-\varepsilon\delta(1-o(1))\log n}), \quad (14)$$

where the second equation holds from the assumption of $|S| = O(\log n)$.

Secondly, we study this event $E = \{\forall i \in S, i \in I_\varepsilon\} \cap \mathcal{H} \cap \mathcal{F}$.

$$
\begin{aligned}
\mathbb{P}(E) &= \mathbb{E}[\mathbb{P}(\{\forall v \in S, \mathrm{maj}_G(v) \leq \varepsilon \log n\} \mid \boldsymbol{\sigma}_*)\mathbf{1}_\mathcal{H}] \\
&= \mathbb{E}[\mathbb{P}(\{\forall v \in S, \mathrm{maj}_{G[[n]\setminus S]}(v) + \mathrm{maj}_{G[S]}(v) \leq \varepsilon \log n\} \mid \boldsymbol{\sigma}_*)\mathbf{1}_\mathcal{H}] \\
&\leq \mathbb{E}[\mathbb{P}(\{\forall v \in S, \mathrm{maj}_{G[[n]\setminus S]}(v) \leq \varepsilon(1+\delta) \log n\} \mid \boldsymbol{\sigma}_*)\mathbf{1}_\mathcal{H}].
\end{aligned}
$$

For any $v \in S$, $\mathrm{maj}_{G[[n]\setminus S]}(v)$ is the difference of two independent binomial random variables $Y_v$ and $Z_v$, where $Y_v \sim \mathrm{Binom}(|V^{\boldsymbol{\sigma}_*(v)}_{G[[n]\setminus S]}|, sa\frac{\log n}{n})$, $Z_v \sim \mathrm{Binom}(|V^{-\boldsymbol{\sigma}_*(v)}_{G[[n]\setminus S]}|, sb\frac{\log n}{n})$. With $\mathcal{H}$ happening, we have $|V^{\boldsymbol{\sigma}_*(v)}_{G[[n]\setminus S]}| = (1 - o(1))\frac{n}{2}, |V^{-\boldsymbol{\sigma}_*(v)}_{G[[n]\setminus S]}| = (1 - o(1))\frac{n}{2}$. Since $Y_v$ and $Z_v$ do not take into account $v \in S$, they are independent for all $v \in S$. Therefore,

$$
\begin{aligned}
\mathbb{P}(E) &= \mathbb{E}[\Pi_{v \in S}\mathbb{P}(\{Y_v - Z_v \leq \varepsilon(1+\delta) \log n\} \mid \boldsymbol{\sigma}_*)\mathbf{1}_\mathcal{H}] \\
&\leq (n^{-(sD_+(a,b) - \frac{\varepsilon(1+\delta)|\log(a/b)|}{2}) + o(1)})^{|S|} \\
&\leq n^{-|S|(sD_+(a,b) - \varepsilon(1+\delta)|\log(a/b)|)},
\end{aligned}
\tag{15}
$$

where the second line holds by Lemma 5 and the last line holds for sufficiently large $n$. We conclude with by Lemma 6, Inequality (14), and Inequality (15) into Inequality (13). $\qquad\square$

**Remark 2.** *For arbitrary $\varepsilon > 0$, we can find $\varepsilon' > 0$ and $\delta > 0$ such that $\varepsilon'(1+\delta) = \varepsilon$. For the sake of convenience, we also denote $D_+(a, b, s, \varepsilon)$ as $sD_+(a, b) - \varepsilon|\log(a/b)|$ and this is equivalent as the $(\varepsilon, \delta)$-parameterization in Lemma 8. We mainly use the $sD_+(a, b, s, \varepsilon)$ notation in the analysis throughout this paper.*

### D.4  Tree node assigning

Before getting to the first moment calculation of the similarity score, we introduce a sub-problem, named *in-community edge counting for node assignment on trees*.

Assume that we have a random graph $G$ with $n$ vertices, which are labeled by a community label vector $\boldsymbol{\sigma}$. We are also given a rooted tree $T(i)$ with $N$ vertices other than the root, where $i$ specifies the root node of $T$ on $G$. Planting $T(i)$ onto $G$ has at least $\binom{n}{N}$ possible positions. If $G$ is a stochastic block model, different planted positions of $T$ would contain different numbers of in-community edges. We are interested in the distribution of the number of in-community edges when planting $T(i)$ onto $G$ uniformly at random.

**Lemma 9.** *Let $K_n$ be the complete graph of a stochastic block model $G$ on $n$ vertices. Let $\boldsymbol{\sigma}_* = \{\boldsymbol{\sigma}_*(i)\}^n_{i=1}, \boldsymbol{\sigma}_*(i) \in \{-1, +1\}$ drawn independently and uniformly at random as the vector of community labels. Let $v \in [n]$ be an arbitrary node of $K_n$. Let $T$ be an arbitrary rooted tree with $N$ vertices other than the root. Consider a uniformly random injective function $\tau : V(T) \to V(K_n)$ such that root $r(T)$ is mapped to $v$ in $V(K_n)$. Define $X$ as the random variable representing the number of in-community edges in the tree $T$ under random $\tau$. Then,*

$$
Bin(N, \frac{1}{2} - 2n^{-\frac{1}{4}}) \preceq X \mid \mathcal{H} \preceq Bin(N, \frac{1}{2} + 2n^{-\frac{1}{4}}).
$$

*Proof.* There are $N$ edges on the tree $T$. For any $(u, v) \in V(K_n) \times V(K_n)$, we denote $X_{(u,v)}$ as the indicator random variable of event $\mathcal{E} = \{u$ and $v$ are from the same community$\}$. Since $\frac{1}{2} - 2n^{-\frac{1}{4}} \leq \mathbb{P}(\mathcal{E} \mid \mathcal{H}) \leq \frac{1}{2} + 2n^{-\frac{1}{4}}$, $X_{(u,v)} \mid \mathcal{H}$ stochastically dominates Bernoulli$(\frac{1}{2} - 2n^{-\frac{1}{4}})$ and is stochastically dominated by Bernoulli$(\frac{1}{2} + 2n^{-\frac{1}{4}})$. Thus, the summation over all edges on tree $T$, $X \mid \mathcal{H} = \sum_{(u,v) \in E(T)} X_{(u,v)} \mid \mathcal{H}$, stochastically dominates Binom$(n, \frac{1}{2} - 2n^{-\frac{1}{4}})$ and is stochastically dominated by Binom$(n, \frac{1}{2} + 2n^{-\frac{1}{4}})$. $\qquad\square$

**Remark 3.** *If $N = o(n^{\frac{1}{4}})$, Lemma 9 implies that with some non-negative integer $N_1 \leq N$, $\mathbb{P}(\{X = N_1\} \cap \mathcal{H}) = (1 + o(1))\binom{N}{N_1}(\frac{1}{2})^N$. Because $\mathbb{P}(\{X = N_1\} \cap \mathcal{H}) = (1 - o(1))\mathbb{P}(X = N_1 \mid \mathcal{H})$, it*

*suffices to show both the upper and lower bound on* $\mathbb{P}(X = N_1 \mid \mathcal{H})$.

$$\mathbb{P}(X = N_1 \mid \mathcal{H}) = \mathbb{P}(X \geq N_1 \mid \mathcal{H}) - \mathbb{P}(X \geq N_1 + 1 \mid \mathcal{H})$$

$$\leq \sum_{t \geq N_1} \binom{N}{t} (\frac{1}{2} + 2n^{-\frac{1}{4}})^t (\frac{1}{2} - 2n^{-\frac{1}{4}})^{N-t} - \sum_{t \geq N_1 + 1} \binom{N}{t} (\frac{1}{2} - 2n^{-\frac{1}{4}})^t (\frac{1}{2} + 2n^{-\frac{1}{4}})^{N-t}$$

$$\leq (1 + o(1)) \binom{N}{N_1} (\frac{1}{2})^N + \sum_{t \geq (N-N_1) \vee (N_1+1)} \binom{N}{t} (\frac{1}{2})^N f(n, N, t),$$

*where* $f(n, N, t) = (1 + 4n^{-\frac{1}{4}})^t (1 - 4n^{-\frac{1}{4}})^{N-t} - (1 - 4n^{-\frac{1}{4}})^t (1 + 4n^{-\frac{1}{4}})^{N-t}$. *The first inequality holds because of the stochastic dominance in Lemma 9. The second inequality holds because* $\binom{N}{N-t} f(n, N, N-t) = \binom{N}{t} f(n, N, t)$ *and thus cancels out every term in the summation indexed from* $t = N_1 + 1$ *to* $t = N - (N_1 + 1)$. *For* $t \geq (N - N_1) \vee (N_1 + 1)$, *we know that* $\binom{N}{t} < \binom{N}{N_1}$. *Also,* $f(n, N, t) \leq (1 + 4n^{-\frac{1}{4}})^N - (1 - 4n^{-\frac{1}{4}})^N < O(n^{-\frac{1}{4}}N) = o(\frac{1}{N})$ *from our assumption on* $N$. *Therefore, summing over* $t \geq (N - N_1) \wedge (N_1 + 1)$, *we have*

$$\mathbb{P}(X = N_1 \mid \mathcal{H}) \leq (1 + o(1)) \binom{N}{N_1} (\frac{1}{2})^N.$$

*From the other direction of stochastic dominance, we have* $\mathbb{P}(X = N_1 \mid \mathcal{H}) = \mathbb{P}(X \geq N_1 \mid \mathcal{H}) - \mathbb{P}(X \geq N_1 + 1 \mid \mathcal{H}) \geq (1 - o(1)) \binom{N}{N_1} (\frac{1}{2})^N$.

**Lemma 10.** *Let* $K_n$ *be the complete graph of a stochastic block model* $G$ *on* $n$ *vertices. Let* $\boldsymbol{\sigma}_* = \{\boldsymbol{\sigma}_*(i)\}_{i=1}^n, \boldsymbol{\sigma}_*(i) \in \{-1, +1\}$ *drawn independently and uniformly at random as the vector of community labels. Let* $v \in [n]$ *be an arbitrary node of* $G$. *Let* $\{T\}_{i=1}^t$ *be a sequence of arbitrary rooted tree with* $N$ *vertices other than the root. Consider a uniformly random injective function* $\tau : V(T) \to V(K_n)$ *such that root* $r(T_1)$ *of the first tree is mapped to* $v$ *in* $V(K_n)$. *Also, the root of* $i$-*th tree is mapped to* $\tau(r(T_i)) = \tau(u)$ *for a fixed vertex on previous trees* $u \in \cup_{t=1}^{i-1} V(T_t)$ *or a fixed vertex on* $K_n$. *Define* $X$ *as the random variable representing the number of in-community edges in all the trees under random* $\tau$. *Then,*

$$Bin(N, \frac{1}{2} - 2n^{-\frac{1}{4}}) \preccurlyeq X \mid \mathcal{H} \preccurlyeq Bin(N, \frac{1}{2} + 2n^{-\frac{1}{4}}).$$

*Proof.* Define $X_i$ as the random variable for number of in-community edges on Tree $T_i$ with random embedding $\tau$. The following holds immediately from Lemma 9,

$$\mathrm{Bin}(N_{T_i}, \frac{1}{2} - 2n^{-\frac{1}{4}}) \preccurlyeq X_i \mid \mathcal{H} \preccurlyeq \mathrm{Bin}(N_{T_i}, \frac{1}{2} + 2n^{-\frac{1}{4}}).$$

Since $X \mid \mathcal{H} = \sum_i X_i \mid \mathcal{H}$,

$$\mathrm{Bin}(N, \frac{1}{2} - 2n^{-\frac{1}{4}}) \preccurlyeq X \mid \mathcal{H} \preccurlyeq \mathrm{Bin}(N, \frac{1}{2} + 2n^{-\frac{1}{4}}). \qquad \square$$

To present the following Lemma 11, we need to introduce a few more definitions: *decorated union graph* and *decorated edges*, which will be explained in more details in the context of chandelier in Section I.

**Decorated union graph.** Let $S_1, S_2, T_1, T_2$ be four rooted graphs. The union graph is defined as $U := S_1 \cup S_2 \cup T_1 \cup T_2$. We define the decorated union graph as a two-tuple $\dot{U} := (U, D_U)$, which is $U$ associating with a decoration set. For each edge,

$$D_U(e) = \begin{cases} \text{The subset of } \{S_1, S_2, T_1, T_2\} \text{ where } e \text{ occurs}, & \text{if } e \in E(U), \\ \emptyset, & \text{otherwise.} \end{cases}$$

We call an edge $e \in E(U)$ is $t$-decorated if $|D_U(e)| = t$ for $t \in \{0, 1, 2, 3, 4\}$. Decorated union graph $\dot{U}$ has one-to-one correspondence with $(S_1, S_2, T_1, T_2)$ as we can uniquely determine $\dot{U}$ given $(S_1, S_2, T_1, T_2)$ and uniquely recover $(S_1, S_2, T_1, T_2)$ given $\dot{U}$.

**Lemma 11** (Asymptotic independence of the counts of $i$-decorated in-community edges)**.** *Let $K_n$ be the complete graph of a stochastic block model $G$ on $n$ vertices. Consider a connected decorated union graph $\dot{U}_P$ with $d_1$ 1-decorated edges, $d_2$ 2-decorated edges, $d_3$ 3-decorated edges, and $d_4$ 4-decorated edges, rooted at $v$ on the complete graph $K_n$. Consider a uniformly random injective function $\tau : V(T) \to V(K_n)$ such that root $r(T)$ is mapped to $v$ in $V(K_n)$. Define $X^{(i)}$ as the random variable representing the number of $i$-decorated in-community edges in $\dot{U}_P$. Assume that $|V(\dot{U}_P)| = O(\log n)$ and $\dot{U}_P$ has excess $k$. If $k = -1$, then,*

$$\mathbb{P}(X^{(1)} = M_1, X^{(2)} = M_2, X^{(3)} = M_3, X^{(4)} = M_4 \mid \mathcal{H})$$
$$= (1 \pm o(1))\mathbb{P}(X^{(1)} = M_1 \mid \mathcal{H})\mathbb{P}(X^{(2)} = M_2 \mid \mathcal{H})\mathbb{P}(X^{(3)} = M_3 \mid \mathcal{H})\mathbb{P}(X^{(4)} = M_4 \mid \mathcal{H})$$
$$= (1 + o(1))\binom{d_1}{M_1}\binom{d_2}{M_2}\binom{d_3}{M_3}\binom{d_4}{M_4}\frac{1}{2^{d_1+d_2+d_3+d_4}}.$$

*Proof.* From the assumption $k = -1$ we have $\dot{U}_P$ is a tree. We can decompose $U_P$ to a sequence of trees as follows: Traverse $U_P$ in BFS order and include each maximal connected component with all edges $i$-decorated as a subtree.

In addition, we break this sequence of tree into four sequences based on the decoration number: $\{T_j^{(1)}\}_{j=1}^c, \{T_j^{(2)}\}_{j=1}^d, \{T_j^{(3)}\}_{j=1}^e, \{T_j^{(4)}\}_{j=1}^f$. In the random mapping, the root of each tree should be mapped to either $v$ or a non-root vertex on the other tree. Let $X_j^{(i)}$ be the random variable for the number of in-community edges for the $j$-th tree of $i$-decoration. $X_j^{(i)} \mid \mathcal{H}$ satisfies the following stochastic dominance:

$$\text{Bin}(|V(T_j^{(i)})|, \frac{1}{2} - 2n^{-\frac{1}{4}}) \preccurlyeq X_j^{(i)} \mid \mathcal{H} \preccurlyeq \text{Bin}(|V(T_j^{(i)})|, \frac{1}{2} + 2n^{-\frac{1}{4}}).$$

Thus, their summation satisfies the following stochastic dominance:

$$\text{Bin}(d_i, \frac{1}{2} - 2n^{-\frac{1}{4}}) \preccurlyeq X^{(i)} \mid \mathcal{H} \preccurlyeq \text{Bin}(d_i, \frac{1}{2} + 2n^{-\frac{1}{4}}), i = 1, 2, 3, 4.$$

As each of the series of trees occupy $O(\log n)$ vertices, every $X^{(i)}$ conditioned on other $X^{(i')}, i' \neq i$ still satisfies the stochastic dominance:

$$\text{Bin}(d_i, \frac{1}{2} - 2n^{-\frac{1}{4}}) \preccurlyeq X^{(i)} | X^{(i')}, \mathcal{H} \preccurlyeq \text{Bin}(d_i, \frac{1}{2} + 2n^{-\frac{1}{4}}).$$

Specifically, following Remark 3, $\mathbb{P}(X^{(4)} = M_4 \mid \mathcal{H}) = (1 + o(1))\binom{d_4}{M_4}\frac{1}{2^{d_4}}$, $\mathbb{P}(X^{(3)} = M_3 | X^{(4)} = M_4, \mathcal{H}) = (1 + o(1))\binom{d_3}{M_3}\frac{1}{2^{d_3}}$, $\mathbb{P}(X^{(2)} = M_2 | X^{(3)} = M_3, X^{(4)} = M_4, \mathcal{H}) = (1 + o(1))\binom{d_2}{M_2}\frac{1}{2^{d_2}}$, and $\mathbb{P}(X^{(1)} = M_1 | X^{(2)} = M_2, X^{(3)} = M_3, X^{(4)} = M_4, \mathcal{H}) = (1 + o(1))\binom{d_1}{M_1}\frac{1}{2^{d_1}}$. Collectively, these form the asymptotic independence of $X^{(i)}$:

$$\mathbb{P}(X^{(1)} = M_1, X^{(2)} = M_2, X^{(3)} = M_3, X^{(4)} = M_4 \mid \mathcal{H})$$
$$= (1 + o(1))\binom{d_1}{M_1}\binom{d_2}{M_2}\binom{d_3}{M_3}\binom{d_4}{M_4}\frac{1}{2^{d_1+d_2+d_3+d_4}}. \quad \square$$

Lemma 11 discusses the case of $k = -1$. We do not expect the same property holds for $k \geq 0$ but we have an auxiliary result as in the following Corollary 1.

**Corollary 1.** *(If $k \geq 0$, then $\dot{U}_P$ **contains cycles**.) By definition, $\dot{U}_P$ can be decomposed into a tree $T_P$ with $v_P$ vertices other than the root and additional $k + 1$ distinct edges $E_{k+1} = \{(u_j, v_j)\}_{i=j}^{k+1}$ fixed. By edges' decorations, $E_{k+1} = E^{(1)} \cup E^{(2)} \cup E^{(3)} \cup E^{(4)}$. Let $X^{(i)}$ still be the random variable representing the number of $i$-decorated in-community edges on $\dot{U}_N$. If $k \geq 0$, let $X^{(i,a)}$ be the counts of $i$-decorated in-community edges on $T_N$ and $X^{(i,b)}$ be the counts of $i$-decorated in-community edges on $\{e_i\}_{i=1}^{k+1}$. From construction, we know $X^{(i)} = X^{(i,a)} + X^{(i,b)}$. Correspondingly, there are $A_i$ $i$-decorated edges on tree, $B_i = |E^{(i)}|$, and $d_i = A_i + B_i$, which are all fixed from $\dot{U}_P$.*

*Then, $\mathbb{P}(X^{(1,a)} = a_1, X^{(2,a)} = a_2, X^{(3,a)} = a_3, X^{(4,a)} = a_4, X^{(2,b)} = b_2, X^{(1,b)} = b_1, X^{(3,b)} = b_3, X^{(4,b)} = b_4 \mid \mathcal{H}) \leq (1 + o(1))\binom{A_1}{a_1}\binom{A_2}{a_2}\binom{A_3}{a_3}\binom{A_4}{a_4}\frac{1}{2^{A_1+A_2+A_3+A_4}}.$*

The proof follows straightforwardly from the proof of Lemma 11.

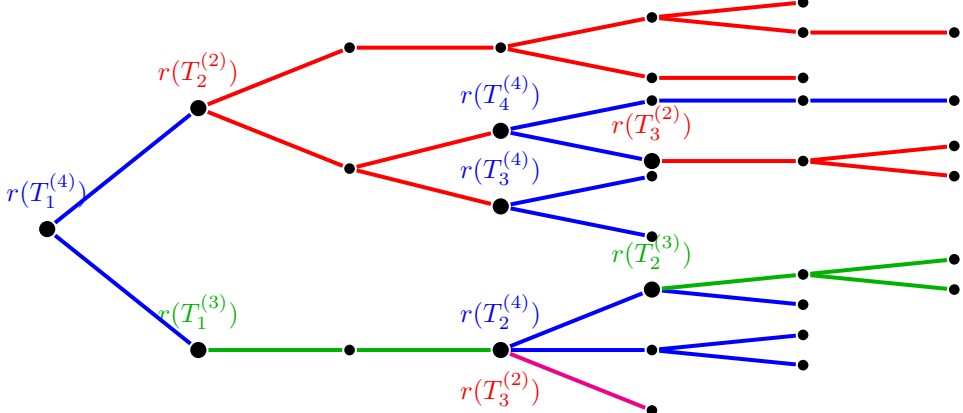

Figure 5: Decomposition of a decorated tree into three sequences of trees. Edges that are 2, 3, 4-decorated are painted as red, green, and blue color correspondingly. Roots of each subtree is marked by larger node and annotated as $r(T_j^{(i)})$, where $i$ is the decoration counts and $j$ is the order in its sequence.

### D.5   Inequalities concerning automorphisms

In this subsection, we study how adding edges can change the number of rooted automorphisms of a rooted tree. With a slight abuse of notation, we denote by $\mathrm{aut}(T)$ the number of rooted automorphisms (i.e., automorphisms which fix the root) of a rooted tree $T$. We start by quantifying the effect of adding an additional edge.

**Lemma 12.** *Let $T = (V, E, r)$ be a rooted tree with root $r$. Let $T' := (V \cup \{u\}, E \cup \{(u, v)\}, r)$ be the rooted tree obtained from $T$ by adding a vertex $u$ and the edge $(u, v)$, where $v \in V$. The following bounds hold:*

$$\mathrm{aut}(T) \times \frac{1}{|V| - 1} \leq \mathrm{aut}(T') \leq \mathrm{aut}(T) \times |V|, \tag{16}$$

*where $|V|$ is the number of vertices in $T$.*

*Proof.* Since $T$ is a rooted tree, there is a natural notion of a *parent* vertex for every vertex other than the root. Namely, the parent of a vertex $v \neq r$ is the neighbor of $v$ which is closest to the root $r$. We also define the *subtree rooted at $v$* to be the subtree of $T$ induced by all vertices whose shortest path to $r$ goes through $v$, rooted at $v$.

We partition $V$ into equivalence classes according to the following rule: $v_1$ and $v_2$ are in the same equivalence class if and only if they share the same parent and the subtrees rooted at $v_1$ and $v_2$ are isomorphic. We denote the resulting partition as $\{V_i\}_{i \in \mathcal{I}}$ and refer to the equivalence classes as orbits. Note that the root $r$ is always in a single-element orbit and thus the size of each orbit is at most $|V| - 1$.

Observe that a permutation of the vertices is a rooted automorphism precisely when it maps each vertex to a vertex in its orbit. Thus we have that

$$\mathrm{aut}(T) = \prod_{i \in \mathcal{I}} |V_i|!. \tag{17}$$

Now consider $T'$, which adds a new vertex $u$ to $T$ with an edge connecting $u$ to a vertex in $T$. By (17), in order to understand $\mathrm{aut}(T')$, we need to understand how the orbits and their sizes change due to the addition of the new vertex and edge. The new vertex $u$ will either join an existing orbit or form its own one. The parent of $u$ may change orbits, so might the parent of its parent, etc. In other words, the vertices on the path from $u$ to the root $r$ might change their orbit, but vertices not on this path will not. In the following, we argue iteratively based on the depth of $u$ in $T'$ (i.e., its distance from the root). When considering the upper bound, we will ignore the possible size decrease of orbits. When considering the lower bound, we will ignore the possible size increase of orbits.

Assume first that the new vertex $u$ is attached to the root $r$. If there are no leaves except for $u$ connecting to $r$, then $u$ forms a new orbit $V_{|\mathcal{I}|+1}$ whose size is 1, which does not change the number of rooted automorphisms. Otherwise, without loss of generality, assume that orbit $V_1$ is the set that contains all leaves connected to $r$. Then $u$ will join this orbit, so the set of orbits of $T'$ is given by $V_1 \cup \{u\}$ and $\{V_i\}_{i=2}^{|\mathcal{I}|}$. Thus, we have that $\mathrm{aut}(T') = \mathrm{aut}(T)(|V_1|+1)$, so in particular

$$\mathrm{aut}(T) \leq \mathrm{aut}(T') = \mathrm{aut}(T)(|V_1|+1) \leq \mathrm{aut}(T) \times |V|.$$

Now suppose that $u$ has depth 2 in $T'$, and let $v^{(1)}$ denote the parent of $u$ in $T'$. Without loss of generality, assume that $v^{(1)} \in V_1$. There are again two cases depending on whether or not there are leaves attached to $v^{(1)}$ in $T$. Suppose first that there are not any leaves attached to $v^{(1)}$ in $T$. Then $u$ has its own orbit (of size 1) in $T'$. The orbit of $v^{(1)}$ changes from $V_1$ in $T$ to either a new orbit or some existing orbit $V_{j_1}$ in $T'$ (for some $2 \leq j_1 \leq |\mathcal{I}|$). In the former case we have that $\mathrm{aut}(T') = \mathrm{aut}(T)/|V_1|$, while in the latter case we have that

$$\mathrm{aut}(T') = \mathrm{aut}(T) \times \frac{|V_{j_1}|+1}{|V_1|}.$$

The desired inequalities thus follow since each orbit has size at most $|V|-1$. Now suppose that there are leaves attached to $v^{(1)}$ in $T$ and let $V_2$ denote the equivalence class of these vertices. Then, $u$ joins the orbit $V_2$ in $T'$. The orbit of $v^{(1)}$ again changes from $V_1$ in $T$ to either a new orbit or some existing orbit $V_{j_1}$ in $T'$ (for some $3 \leq j_1 \leq |\mathcal{I}|$). In the former case we have that $\mathrm{aut}(T') = \mathrm{aut}(T) \times (|V_2|+1)/|V_1|$, while in the latter case we have that

$$\mathrm{aut}(T') = \mathrm{aut}(T) \times \frac{(|V_2|+1)(|V_{j_1}|+1)}{|V_1|}.$$

The lower bound follows since $|V_1| \leq |V|-1$. For the upper bound in the latter case, note that by the definition of orbits, in $T'$ there are $|V_{j_1}|+1$ nodes at depth 1 who each have $|V_2|+1$ children that are leaves. This implies that $T'$ has at least $(|V_2|+1)(|V_{j_1}|+1)$ non-root vertices, so $(|V_2|+1)(|V_{j_1}|+1) \leq (|V|+1)-1 = |V|$.

The general case when $u$ has depth $\ell$ in $T'$ is analogous. Let $v^{(\ell-1)}$ denote the parent of $u$ in $T'$, let $v^{(\ell-2)}$ denote the parent of $v^{(\ell-1)}$, etc. Without loss of generality, let $V_i$ denote the orbit of $v^{(i)}$ in $T$, for $i \in [\ell-1]$. Suppose that $v^{(\ell-1)}$ has children that are leaves in $T$ (the other case, when it does not, is similar and simpler), and let $V_\ell$ denote the equivalence class of these vertices. Then, using similar observations as above, we obtain the following upper and lower bounds on $\mathrm{aut}(T')$:

$$\mathrm{aut}(T) \times \frac{1}{\prod_{i=1}^{\ell-1}|V_i|} \leq \mathrm{aut}(T') \leq \mathrm{aut}(T) \times (|V_\ell|+1) \times \prod_{i=1}^{\ell-1}(|V_{j_i}|+1).$$

Here, for every $i \in [\ell-1]$, either $j_i \in [\ell+1,|\mathcal{I}|]$ (which corresponds to $v^{(i)}$ changing from $V_i$ in $T$ to some existing orbit $V_{j_i}$ in $T'$) or $|V_{j_i}| = 0$ (which corresponds to $v^{(i)}$ changing from $V_i$ in $T$ to a new orbit in $T'$). To conclude the lower bound, observe (using the definition of orbits) that $T$ contains a subtree consisting of the root and $\prod_{i=1}^{\ell-1}|V_i|$ additional vertices, so $\prod_{i=1}^{\ell-1}|V_i| \leq |V|-1$. For the upper bound, observe similarly that $T'$ contains a subtree consisting of the root and $(|V_\ell|+1) \times \prod_{i=1}^{\ell-1}(|V_{j_i}|+1)$ additional vertices, so $(|V_\ell|+1) \times \prod_{i=1}^{\ell-1}(|V_{j_i}|+1) \leq |V|$. □

**Lemma 12 is tight.** Let $T$ be a star rooted at its center (i.e., a tree where all vertices except the root are connected to the root). Then all permutations of the vertices that fix the root are rooted automorphisms, so $\mathrm{aut}(T) = (|V(T)|-1)!$. Now let $T'$ be a tree obtained from $T$ by adding an additional child to the root (see Figure 6, tree in the middle). Then $\mathrm{aut}(T') = (|V(T)|)! = \mathrm{aut}(T) \times |V(T)|$. This shows that the upper bound in Lemma 12 is tight. Now let $T''$ be a tree obtained from $T$ by adding a child to one of the leaves of $T$ (see Figure 6, tree on the right). The rooted automorphisms of $T''$ are precisely the permutations of the vertices that permute the neighbors of the root which are leaves, so $\mathrm{aut}(T'') = (|V(T)|-2)! = \mathrm{aut}(T)/(|V(T)|-1)$. This shows that the lower bound in Lemma 12 is tight.

From Lemma 12, we can derive the following corollary by an iterative argument.

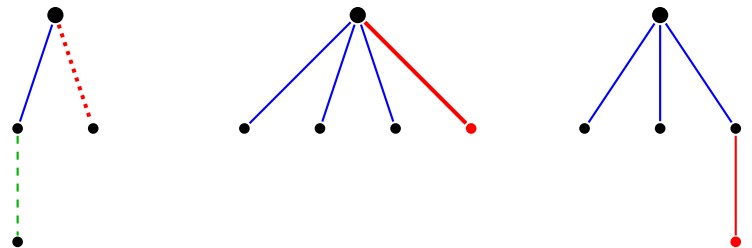

Figure 6: *Left*: The simplest example of two rooted trees $T_1$ and $T_2$ satisfying $\sqrt{\mathrm{aut}(T_1)\mathrm{aut}(T_2)} > \mathrm{aut}(T_1 \cup T_2)$; here $T_1$ is induced by the blue and green edges, $T_2$ is induced by the blue and red edges, and both trees are rooted at the vertex at the top. *Middle*: Example that shows that the upper bound in Lemma 12 is tight, where blue lines and black vertices represent $T$, and the red vertex is additionally added with an edge attaching it to the root. *Right*: Example that shows that the lower bound in Lemma 12 is tight.

**Corollary 2.** *Let $T_1 = (V_1, E_1, r)$ and $T_2 = (V_2, E_2, r)$ be two trees rooted at the same vertex $r$. Define the union $T_1 \cup T_2 := (V_1 \cup V_2, E_1 \cup E_2, r)$ by taking the union of the two vertex sets (both of which contain $r$) and the union of the two edge sets, with multiple edges ignored (i.e., if an edge appears in both $E_1$ and $E_2$, then it appears in $E_1 \cup E_2$ exactly once). Suppose that $T_1 \cup T_2$ is also a tree. Let $d := |E_1 \triangle E_2|$ denote the size of the symmetric difference of the edge sets. Then*

$$\sqrt{\mathrm{aut}(T_1)\mathrm{aut}(T_2)} \leq \mathrm{aut}(T_1 \cup T_2) \times (2\max\{|V_1|, |V_2|\})^d. \tag{18}$$

*Proof.* By the assumptions on $T_1$ and $T_2$, the union $T_1 \cup T_2$ has $(|V_1| + |V_2| + d)/2 - 1$ edges. There are two natural ways that we can think of $T_1 \cup T_2$. First, we can start from $T_1$ and add $(|V_2| - |V_1| + d)/2$ new vertices—those that are in $V_2$ but not in $V_1$—one at a time, together with a new edge for each new added vertex, connecting it to an existing vertex, to obtain $T_1 \cup T_2$. With this perspective, applying the lower bound in (16) from Lemma 12 across each of the $(|V_2| - |V_1| + d)/2$ steps, we obtain that

$$\mathrm{aut}(T_1) \leq \mathrm{aut}(T_1 \cup T_2) \prod_{k=|V_1|-1}^{(|V_1|+|V_2|+d)/2-2} k. \tag{19}$$

On the other hand, we can equally well start from $T_2$ and add $(|V_1| - |V_2| + d)/2$ new edges and vertices to obtain $T_1 \cup T_2$. Thus, analogously, (19) also holds with $\mathrm{aut}(T_1)$ on the left hand side replaced with $\mathrm{aut}(T_2)$, and $|V_1| - 1$ on the right hand side (the minimum value of $k$ in the product) replaced with $|V_2| - 1$.

Combining these two inequalities, we obtain that

$$\sqrt{\mathrm{aut}(T_1)\mathrm{aut}(T_2)} \leq \mathrm{aut}(T_1 \cup T_2) \times \prod_{k=\min\{|V_1|,|V_2|\}-1}^{(|V_1|+|V_2|+d)/2-2} k.$$

Since $d \leq 2(\max\{|V_1, V_2|\} - 1)$, it follows that $(|V_1| + |V_2| + d)/2 - 2 \leq 2\max\{|V_1|, |V_2|\}$, so all factors in the product above are at most $2\max\{|V_1|, |V_2|\}$. Note also that $d \geq \max\{|V_1, V_2|\} - \min\{|V_1, V_2|\}$, so the number of factors in the product above is $(\max\{|V_1, V_2|\} - \min\{|V_1, V_2|\} + d)/2 \leq d$. Putting these observations together we see that (18) holds. $\square$

**Remark 4.** *The leftmost example in Figure 6 provides an example showing that the right hand side of (18) would no longer be an upper bound without the factor $(2\max\{|V_1|, |V_2|\})^d$.*

### D.6  Auxiliary result: Bounds on the cross-moments

In this section, we summarize the upper bound on cross-moments. In Lemma 13, we work on the regime $sD_+(a, b) > 1$ where we can recover the community label exactly first. In Lemma 14, we work on the regime $sD_+(a, b) < 1$ where we we have a inverse polynomial fraction of vertices being labeled incorrectly.

**Lemma 13.** *Let $(G_1, G_2) \sim \mathrm{CSBM}(n, p, q, s), p = \frac{a \log n}{n}, q = \frac{b \log n}{n}$. Denote $\overline{A}, \overline{B}$ as the centralized adjacency matrices of $G_1$ and $G_2$ correspondingly. Let $e := (u, v) \in [n] \times [n]$ be an arbitrary edge. Define the edge type indicator $c(e) = +$ if $e$ is an in-community edge and $c(e) = -$ if $e$ is a cross-community edge under $\boldsymbol{\sigma}_*$. The following holds for $0 \leq \ell, m \leq 2, 2 \leq \ell + m \leq 4$, and any $\boldsymbol{\sigma}_*$,*

$$\beta_{\ell,m}(e) := \sigma_{c(e)}^{-(\ell+m)} \mathbb{E}[\overline{A}_e^\ell \overline{B}_e^m \mid \boldsymbol{\sigma}_*] \leq \begin{cases} 1 & (\ell, m) = (2, 0), (0, 2) \\ (1 + \Theta(\frac{\log n}{n}))\rho & (\ell, m) = (1, 1) \\ \frac{1}{\sqrt{s(p \wedge q)}} & (\ell, m) = (2, 1), (1, 2) \\ \frac{1}{s(p \wedge q)} & (\ell, m) = (2, 2). \end{cases}$$

*Proof.* Conditioning on a specific true community label vector that satisfies the balanced community event $\mathcal{H}$, we can apply Lemma 5 from [42] for each case of $c(e) = +$ and $c(e) = -$. Then, the upper bound are different in terms of $p, q$ and $\rho_+, \rho_-$. When $(\ell, m) = (1, 1)$, $\beta_{\ell,m} \leq \max\{|\rho_+|, |\rho_-|\} \leq (1 + \Theta(\frac{\log n}{n}))\rho$. When $(\ell, m) = (2, 1)$ or $(1, 2)$, if $c(e) = +$, then we have $\beta_{\ell,m} \leq \frac{1}{\sqrt{sp}}$, else we have $\beta_{\ell,m} \leq \frac{1}{\sqrt{sq}}$. Therefore, $\beta_{\ell,m} \leq \frac{1}{\sqrt{s(p \wedge q)}}$. The same argument holds for the case when $(\ell, m) = (2, 2)$. $\square$

**Lemma 14.** *Let $(G_1, G_2) \sim \mathrm{CSBM}(n, p, q, s), p = \frac{a \log n}{n}, q = \frac{b \log n}{n}$. Denote $\overline{A}^{\widehat{\boldsymbol{\sigma}}_A}, \overline{B}^{\widehat{\boldsymbol{\sigma}}_B}$ as the approximately centralized adjacency matrices of $G_1$ and $G_2$ correspondingly. Let $e := (u, v) \in [n] \times [n]$ be an arbitrary edge. Define the edge type indicator $c(e) = +$ if $e$ is an in-community edge and $c(e) = -$ if $e$ is a cross-community edge under $\boldsymbol{\sigma}_*$. Define $\Delta := |sp - sq|$. The following holds for $0 \leq \ell, m \leq 2, 1 \leq \ell + m \leq 4$, and any $\boldsymbol{\sigma}_*, \widehat{\boldsymbol{\sigma}} = (\widehat{\boldsymbol{\sigma}}_A, \widehat{\boldsymbol{\sigma}}_B)$,*

$$\eta_{\ell,m}(e) := \sigma_{c(e)}^{-(\ell+m)} \mathbb{E}[\overline{A}_e^{\widehat{\boldsymbol{\sigma}}_A, \ell} \overline{B}_e^{\widehat{\boldsymbol{\sigma}}_B, m} \mid \boldsymbol{\sigma}_*, \widehat{\boldsymbol{\sigma}}] \leq \begin{cases} \Theta(\sqrt{\Delta}) & (\ell, m) = (0, 1), (1, 0) \\ 1 + \Theta(\frac{\log n}{n}) & (\ell, m) = (2, 0), (0, 2) \\ \rho(1 + \Theta(\frac{\log n}{n})) & (\ell, m) = (1, 1) \\ \frac{1}{\sqrt{s(p \wedge q)}}(1 + \Theta(\frac{\log n}{n})) & (\ell, m) = (2, 1), (1, 2) \\ \frac{1}{s(p \wedge q)}(1 + \Theta(\frac{\log n}{n})) & (\ell, m) = (2, 2). \end{cases}$$

*Proof.* Denote the two vertices on $e$ as $u$ and $v$. We denote $p' := sp$ and $q' := sq$.

**(a) $\ell + m = 1$.** Without loss of generality, we consider $\ell = 1$ and $m = 0$.

$$\mathbb{E}[\overline{A}_e^{\widehat{\boldsymbol{\sigma}}_A, 1} \mid \boldsymbol{\sigma}_*, \widehat{\boldsymbol{\sigma}}] = \begin{cases} 0, & \boldsymbol{\sigma}_*(u)\boldsymbol{\sigma}_*(v) = \widehat{\boldsymbol{\sigma}}_A(u)\widehat{\boldsymbol{\sigma}}_A(v), \\ \pm\Delta, & \boldsymbol{\sigma}_*(u)\boldsymbol{\sigma}_*(v) \neq \widehat{\boldsymbol{\sigma}}_A(u)\widehat{\boldsymbol{\sigma}}_A(v). \end{cases}$$

When $\mathbb{E}[\overline{A}_e^{\widehat{\boldsymbol{\sigma}}_A, 1} \mid \boldsymbol{\sigma}_*, \widehat{\boldsymbol{\sigma}}]$ is non-zero, $\sigma_{c(e)}^{-1} \mathbb{E}[\overline{A}_e^{\widehat{\boldsymbol{\sigma}}_A, 1} \mid \boldsymbol{\sigma}_*, \widehat{\boldsymbol{\sigma}}] = \Theta(\sqrt{\Delta})$.

**(b) $\ell + m = 2$.** We first consider the case of $\ell = 2$ or $m = 2$.

We explicitly calculate the expectation on the following two cases. If $\boldsymbol{\sigma}_*(u)\boldsymbol{\sigma}_*(v) = \widehat{\boldsymbol{\sigma}}(u)\widehat{\boldsymbol{\sigma}}(v)$, then $\mathbb{E}[\overline{A}_e^{\widehat{\boldsymbol{\sigma}}_A, 2} \mid \boldsymbol{\sigma}_*, \widehat{\boldsymbol{\sigma}}] = p'(1 - p') = \sigma_{c(e)}^2$. If $\boldsymbol{\sigma}_*(u)\boldsymbol{\sigma}_*(v) \neq \widehat{\boldsymbol{\sigma}}(u)\widehat{\boldsymbol{\sigma}}(v)$, then $\mathbb{E}[\overline{A}_e^{\widehat{\boldsymbol{\sigma}}_A, 2} \mid \boldsymbol{\sigma}_*, \widehat{\boldsymbol{\sigma}}] = p' - 2p'q' + q'^2 = \sigma_{c(e)}^2 + \Delta^2$.

In summary, we have

$$\mathbb{E}[\overline{A}_e^{\widehat{\boldsymbol{\sigma}}_A, 2} \mid \boldsymbol{\sigma}_*, \widehat{\boldsymbol{\sigma}}] = \mathbb{E}[\overline{B}_e^{\widehat{\boldsymbol{\sigma}}_B, 2} \mid \boldsymbol{\sigma}_*, \widehat{\boldsymbol{\sigma}}] \leq \sigma_{c(e)}^2 + \Delta^2 = \sigma_{c(e)}^2(1 + \Theta(\frac{\log n}{n})).$$

We next consider the case of $\ell = m = 1$. If $\boldsymbol{\sigma}_*(u)\boldsymbol{\sigma}_*(v) = \widehat{\boldsymbol{\sigma}}(u)\widehat{\boldsymbol{\sigma}}(v)$, then $\mathbb{E}[\overline{A}_e^{\widehat{\boldsymbol{\sigma}}_A, 1} \overline{B}_e^{\widehat{\boldsymbol{\sigma}}_B, 1} \mid \boldsymbol{\sigma}_*, \widehat{\boldsymbol{\sigma}}] = \rho_{c(e)} \sigma_{c(e)}^2$. If $\boldsymbol{\sigma}_*(u)\boldsymbol{\sigma}_*(v) \neq \widehat{\boldsymbol{\sigma}}(u)\widehat{\boldsymbol{\sigma}}(v)$, then $\mathbb{E}[\overline{A}_e^{\widehat{\boldsymbol{\sigma}}_A, 1} \overline{B}_e^{\widehat{\boldsymbol{\sigma}}_B, 1} \mid \boldsymbol{\sigma}_*, \widehat{\boldsymbol{\sigma}}] = \rho_{c(e)} \sigma_{c(e)}^2 + \Delta^2$.

In summary, we have

$$\mathbb{E}[\overline{A}_e^{\widehat{\boldsymbol{\sigma}}_A, 1} \overline{B}_e^{\widehat{\boldsymbol{\sigma}}_B, 1} \mid \boldsymbol{\sigma}_*, \widehat{\boldsymbol{\sigma}}] \leq \rho_{c(e)} \sigma_{c(e)}^2 + \Delta^2 = \rho \sigma_{c(e)}^2(1 + \Theta(\frac{\log n}{n}))$$

**(c) $\ell + m = 3$.** With loss of generality, we assume that this edge connects two vertices from different communities. Conditioned on correct centralization, we can compute

$$\mathbb{E}[\overline{A}_e^{\widehat{\sigma}_A,2} \overline{B}_e^{\widehat{\sigma}_B,1} \mid \boldsymbol{\sigma}_*, \widehat{\boldsymbol{\sigma}}] = \mathbb{E}[\overline{A}_e^2 \overline{B}_e \mid \boldsymbol{\sigma}_*] = qs^2(1-q)(1-2qs) = \sigma_-^3 \frac{\rho_-(1-2q')}{\sqrt{q'(1-q')}} \le \sigma_-^3 \frac{1}{\sqrt{q'}},$$

which is the same as in Lemma 13. If the centralization is incorrect for both graphs, the expectation changes to the following:

$$\mathbb{E}[\overline{A}_e^{\widehat{\sigma}_A,2} \overline{B}_e^{\widehat{\sigma}_B,1} \mid \boldsymbol{\sigma}_*, \widehat{\boldsymbol{\sigma}}] = q's - p'q' - 2p'q's + 3p'^2q' - p'^3.$$

From observation, we see that the dominant part in both cases are the same, which is $q's = \Theta(\frac{\log n}{n})$. Then, we can show that the difference between two quantities are minor:

$$\frac{\mathbb{E}[\overline{A}_e^{\widehat{\sigma}_A,2} \overline{B}_e^{\widehat{\sigma}_B,1} \mid \boldsymbol{\sigma}_*, \widehat{\boldsymbol{\sigma}}] - \mathbb{E}[\overline{A}_e^2 \overline{B}_e \mid \boldsymbol{\sigma}_*]}{\sigma_-^3} = \frac{(p'-q')(p'^2 - 2p'q' - 2q'^2 + q' + 2q's)}{q'(1-q')\sqrt{q'(1-q')}} = \Theta(\sqrt{q'}). \tag{20}$$

The above (20) also holds for the case when $e$ is classified correctly in $A$ or $B$ only.

By also considering the other case, that is, this edge is connects two vertices from the same community but wrongly centralized according to $\widehat{\boldsymbol{\sigma}}$, we conclude with

$$\mathbb{E}[\overline{A}_e^{\widehat{\sigma}_A,2} \overline{B}_e^{\widehat{\sigma}_B,1} \mid \boldsymbol{\sigma}_*, \widehat{\boldsymbol{\sigma}}] \le \sigma_{c(e)}^3 \frac{1}{\sqrt{s(p \wedge q)}}(1 + \Theta(s(p \wedge q))) = \sigma_{c(e)}^3 \frac{1}{\sqrt{s(p \wedge q)}}(1 + \Theta(\frac{\log n}{n}))$$

**(d) $\ell + m = 4$.** With loss of generality, we assume that this edge connects two vertices from different communities. For the correct centralization, the moment stays the same as in Lemma 13,

$$\mathbb{E}[\overline{A}_e^2 \overline{B}_e^2 \mid \boldsymbol{\sigma}_*] = q's - 4q'^2s + 4q'^3s + 2q'^3 - 3q'^4 = q'^2(1-q')^2 + q'(1-q')\rho_-(1-2q')^2 \le \sigma_-^4 \frac{1}{q'}.$$

If the centralization is incorrect for both graphs, we have

$$\mathbb{E}[\overline{A}_e^{\widehat{\sigma}_A,2} \overline{B}_e^{\widehat{\sigma}_B,2} \mid \boldsymbol{\sigma}_*, \widehat{\boldsymbol{\sigma}}] = q's - 4p'q's + 4p'^2q's + 2p'^2q' - 4p'^3q' + p'^4,$$

and we can show that the error has the following order

$$\mathbb{E}[\overline{A}_e^{\widehat{\sigma}_A,2} \overline{B}_e^{\widehat{\sigma}_B,2} \mid \boldsymbol{\sigma}_*, \widehat{\boldsymbol{\sigma}}] - \mathbb{E}[\overline{A}_e^{\widehat{\sigma}_A,2} \overline{B}_e^{\widehat{\sigma}_B,2} \mid \boldsymbol{\sigma}_*] \le \Theta((\frac{\log n}{n})^2) = \Theta(\sigma_{c(e)}^4). \tag{21}$$

When the centralization is incorrect for only one graph, (21) still holds.

By also considering the other case, that is, this edge is connects two vertices from the same community but wrongly centralized according to $\widehat{\boldsymbol{\sigma}}$, we conclude with

$$\mathbb{E}[\overline{A}_e^{\widehat{\sigma}_A,2} \overline{B}_e^{\widehat{\sigma}_B,2} \mid \boldsymbol{\sigma}_*, \widehat{\boldsymbol{\sigma}}] \le \Theta(\sigma_{c(e)}^4) + \sigma_{c(e)}^4 \frac{1}{s(p \wedge q)} \le \sigma_{c(e)}^4 \frac{1}{s(p \wedge q)}(1 + \Theta(\frac{\log n}{n})). \qquad \square$$

## E   Proof of Theorem 3: Almost exact graph matching

Fix constants $a \neq b > 0, s \in [0,1]$. Throughout the paper, we refer to $sD_+(a,b) > 1$ as Regime I and $sD_+(a,b) < 1$ as Regime II. We define $\mu := |\mathcal{T}| n^N \rho^N \sigma_{\text{eff}}^{2N}$, where $\sigma_{\text{eff}}^2 = \frac{\sigma_+^2 + \sigma_-^2}{2}$. Depending on the different parameter regimes, the analysis is different. We first present the first and second moment bounds for both regimes and then prove Theorem 3.

### E.1   Regime I: Exact community recovery is possible for a single graph

**Proposition 1** (Mean calculation, $sD_+(a,b) \ge 1$). *Given $(G_1, G_2) \sim \text{CSBM}(n, \frac{a \log n}{n}, \frac{b \log n}{n}, s)$, the similarity score satisfies,*

$$\mathbb{E}[\Phi_{ij} \mathbf{1}_{\mathcal{H}}] = \begin{cases} (1 + o(1))\mu, & \text{if } \pi_*(i) = j, \\ 0, & \text{if } \pi_*(i) \neq j. \end{cases}$$

**Proposition 2** (Variance calculation–True pairs, $sD_+(a,b) > 1$). *Suppose that $j = \pi_*(i)$, that $sD_+(a,b) > 1$, and that*

$$\frac{14L^2}{\rho^{2(K+M)}(|\mathcal{J}|)} \leq \frac{1}{2}, \quad \frac{22R^4(2N+1)(11\beta)^{2(K+M)}}{n} \leq \frac{1}{2},$$

$$\frac{4R^{\frac{4}{M}}(11\beta)^{\frac{4K+4M}{M}}}{ns(p \wedge q)} \leq \frac{1}{2}, \quad \frac{1+2L^2}{ns(p \wedge q)} \leq \frac{1}{4}. \tag{22}$$

*Then, for any $i \in [n]$, we have*

$$\frac{\mathrm{Var}[\Phi_{ij}\mathbf{1}_{\mathcal{H}}]}{\mathbb{E}[\Phi_{i\pi_*(i)}\mathbf{1}_{\mathcal{H}}]^2} = O\left(\frac{L^2}{\rho^2 ns(p \wedge q)} + \frac{L^2}{\rho^{2(K+M)}|\mathcal{J}|}\right).$$

**Proposition 3** (Variance calculation–Fake pairs, $sD_+(a,b) > 1$). *Suppose that $j \neq \pi_*(i)$, that $sD_+(a,b) > 1$, and that*

$$4^{L+4}L^{2L\wedge(4K+2)}(11\beta)^{8(K+M)}R^4(2N+1)^3 \leq \frac{n}{2}, \quad \frac{4R^{\frac{2}{M}}(11\beta)^{\frac{4(K+M)}{M}}}{ns(p \wedge q)} \leq \frac{1}{2}. \tag{23}$$

*Then, for any $i \in [n]$, we have*

$$\frac{\mathrm{Var}[\Phi_{ij}\mathbf{1}_{\mathcal{H}}]}{\mathbb{E}[\Phi_{i\pi_*(i)}\mathbf{1}_{\mathcal{H}}]^2} = O(\frac{1}{|\mathcal{T}|\rho^{2N}}).$$

### E.2 Regime II: Exact community recovery is impossible for a single graph

Before getting into the details of proof, let us first give more intuition on the approximately centralized adjacency matrix.

In this regime, for each element in the adjacency matrix, there are four cases: (1) Correct centralization for in-community edges; (2) Incorrect centralization for in-community edges; (3) Correct centralization for cross-community edges; (4) Incorrect centralization for cross-community edges. Figure 7 gives an example of how the graphon for balanced 2-community SBM changes under community label misclassifications.

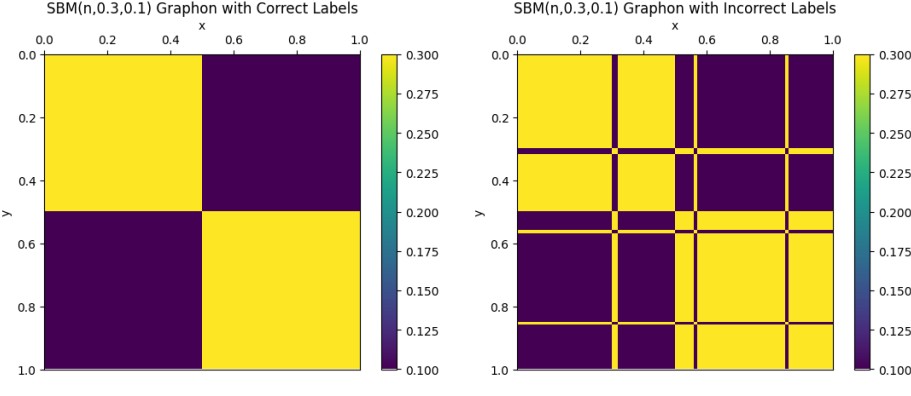

(a) Regime I: exact recovery is possible.  (b) Regime II: exact recovery is impossible.

Figure 7: Illustration on the impact of incorrect community labels on adjacency matrix centralization with $G \sim \mathrm{SBM}(n, 0.3, 0.1)$. Centralized adjacency matrix $\overline{A}^{\widehat{\sigma}_A} = A - \mathbb{E}[A]$, where matrix $\mathbb{E}[A]$ takes the value of discretized graphon as above.

**Proposition 4** (Mean calculation, $sD_+(a,b) < 1$). *Let $(G_1, G_2) \sim \mathrm{CSBM}(n, \frac{a\log n}{n}, \frac{b\log n}{n}, s)$. Given $(K, L, M, R, D)$-Chandelier class $\mathcal{T}$. Assume that $D = o(\frac{\log n}{\log\log n})$.*

*For all $i \in [n]$ with $j = \pi_*(i)$, we have*

$$\mathbb{E}[\Phi_{i\pi_*(i)}^{\widehat{\sigma}}\mathbf{1}_{\mathcal{H}}] = (1+o(1))\mu.$$

*For all $i \in [n]$ with $j \neq \pi_*(i)$, we have*

$$\mathbb{E}[\Phi_{ij}^{\widehat{\sigma}} \mathbf{1}_{\mathcal{H}}] = o(\mu).$$

**Proposition 5** (Variance calculation–True pairs, $sD_+(a,b) < 1$). *Suppose that $j = \pi_*(i)$, $sD_+(a,b) < 1$, $L = o(n)$, and that for some $c > 0$,*

$$\frac{4R^{\frac{2}{M}}(15\beta)^{2\frac{K+M}{M}}}{ns(p \wedge q)} \leq \frac{1}{2}, \quad \frac{30R^4(2N+1)^2(15\beta)^{4(K+M)}}{n} \leq \frac{1}{2},$$

$$\frac{sD_+(a,b)}{D} \geq \frac{(\log \log n)^2}{\log n}, \quad 2NL(4LM)^{6L} \leq \log^c n. \tag{24}$$

*Then, for any $i \in [n]$, we have*

$$\frac{\mathrm{Var}[\Phi_{ij}^{\widehat{\sigma}} \mathbf{1}_{\mathcal{H}}]}{\mathbb{E}[\Phi_{i\pi_*(i)}^{\widehat{\sigma}} \mathbf{1}_{\mathcal{H}}]^2} = O\left(\frac{L^2}{\rho^2 ns(p \wedge q)} + \frac{L^2}{\rho^{2(K+M)}|\mathcal{J}|}\right).$$

**Proposition 6** (Variance calculation–Fake pairs, $sD_+(a,b) < 1$). *Suppose that $j \neq \pi_*(i)$, $sD_+(a,b) < 1$, $N = \Theta(\log n)$, $D = o(\frac{\log n}{\log \log n})$, and that*

$$\frac{4R^{\frac{2}{M}}(15\beta)^{2\frac{K+M}{M}}}{ns(p \wedge q)} \leq \frac{1}{2}, \quad \frac{sD_+(a,b)}{D} \geq \frac{(\log \log n)^2}{\log n},$$

$$\left(\frac{(15\beta)^{2(K+M)}30R^4(4n+1)^2}{n}\right)(2\beta)^{8(K+M)}(4LM)^{4L}(4L)! \leq \frac{1}{2}, \tag{25}$$

*Then, for any $i \in [n]$, we have*

$$\frac{\mathrm{Var}[\Phi_{ij}^{\widehat{\sigma}} \mathbf{1}_{\mathcal{H}}]}{\mathbb{E}[\Phi_{i\pi_*(i)}^{\widehat{\sigma}} \mathbf{1}_{\mathcal{H}}]} = O(\frac{1}{|\mathcal{T}|\rho^{2N}}).$$

### E.3 Putting things together: Proof of Theorem 3

*Proof of Proposition 3.* Firstly, the following conditions imply (22), (23), (24), and (25):

$$L \leq \frac{c_1 \log n}{\log \log n} \wedge c_6 \sqrt{ns(p \wedge q)}, \quad \frac{c_2}{\log(ns(p \wedge q))} \leq \frac{M}{K} \leq \frac{\log \frac{\rho^2}{\alpha}}{2 \log \frac{1}{\rho^2}}, \quad KL \geq \frac{c_3 \log n}{\log \frac{\rho^2}{\alpha}},$$

$$K + M \leq c_4 \log n, \quad R = \exp(c_5 K), \quad D \leq c_7 \frac{\log n}{(\log \log n)^2}. \tag{26}$$

for some absolute constants $c_1, c_2, \ldots, c_7 > 0$.

Furthermore, with growing $ns(p \wedge q)$, we have for $j = \pi_*(i)$, $\frac{\mathrm{Var}[\Phi_{ij}\mathbf{1}_{\mathcal{H}}]}{\mathbb{E}[\Phi_{i\pi_*(i)}\mathbf{1}_{\mathcal{H}}]^2} = o(1)$ when $sD_+(a,b) > 1$ and $\frac{\mathrm{Var}[\Phi_{ij}\mathbf{1}_{\mathcal{H}}]}{\mathbb{E}[\Phi_{i\pi_*(i)}\mathbf{1}_{\mathcal{H}}]^2} = o(1)$ when $sD_+(a,b) < 1$. For any $\varepsilon > 0$ there exists $\varepsilon' > 0$ such that $s^2 \geq \alpha + \varepsilon \Leftrightarrow \rho^2 \geq \alpha + \varepsilon'$. We have for $j \neq \pi_*(i)$, $\frac{\mathrm{Var}[\Phi_{ij}\mathbf{1}_{\mathcal{H}}]}{\mathbb{E}[\Phi_{i\pi_*(i)}\mathbf{1}_{\mathcal{H}}]^2} = o(\frac{1}{n^2})$ when $sD_+(a,b) > 1$ and $\frac{\mathrm{Var}[\Phi_{ij}\mathbf{1}_{\mathcal{H}}]}{\mathbb{E}[\Phi_{i\pi_*(i)}\mathbf{1}_{\mathcal{H}}]^2} = o(\frac{1}{n^2})$ when $sD_+(a,b) < 1$.

**Next, we claim that almost exact recovery is achievable by counting chandeliers.** Let $\tau = c\mu$, for arbitrary $c \in (0,1)$, where $\mu = \mathbb{E}[\Phi_{i\pi_*(i)}\mathbf{1}_{\mathcal{H}}]$. By Chebyshev's inequality, the probability that the similarity score of a fake pair ($j \neq \pi_*(i)$) of vertices exceeding $\tau$, is upper bounded as

$$\mathbb{P}(\Phi_{ij}\mathbf{1}_{\mathcal{H}} \geq \tau) \leq \mathbb{P}\left(|\Phi_{ij}\mathbf{1}_{\mathcal{H}} - \mathbb{E}[\Phi_{ij}\mathbf{1}_{\mathcal{H}}]| \geq c\mathbb{E}[\Phi_{ij}\mathbf{1}_{\mathcal{H}}]\right) \leq \frac{\mathrm{Var}[\Phi_{ij}\mathbf{1}_{\mathcal{H}}]}{c^2\mathbb{E}[\Phi_{ii}\mathbf{1}_{\mathcal{H}}]^2}$$

$$= O\left(\frac{1}{|\mathcal{T}|\rho^{2N}}\right) = o(\frac{1}{n^2}).$$

Next, by the definition of $\mathcal{H}$, we have

$$\mathbb{P}(\Phi_{ij} \geq \tau) \leq \mathbb{P}(\Phi_{ij}\mathbf{1}_{\mathcal{H}} \leq c) + \mathbb{P}(\mathcal{H}^c) = o(\frac{1}{n^2}).$$

Applying union bound over all $i \neq j \in [n]$, we have $\mathbb{P}\{\exists i \neq j, \Phi_{ij} \geq \tau\} = o(1)$. Thus, for all possible pairs of vertices $(i, j) \in [n] \times [n]$, $\Phi_{ij} < \tau$ with high probability.

We next study the probability that the similarity score of a true pair of vertices falling below the threshold $\tau$. We even consider a larger set containing $F := \{i \in [n] : \Phi_{i\pi_*(i)} < \tau\}$,

$$
\begin{aligned}
\mathbb{P}(|\Phi_{i\pi_*(i)}\mathbf{1}_{\mathcal{H}} - \mu| > (1-c)\mu) &\leq \frac{\mathrm{Var}[\Phi_{i\pi_*(i)}\mathbf{1}_{\mathcal{H}}]}{(1-c)^2\mathbb{E}[\Phi_{i\pi_*(i)}\mathbf{1}_{\mathcal{H}}]^2} \\
&= O\left(\frac{L^2}{\rho^2 ns(p \wedge q)} + \frac{L^2}{\rho^{2(K+M)}|\mathcal{J}|}\right) =: \gamma = o(1).
\end{aligned}
$$

Therefore, $\mathbb{E}[|F|] \leq n\gamma$. Denote $I \in [n]$ the vertex set this algorithm matches. For every vertex not in the set $F$, it is guaranteed to be matched up correctly, so

$$
\mathbb{E}[|I|] \geq n - \mathbb{E}[|F|] = n(1 - \gamma).
$$

By Markov's inequality,

$$
\mathbb{P}(|F| >= \sqrt{\gamma}n) \leq \frac{\gamma n}{\sqrt{\gamma}n} = \sqrt{\gamma} = o(1).
$$

Therefore, Algorithm 1 matches $|I| \geq (1 - \sqrt{\gamma})n$ vertices correctly with high probability.

In Regime II, with Proposition 4, Proposition 5, and Proposition 6, the same argument follows. These two regimes together complete the proof. $\qquad\square$

## F Proof of Theorem 4: Efficient algorithm for almost exact graph matching

### F.1 Color-coding-based similarity score estimation

In this section, we re-state and modify the efficient graph matching algorithm for correlated Erdős–Rényi graphs discussed in Section 5 of [41], and then analyze it for correlated SBMs.

Let $(G_1, G_2)$ be a pair of correlated SBMs with adjacency matrices $A$ and $B$. Let $H$ be a rooted connected graph with $N + 1$ vertices. We now want to approximately count the signed subgraphs rooted at $i \in [n]$ on the centralized adjacency matrices. In general, we do not have access to the centralized adjacency matrices $\overline{A}$ and $\overline{B}$, we would use community label estimates $\widehat{\boldsymbol{\sigma}}_A$ and $\widehat{\boldsymbol{\sigma}}_B$ from Algorithm 3 to approximately centralize the adjacency matrices as $\overline{A}^{\widehat{\boldsymbol{\sigma}}_A}$ and $\overline{B}^{\widehat{\boldsymbol{\sigma}}_B}$.

First, we generate a random coloring $\mu : [n] \to [N + 1]$, which is assigning every node on $\overline{A}$ to one of $N + 1$—the same as the number of vertices on a chandelier—colors independently and uniformly at random. For any vertex set $V \subset [n]$, we define $\chi_\mu(V) = \mathbf{1}_{\{\forall x,y \in V, x \neq y : \mu(x) \neq \mu(y)\}}$. We call the vertex set $V$ being colorful if $\chi_\mu(V) = 1$. We denote $r := \mathbb{P}(\chi_\mu(V) = 1) = \frac{(N+1)!}{(N+1)^{N+1}}$. We define the **approximate signed rooted subgraph count** as

$$
X_{i,H}(\overline{A}^{\widehat{\boldsymbol{\sigma}}_A}, \mu) := \sum_{S(i) \cong H} \chi_\mu(V(S))\Pi_{e \in E(S)}\overline{A}_e^{\widehat{\boldsymbol{\sigma}}_A}. \tag{27}
$$

Observe that $\mathbb{E}[X_{i,H}(\overline{A}^{\widehat{\boldsymbol{\sigma}}_A}, \mu)] = rW_{i,H}(\overline{A}^{\widehat{\boldsymbol{\sigma}}_A})$, where $W_{i,H}(\overline{A}^{\widehat{\boldsymbol{\sigma}}_A})$ is the ground truth of signed counts as defined in Section C. In another word, $X_{i,H}(\overline{A}^{\widehat{\boldsymbol{\sigma}}_A}, \mu)/r$ is an unbiased estimator of $W_{i,H}(\overline{A}^{\widehat{\boldsymbol{\sigma}}_A})$.

Let $t := \lceil 1/r \rceil$, we repeat the random coloring for $t$ times and then average over the estimates before taking the inner product of two signed counts vectors. Formally, we generate $2t$ independent colorings independently and uniformly at random, denoted as $\{\mu_a\}_{a=1}^t$ and $\{\nu_b\}_{b=1}^t$. For vertex $i$ in $G_1$ and vertex $j$ in $G_2$, we define the **approximate similarity score** as follows

$$
\widetilde{\Phi}_{ij}^{\boldsymbol{\sigma}} := \frac{1}{r^2} \sum_{H \in \mathcal{T}} \mathrm{aut}(H) \left(\frac{1}{t}\sum_{a=1}^t X_{i,H}(\overline{A}^{\widehat{\boldsymbol{\sigma}}_A}, \mu_a)\right) \left(\frac{1}{t}\sum_{b=1}^t X_{j,H}(\overline{B}^{\widehat{\boldsymbol{\sigma}}_B}, \nu_b)\right).
$$

We have $\mathbb{E}[\widetilde{\Phi}_{ij}^{\widehat{\sigma}}|\overline{A}^{\widehat{\sigma}_A}, \overline{B}^{\widehat{\sigma}_B}] = \Phi_{ij}^{\widehat{\sigma}}$ and also $\mathbb{E}[\widetilde{\Phi}_{ij}^{\widehat{\sigma}}\mathbf{1}_{\mathcal{H}}|\overline{A}^{\widehat{\sigma}_A}, \overline{B}^{\widehat{\sigma}_B}, \boldsymbol{\sigma}_*] = \Phi_{ij}^{\widehat{\sigma}}\mathbf{1}_{\mathcal{H}}$ also holds as $\mathbf{1}_{\mathcal{H}}$ is a deterministic function on $\boldsymbol{\sigma}_*$. We summarize the algorithm as Algorithm 4. There are two additional steps than Algorithm 2 in [41]: First, in the construction of the chandelier class in this work, we need to filter out instances with maximum degree greater than a threshold $D$, which takes $O(K)$ time for each tree; Second, we need to obtain the community label estimates before estimating the signed subgraph counts.

---

**Algorithm 4** Efficient Almost Exact Graph Matching Algorithm

---

**Input:** Adjacency matrices $A$ and $B$ on $n$ vertices for correlated stochastic block models $(G_1, G_2)$.

**Step 1 - Construct the chandelier class**

1: (Rooted tree generation [6]) List all non-isomorphic rooted trees with $K$ edges.
2: (Automorphism constraint [12]) Compute $\mathrm{aut}(H)$ for each rooted tree using the automorphism algorithm for trees.
3: (Maximum degree constraint) Compute the maximum degree of vertices.
4: (Chandelier class) Return $\mathcal{J}$ as the subset of rooted trees whose number of automorphisms is at most $R$ and maximum degree is at most $D$. Construct $(K, L, M, R, D)$-Chandelier class $\mathcal{T}$.

**Step 2 - Estimation of the similarity score**

(Random Coloring) Generate i.i.d. uniformly random colorings $\{\mu_a\}_{a=1}^t$ and $\{\nu_b\}_{b=1}^t$, each maps from $[n]$ to $[N+1]$.
(Community recovery) Obtaining $\widehat{\boldsymbol{\sigma}}_A$ and $\widehat{\boldsymbol{\sigma}}_B$ for $A$ and $B$ independently by Algorithm 3.
**for** all $(i,j) \in [n] \times [n]$ **do**
    **for** all $H \in \mathcal{T}$ **do**
        (Signed counts estimation [42]) Compute $\{X_{i,H}(\overline{A}^{\widehat{\sigma}_A}, \mu_a)\}_{a=1}^t$ and $\{X_{j,H}(\overline{B}^{\widehat{\sigma}_B}, \nu_b)\}_{b=1}^t$.
    **end for**
**end for**

**Output:** The approximate similarity scores $\{\widetilde{\Phi}_{ij}\}_{i,j\in[n]}$.

---

When we can obtain the correct centralized adjacency matrices $\overline{A}$ and $\overline{B}$, we replace $\overline{A}^{\widehat{\sigma}_A}$ (resp. $\overline{B}^{\widehat{\sigma}_B}$) with $\overline{A}$ (resp. $\overline{B}$), and everything defined above is still valid. In that case, we denote the approximate similarity score as $\widetilde{\Phi}$ rather than $\widetilde{\Phi}^{\widehat{\sigma}}$.

### F.2 Analysis

The analysis is similar to that in Section 5.1 of [41], while we split the analysis into two regimes.

Under Regime I ($sD_+(a,b) > 1$), we can recover the community labels on $G_1$ and $G_2$ exactly correct with high probability. We define $\Gamma_{ij}$ as an upper bound of $\mathrm{Var}[\Phi_{ij}]$ as follows:

$$\mathrm{Var}[\Phi_{ij}\mathbf{1}_{\mathcal{H}}] \leq \sum_{H,I \in \mathcal{T}} \mathrm{aut}(H)\mathrm{aut}(I) \sum_{S_1(i),S_2(j)\cong H} \sum_{T_1(i),T_2(j)\cong I} |\mathbb{E}[\overline{A}_{S_1}\overline{B}_{S_2}\overline{A}_{T_1}\overline{B}_{T_2}\mathbf{1}_{\mathcal{H}}]|$$
$$\mathbf{1}_{\{S_1 \neq S_2 \text{ or } T_1 \neq T_2 \text{ or } V(S_1)\cap V(T_1)\neq\{i\}\}}\mathbf{1}_{\{S_1\triangle T_1 \subset S_2\cup T_2, S_2\triangle T_2 \subset S_1\cup T_1\}} =: \Gamma_{ij}. \quad (28)$$

If $\mathbf{1}_{\{S_1\triangle T_1 \subset S_2\cup T_2, S_2\triangle T_2 \subset S_1\cup T_1\}} = 0$, then there exists some edge occurring only once among $S_1$, $S_2$, $T_1$, and $T_2$, and so $|\mathbb{E}[\overline{A}_{S_1}\overline{B}_{S_2}\overline{A}_{T_1}\overline{B}_{T_2}\mathbf{1}_{\mathcal{H}}]| = 0$. Therefore, we only look at the cases where every edge occurs at least two times among these four chandeliers. If $\mathbf{1}_{\{S_1 \neq S_2 \text{ or } T_1 \neq T_2 \text{ or } V(S_1)\cap V(T_1)\neq\{i\}\}} = 0$, then $S_1 = S_2$, $T_1 = T_2$, and $S_1$ has no common vertex with $T_1$ except for the root. In this case, $S_1$ and $T_1$, and $S_2$ and $T_2$ have no common edges, so the covariance between $\overline{A}_{S_1}\overline{B}_{T_1}$ and $\overline{A}_{S_2}\overline{B}_{T_2}$ is always zero.

For Regime II ($sD_+(a,b) < 1$), where we cannot recover the correct centralized adjacency matrices with high probability. Alternatively, we define $\Gamma'_{ij}$ as:

$$\Gamma'_{ij} := \sum_{H,I \in \mathcal{T}} \mathrm{aut}(H)\mathrm{aut}(I) \sum_{S_1,S_2\cong H, T_1,T_2\cong I} \mathbb{E}[\overline{A}_{S_1}^{\widehat{\sigma}_A}\overline{B}_{S_2}^{\widehat{\sigma}_B}\overline{A}_{T_1}^{\widehat{\sigma}_A}\overline{B}_{T_2}^{\widehat{\sigma}_B}\mathbf{1}_{\mathcal{H}}]. \quad (29)$$

From the definition of variance,

$$\mathrm{Var}[\Phi_{ij}^{\widehat{\sigma}}\mathbf{1}_{\mathcal{H}}] \leq \Gamma'_{ij}.$$

**Lemma 15.** *Fix constants $a \neq b > 0, s \in [0,1]$. Let $(G_1, G_2) \sim \text{CSBM}(n, \frac{a \log n}{n}, \frac{b \log n}{n}, s)$. The variance of estimation error has the following upper bound:*

$$\text{Var}[(\widetilde{\Phi}_{ij} - \Phi_{ij})\mathbf{1}_\mathcal{H}] \leq 3\Gamma_{ij} \qquad \text{and} \qquad \text{Var}[(\widetilde{\Phi}_{ij}^{\widehat{\sigma}} - \Phi_{ij}^{\widehat{\sigma}})\mathbf{1}_\mathcal{H}] \leq 3\Gamma'_{ij},$$

*where $\Gamma_{ij}$ is defined in (28) and $\Gamma'_{ij}$ is defined in (29).*

*Proof.* **Regime I:** $sD_+(a,b) > 1$. Define

$$Y_{ij}(\mu, \nu) := \sum_{H \in \mathcal{T}} \text{aut}(H) X_{i,H}(\overline{A}, \mu) X_{j,H}(\overline{B}, \nu),$$

where $\mu, \nu$ are two $(N+1)$-coloring of the vertices in $[n]$. Then, we can represent the approximate similarity score as

$$\widetilde{\Phi}_{ij} = \frac{1}{r^2 t^2} \sum_{c=1}^t \sum_{d=1}^t Y_{ij}(\mu_c, \nu_d). \tag{30}$$

For any $\mu_c, \nu_d, 1 \leq c, d \leq t$, $Y_{ij}(\mu_c, \nu_d)/r^2$ is an unbiased estimator of $\Phi_{ij}$ given $\overline{A}, \overline{B}$ as

$$\mathbb{E}[Y_{ij}(\mu_c, \nu_d)\mathbf{1}_\mathcal{H}|\overline{A}, \overline{B}, \boldsymbol{\sigma}_*] = r^2 \sum_{H \in \mathcal{T}} \text{aut}(H) W_{i,H}(\overline{A}) W_{j,H}(\overline{B})\mathbf{1}_\mathcal{H} = r^2 \Phi_{ij}\mathbf{1}_\mathcal{H}.$$

Note that $X_{i,H}(\overline{A}, \mu)$ and $X_{j,H}(\overline{B}, \nu)$ are independent conditioned on $\overline{A}, \overline{B}$. Moreover, note that $\{Y_{ij}(\mu_c, \nu_d)\}_{1 \leq c, d \leq t}$ are identically distributed. Hence, we have

$$\mathbb{E}[\widetilde{\Phi}_{ij}\mathbf{1}_\mathcal{H}] = \mathbb{E}[\mathbb{E}_{\mu,\nu}[\frac{1}{r^2 t^2} \sum_{c=1}^t \sum_{d=1}^t Y_{ij}(\mu_c, \nu_d)\mathbf{1}_\mathcal{H}|\overline{A}, \overline{B}, \boldsymbol{\sigma}_*]] = \mathbb{E}[\Phi_{ij}\mathbf{1}_\mathcal{H}].$$

Next, we bound the variance of $\widetilde{\Phi}_{ij}$. In particular, we can get

$$\text{Var}[(\widetilde{\Phi}_{ij} - \Phi_{ij})\mathbf{1}_\mathcal{H}] = \text{Var}(\mathbb{E}[(\widetilde{\Phi}_{ij} - \Phi_{ij})\mathbf{1}_\mathcal{H}|\overline{A}, \overline{B}, \boldsymbol{\sigma}_*]) + \mathbb{E}[\text{Var}((\widetilde{\Phi}_{ij} - \Phi_{ij})\mathbf{1}_\mathcal{H}|\overline{A}, \overline{B}, \boldsymbol{\sigma}_*)]$$
$$= \mathbb{E}[\text{Var}(\widetilde{\Phi}_{ij}|\overline{A}, \overline{B}, \boldsymbol{\sigma}_*)\mathbf{1}_\mathcal{H}],$$

because $\Phi_{ij}$ and $\mathbf{1}_\mathcal{H}$ are fixed conditional on $\overline{A}, \overline{B}, \boldsymbol{\sigma}_*$. Furthermore, conditioned on $\overline{A}$ and $\overline{B}$, for any $1 \leq c, d, e, f \leq t$, $Y_{ij}(\mu_c, \nu_d)$ are independent with $Y_{ij}(\mu_e, \nu_f)$ if and only if $c \neq e$ and $d \neq f$.

$$\text{Var}[(\widetilde{\Phi}_{ij} - \Phi_{ij})\mathbf{1}_\mathcal{H}] \leq \frac{1}{r^4 t^4} \sum_{c=1}^t \sum_{d=1}^t \sum_{e=1}^t \sum_{f=1}^t \mathbb{E}[\text{Cov}(Y_{ij}(\mu_c, \nu_d), Y_{ij}(\mu_e, \nu_f))|\overline{A}, \overline{B}, \boldsymbol{\sigma}_*)\mathbf{1}_\mathcal{H}].$$

Applying Lemma 16, we have

$$\text{Var}[(\widetilde{\Phi}_{ij} - \Phi_{ij})\mathbf{1}_\mathcal{H}] \leq \frac{1}{r^4 t^4} \sum_{c=1}^t \sum_{d=1}^t \sum_{e=1}^t \sum_{f=1}^t (r^{2+\mathbf{1}_{\{c \neq e\}}+\mathbf{1}_{\{d \neq f\}}} - r^4)\Gamma_{ij}$$
$$= \frac{1}{r^2 t^2} \left( t^2 r^2 + 2t^3 r^3 - (t^2 + 2t^3)r^4 \right) \Gamma_{ij}$$
$$\leq 3\Gamma_{ij},$$

where the last inequality holds because $t = \lceil 1/r \rceil, tr \geq 1$.

**Regime II:** $sD_+(a,b) < 1$. With community label estimates $\widehat{\boldsymbol{\sigma}}_A$ and $\widehat{\boldsymbol{\sigma}}_B$, we define

$$Y_{ij}^{\widehat{\sigma}}(\mu, \nu) := \sum_{H \in \mathcal{T}} \text{aut}(H) X_{i,H}(\overline{A}^{\widehat{\sigma}_A}, \mu) X_{j,H}(\overline{B}^{\widehat{\sigma}_B}, \nu), \tag{31}$$

where $\mu, \nu$ are two $(N+1)$–coloring of the vertices in $[n]$. The proof for Regime I still holds in this case by replacing $\overline{A}, \overline{B}, Y_{ij}, \Phi_{ij}, \widetilde{\Phi}_{ij}$ and $\Gamma_{ij}$ with $\overline{A}^{\widehat{\sigma}_A}, \overline{B}^{\widehat{\sigma}_B}, Y_{ij}^{\widehat{\sigma}}, \Phi_{ij}^{\widehat{\sigma}}, \widetilde{\Phi}_{ij}^{\widehat{\sigma}}$ and $\Gamma'_{ij}$ correspondingly. $\square$

**Lemma 16** (Extension of Lemma 12 in [41] to correlated SBMs). *Fix constants $a \neq b > 0, s \in [0,1]$. Let $(G_1, G_2) \sim \mathrm{CSBM}(n, a\frac{\log n}{n}, b\frac{\log n}{n}, s)$. Fix any $1 \leq c, d, e, f \leq t$, and $i, j \in [n]$. If $sD_+(a,b) > 1$, then*

$$\mathbb{E}[\mathrm{Cov}(Y_{ij}(\mu_c, \nu_d), Y_{ij}(\mu_e, \nu_f)|\overline{A}, \overline{B}, \boldsymbol{\sigma}_*)\mathbf{1}_{\mathcal{H}}] \leq (r^{2+\mathbf{1}_{\{d \neq f\}}+\mathbf{1}_{\{c \neq e\}}} - r^4)\Gamma_{ij}. \tag{32}$$

*If $sD_+(a,b) < 1$, then*

$$\mathbb{E}[\mathrm{Cov}(Y_{ij}(\mu_c, \nu_d), Y_{ij}(\mu_e, \nu_f)|\overline{A}^{\widehat{\boldsymbol{\sigma}}_A}, \overline{B}^{\widehat{\boldsymbol{\sigma}}_B}, \boldsymbol{\sigma}_*)\mathbf{1}_{\mathcal{H}}] \leq (r^{2+\mathbf{1}_{\{d \neq f\}}+\mathbf{1}_{\{c \neq e\}}} - r^4)\Gamma'_{ij}. \tag{33}$$

*Proof.* We first prove (33) and then explain why it implies (32).

For two independent $(N+1)$–colorings $\mu$ and $\nu$ of the vertices in $[n]$. From the definition of $Y_{ij}^{\widehat{\boldsymbol{\sigma}}}(\mu, \nu)$ (31), $\mathbb{E}[Y_{ij}^{\widehat{\boldsymbol{\sigma}}}(\mu, \nu)|\overline{A}^{\widehat{\boldsymbol{\sigma}}_A}, \overline{B}^{\widehat{\boldsymbol{\sigma}}_B}, \boldsymbol{\sigma}_*] = r^2 \Phi_{ij}^{\widehat{\boldsymbol{\sigma}}}$. Then,

$$\mathrm{Cov}(Y_{ij}^{\widehat{\boldsymbol{\sigma}}}(\mu_c, \nu_d), Y_{ij}^{\widehat{\boldsymbol{\sigma}}}(\mu_e, \nu_f)|\overline{A}^{\widehat{\boldsymbol{\sigma}}_A}, \overline{B}^{\widehat{\boldsymbol{\sigma}}_B}, \boldsymbol{\sigma}_*)$$
$$= \mathbb{E}[Y_{ij}^{\widehat{\boldsymbol{\sigma}}}(\mu_c, \nu_d)Y_{ij}^{\widehat{\boldsymbol{\sigma}}}(\mu_e, \nu_f)|\overline{A}^{\widehat{\boldsymbol{\sigma}}_A}, \overline{B}^{\widehat{\boldsymbol{\sigma}}_B}, \boldsymbol{\sigma}_*] - r^4(\Phi_{ij}^{\widehat{\boldsymbol{\sigma}}})^2. \tag{34}$$

From the definition of $Y_{ij}^{\widehat{\boldsymbol{\sigma}}}(\mu, \nu)$ in (31) again,

$$\mathbb{E}[Y_{ij}(\mu_c, \nu_d)Y_{ij}(\mu_e, \nu_f)|\overline{A}^{\widehat{\boldsymbol{\sigma}}_A}, \overline{B}^{\widehat{\boldsymbol{\sigma}}_B}, \boldsymbol{\sigma}_*]$$
$$= \mathbb{E}[\sum_{H \in \mathcal{T}} \mathrm{aut}(H)X_{i,H}(\overline{A}^{\widehat{\boldsymbol{\sigma}}_A}, \mu_c)X_{j,H}(\overline{B}^{\widehat{\boldsymbol{\sigma}}}, \nu_d)$$
$$\times \sum_{I \in \mathcal{T}} \mathrm{aut}(H)X_{i,I}(\overline{A}^{\widehat{\boldsymbol{\sigma}}_A}, \mu_e)X_{j,I}(\overline{B}^{\widehat{\boldsymbol{\sigma}}_B}, \nu_f)|\overline{A}^{\widehat{\boldsymbol{\sigma}}_A}, \overline{B}^{\widehat{\boldsymbol{\sigma}}_B}, \boldsymbol{\sigma}_*].$$

From (27) and the independence of colorings,

$$\mathbb{E}[Y_{ij}(\mu_c, \nu_d)Y_{ij}(\mu_e, \nu_f)|\overline{A}^{\widehat{\boldsymbol{\sigma}}_A}, \overline{B}^{\widehat{\boldsymbol{\sigma}}_B}, \boldsymbol{\sigma}_*]$$
$$= \mathbb{E}\Bigg[\sum_{H,I \in \mathcal{T}} \mathrm{aut}(H)\mathrm{aut}(I) \sum_{S_1(i), S_2(j) \cong H} \sum_{T_1(i), T_2(j) \cong I} \chi_{\mu_c}(V(S_1))\overline{A}_{S_1}^{\widehat{\boldsymbol{\sigma}}_A}$$
$$\times \chi_{\nu_d}(V(S_2))\overline{B}_{S_2}^{\widehat{\boldsymbol{\sigma}}_B} \times \chi_{\mu_e}(V(T_1))\overline{A}_{T_1}^{\widehat{\boldsymbol{\sigma}}_A} \times \chi_{\nu_f}(V(T_2))\overline{B}_{T_2}^{\widehat{\boldsymbol{\sigma}}_B}|\overline{A}^{\widehat{\boldsymbol{\sigma}}_A}, \overline{B}^{\widehat{\boldsymbol{\sigma}}_B}, \boldsymbol{\sigma}_*\Bigg]$$
$$= \sum_{H,I \in \mathcal{T}} \mathrm{aut}(H)\mathrm{aut}(I) \sum_{S_1(i), S_2(j) \cong H} \sum_{T_1(i), T_2(j) \cong I} \mathbb{E}[\chi_{\mu_c}(V(S_1))\chi_{\mu_e}(V(T_1))]$$
$$\times \mathbb{E}[\chi_{\nu_d}(V(S_2))\chi_{\nu_f}(V(T_2))]\overline{A}_{S_1}^{\widehat{\boldsymbol{\sigma}}_A}\overline{B}_{S_2}^{\widehat{\boldsymbol{\sigma}}_B}\overline{A}_{T_1}^{\widehat{\boldsymbol{\sigma}}_A}\overline{B}_{T_2}^{\widehat{\boldsymbol{\sigma}}_B}. \tag{35}$$

From (34), (35), and (10),

$$\mathbb{E}[\mathrm{Cov}(Y_{ij}(\mu_c, \nu_d), Y_{ij}(\mu_e, \nu_f)|\overline{A}^{\widehat{\boldsymbol{\sigma}}_A}, \overline{B}^{\widehat{\boldsymbol{\sigma}}_B}, \boldsymbol{\sigma}_*)\mathbf{1}_{\mathcal{H}}]$$
$$= \sum_{H,I \in \mathcal{T}} \mathrm{aut}(H)\mathrm{aut}(I) \sum_{S_1(i), S_2(j) \cong H} \sum_{T_1(i), T_2(j) \cong I} \mathbb{E}[\overline{A}_{S_1}^{\widehat{\boldsymbol{\sigma}}_A}\overline{B}_{S_2}^{\widehat{\boldsymbol{\sigma}}_B}\overline{A}_{T_1}^{\widehat{\boldsymbol{\sigma}}_A}\overline{B}_{T_2}^{\widehat{\boldsymbol{\sigma}}_B}\mathbf{1}_{\mathcal{H}}]$$
$$\times \left(\mathbb{E}[\chi_{\mu_c}(V(S_1))\chi_{\mu_e}(V(T_1))]\mathbb{E}[\chi_{\nu_d}(V(S_2))\chi_{\nu_f}(V(T_2))] - r^4\right).$$

Observe that $\mathbb{E}[\chi_{\mu_c}(V(S_1))\chi_{\mu_e}(V(T_1))] \leq r^{1+\mathbf{1}_{\{c \neq e\}}}$ and $\mathbb{E}[\chi_{\nu_d}(V(S_2))\chi_{\nu_f}(V(T_2))] \leq r^{1+\mathbf{1}_{\{d \neq f\}}}$,

$$\mathbb{E}[\mathrm{Cov}(Y_{ij}(\mu_c, \nu_d), Y_{ij}(\mu_e, \nu_f)|\overline{A}^{\widehat{\boldsymbol{\sigma}}_A}, \overline{B}^{\widehat{\boldsymbol{\sigma}}_B}, \boldsymbol{\sigma}_*)\mathbf{1}_{\mathcal{H}}]$$
$$\leq \sum_{H,I \in \mathcal{T}} \mathrm{aut}(H)\mathrm{aut}(I) \sum_{S_1(i), S_2(j) \cong H} \sum_{T_1(i), T_2(j) \cong I} \mathbb{E}[\overline{A}_{S_1}^{\widehat{\boldsymbol{\sigma}}_A}\overline{B}_{S_2}^{\widehat{\boldsymbol{\sigma}}_B}\overline{A}_{T_1}^{\widehat{\boldsymbol{\sigma}}_A}\overline{B}_{T_2}^{\widehat{\boldsymbol{\sigma}}_B}\mathbf{1}_{\mathcal{H}}](r^{2+\mathbf{1}_{\{d \neq f\}}+\mathbf{1}_{\{c \neq e\}}} - r^4)$$
$$= \Gamma'_{ij}(r^{2+\mathbf{1}_{\{d \neq f\}}+\mathbf{1}_{\{c \neq e\}}} - r^4),$$

and this completes the proof for Regime II.

In Regime I, $\mathbb{E}[\overline{A}_{S_1}\overline{B}_{S_2}\overline{A}_{T_1}\overline{B}_{T_2}\mathbf{1}_{\mathcal{H}}] \neq 0$ only if $\{S_1 \triangle T_1 \subset S_2 \cup T_2, S_2 \triangle T_2 \subset S_1 \cup T_1\}$ happens. In addition, if $S_1 = S_2$, $T_1 = T_2$, and $V(S_1) \cap V(T_1) = \{i\}$, then $\mathbb{E}[\chi_{\mu_c}(V(S_1))\chi_{\mu_e}(V(T_1))] = \mathbb{E}[\chi_{\nu_d}(V(S_2))\chi_{\nu_f}(V(T_2))] = r^2$, because $S_1$ (resp. $S_2$) shares no common edges with $T_1$ (resp. $T_2$). Therefore,

$$\mathbb{E}[\mathrm{Cov}(Y_{ij}(\mu_c, \nu_d), Y_{ij}(\mu_e, \nu_f)|\overline{A}, \overline{B}, \boldsymbol{\sigma}_*)\mathbf{1}_{\mathcal{H}}]$$
$$\leq \sum_{H,I \in \mathcal{T}} \mathrm{aut}(H)\mathrm{aut}(I) \sum_{S_1(i),S_2(j) \cong H} \sum_{T_1(i),T_2(j) \cong I} \mathbb{E}[\overline{A}_{S_1}\overline{B}_{S_2}\overline{A}_{T_1}\overline{B}_{T_2}\mathbf{1}_{\mathcal{H}}](r^{2+\mathbf{1}_{\{d \neq f\}}+\mathbf{1}_{\{c \neq e\}}} - r^4)$$
$$= \sum_{H,I \in \mathcal{T}} \mathrm{aut}(H)\mathrm{aut}(I) \sum_{S_1(i),S_2(j) \cong H} \sum_{T_1(i),T_2(j) \cong I} \mathbb{E}[\overline{A}_{S_1}\overline{B}_{S_2}\overline{A}_{T_1}\overline{B}_{T_2}\mathbf{1}_{\mathcal{H}}](r^{2+\mathbf{1}_{\{d \neq f\}}+\mathbf{1}_{\{c \neq e\}}} - r^4)$$
$$\times \mathbf{1}_{\{S_1 \neq S_2 \text{ or } T_1 \neq T_2 \text{ or } V(S_1) \cap V(T_1) \neq \{i\}\}}\mathbf{1}_{\{S_1 \triangle T_1 \subset S_2 \cup T_2, S_2 \triangle T_2 \subset S_1 \cup T_1\}}$$
$$= \Gamma_{ij}(r^{2+\mathbf{1}_{\{d \neq f\}}+\mathbf{1}_{\{c \neq e\}}} - r^4),$$

where the first line follows from the proof in Regime II and the last line applies the definition of $\Gamma_{ij}$ (28). $\qquad\square$

Now, we are ready to prove Theorem 4.

**Proof of Theorem 4:** *For the first part of this proof*, our goal is to show that the estimated score preserves the asymptotic upper bound on $\frac{\mathrm{Var}[\Phi_{ij}\mathbf{1}_{\mathcal{H}}]}{\mathbb{E}[\Phi_{i\pi_*(i)}\mathbf{1}_{\mathcal{H}}]^2}$ for Regime I and $\frac{\mathrm{Var}[\Phi_{ij}^{\widehat{\sigma}}\mathbf{1}_{\mathcal{H}}]}{\mathbb{E}[\Phi_{i\pi_*(i)}^{\widehat{\sigma}}\mathbf{1}_{\mathcal{H}}]^2}$ for Regime II.

For Regime I ($sD_+(a,b) > 1$), we have

$$\frac{\mathrm{Var}[\widetilde{\Phi}_{ij}\mathbf{1}_{\mathcal{H}}]}{\mathbb{E}[\widetilde{\Phi}_{i\pi_*(i)}\mathbf{1}_{\mathcal{H}}]^2} = \frac{\mathrm{Var}[\widetilde{\Phi}_{ij}\mathbf{1}_{\mathcal{H}}]}{\mathbb{E}[\Phi_{i\pi_*(i)}\mathbf{1}_{\mathcal{H}}]^2}$$
$$= \frac{1}{\mathbb{E}[\Phi_{i\pi_*(i)}\mathbf{1}_{\mathcal{H}}]^2} \left[ \mathrm{Var}[\Phi_{ij}\mathbf{1}_{\mathcal{H}}] + \mathrm{Var}[(\widetilde{\Phi}_{ij} - \Phi_{ij})\mathbf{1}_{\mathcal{H}}] + 2\,\mathrm{Cov}((\widetilde{\Phi}_{ij} - \Phi_{ij})\mathbf{1}_{\mathcal{H}}, \Phi_{ij}\mathbf{1}_{\mathcal{H}}) \right],$$

where the first equality is from the fact that $\widetilde{\Phi}_{ij}\mathbf{1}_{\mathcal{H}}$ is an unbiased estimator of $\Phi_{ij}\mathbf{1}_{\mathcal{H}}$ conditioned on $\overline{A}, \overline{B},$ and $\boldsymbol{\sigma}_*$. Further,

$$\frac{\mathrm{Var}[\widetilde{\Phi}_{ij}\mathbf{1}_{\mathcal{H}}]}{\mathbb{E}[\widetilde{\Phi}_{i\pi_*(i)}\mathbf{1}_{\mathcal{H}}]^2} \leq \frac{1}{\mathbb{E}[\Phi_{i\pi_*(i)}\mathbf{1}_{\mathcal{H}}]^2} \left[ \Gamma_{ij} + 3\Gamma_{ij} + 2\mathbb{E}[(\widetilde{\Phi}_{ij} - \Phi_{ij})\Phi_{ij}\mathbf{1}_{\mathcal{H}}] \right] \leq 4\frac{\Gamma_{ij}}{\mathbb{E}[\Phi_{i\pi_*(i)}\mathbf{1}_{\mathcal{H}}]^2},$$

where the first inequality holds from (28), Lemma 15, and $\mathbb{E}[(\widetilde{\Phi}_{ij} - \Phi_{ij})\mathbf{1}_{\mathcal{H}}] = 0$, and the second inequality holds because $\mathbb{E}[((\widetilde{\Phi}_{ij} - \Phi_{ij})\Phi_{ij}\mathbf{1}_{\mathcal{H}})] = \mathbb{E}[\mathbb{E}[(\widetilde{\Phi}_{ij} - \Phi_{ij})|\overline{A}, \overline{B}, \boldsymbol{\sigma}_*]\Phi_{ij}\mathbf{1}_{\mathcal{H}}] = 0$.

The proof of Proposition 2 and Proposition 3 are based on analyzing $\Gamma_{ij}$ and thus the upper bounds on $\frac{\mathrm{Var}[\Phi_{ij}\mathbf{1}_{\mathcal{H}}]}{\mathbb{E}[\Phi_{i\pi_*(i)}\mathbf{1}_{\mathcal{H}}]^2}$ all hold for $\frac{\Gamma_{ij}}{\mathbb{E}[\Phi_{i\pi_*(i)}\mathbf{1}_{\mathcal{H}}]^2}$. Therefore, we conclude that for all $i \in [n]$, if (22) holds, then

$$\frac{\mathrm{Var}[\widetilde{\Phi}_{i\pi_*(i)}\mathbf{1}_{\mathcal{H}}]}{\mathbb{E}[\widetilde{\Phi}_{i\pi_*(i)}\mathbf{1}_{\mathcal{H}}]^2} = O\left( \frac{L^2}{\rho^2 ns(p \wedge q)} + \frac{L^2}{\rho^{2(K+M)}|\mathcal{J}|} \right);$$

and that for all $i, j \in [n]$, $j \neq \pi_*(i)$, if (23) holds, then

$$\frac{\mathrm{Var}[\widetilde{\Phi}_{ij}\mathbf{1}_{\mathcal{H}}]}{\mathbb{E}[\widetilde{\Phi}_{i\pi_*(i)}\mathbf{1}_{\mathcal{H}}]^2} = O\left( \frac{1}{|\mathcal{T}|\rho^{2N}} \right).$$

For Regime II ($sD_+(a,b) < 1$), from (29) and Lemma 15,

$$\frac{\mathrm{Var}[\widetilde{\Phi}_{ij}^{\widehat{\sigma}}\mathbf{1}_{\mathcal{H}}]}{\mathbb{E}[\widetilde{\Phi}_{ij}^{\widehat{\sigma}}\mathbf{1}_{\mathcal{H}}]^2} \leq \frac{1}{\mathbb{E}[\Phi_{i\pi_*(i)}^{\widehat{\sigma}}\mathbf{1}_{\mathcal{H}}]^2} \left[ \Gamma'_{ij} + 3\Gamma'_{ij} + 2\mathbb{E}[(\widetilde{\Phi}_{ij}^{\widehat{\sigma}} - \Phi_{ij}^{\widehat{\sigma}})\Phi_{ij}^{\widehat{\sigma}}\mathbf{1}_{\mathcal{H}}]) \right] \leq 4\frac{\Gamma'_{ij}}{\mathbb{E}[\Phi_{i\pi_*(i)}^{\widehat{\sigma}}\mathbf{1}_{\mathcal{H}}]^2},$$

The proof of Proposition 5 and Proposition 6 are based on analyzing $\Gamma'_{ij}$ and thus the upper bounds on $\frac{\mathrm{Var}[\Phi^{\widehat{\sigma}}_{ij}\mathbf{1}_{\mathcal{H}}]}{\mathbb{E}[\Phi^{\widehat{\sigma}}_{i\pi_*(i)}\mathbf{1}_{\mathcal{H}}]^2}$ all hold for $\frac{\Gamma'_{ij}}{\mathbb{E}[\Phi^{\widehat{\sigma}}_{i\pi_*(i)}\mathbf{1}_{\mathcal{H}}]^2}$. Therefore, we conclude that for all $i \in [n]$, if (24) holds, then

$$\frac{\mathrm{Var}[\widetilde{\Phi}^{\widehat{\sigma}}_{i\pi_*(i)}\mathbf{1}_{\mathcal{H}}]}{\mathbb{E}[\widetilde{\Phi}^{\widehat{\sigma}}_{i\pi_*(i)}\mathbf{1}_{\mathcal{H}}]^2} = O\left(\frac{L^2}{\rho^2 ns(p \wedge q)} + \frac{L^2}{\rho^{2(K+M)}|\mathcal{J}|}\right);$$

and that for all $i, j \in [n]$, $j \neq \pi_*(i)$, if (25) holds, then

$$\frac{\mathrm{Var}[\widetilde{\Phi}^{\widehat{\sigma}}_{ij}\mathbf{1}_{\mathcal{H}}]}{\mathbb{E}[\widetilde{\Phi}^{\widehat{\sigma}}_{i\pi_*(i)}\mathbf{1}_{\mathcal{H}}]^2} = O\left(\frac{1}{|\mathcal{T}|\rho^{2N}}\right).$$

*For the second part of this proof, we are going to show the time complexity of Algorithm 4.*

First, we know that step 1-1 costs time $O(\beta^K)$ by the algorithm in [6], step 1-2 takes time $O(K)$ by the algorithm in [12], and step 1-3 takes time $O(K)$ by enumerating through all edges. In summary, he total time complexity to generate $\mathcal{J}$ is $O(K^2 \alpha^K)$. Afterwards, it takes $O(|\mathcal{T}|)$ time to complete step 1-4, the generation of chandelier class.

The signed counts estimation step takes $O(|\mathcal{T}|N3^N n^2)$ times as shown in the proof of Proposition 5 in [41]. Then, the total time complexity is

$$O\left(K^2\beta^K + |\mathcal{T}|(1 + N(3\mathrm{e})^N n^2)\right) = O\left(\binom{|\mathcal{J}|}{L}N(3\mathrm{e})^N n^2\right)$$
$$= O\left(\beta^{KL}(3\mathrm{e})^N n^2\right) = O\left((3\mathrm{e}\beta)^N n^2\right).$$

Under condition (3) and for large enough $\log(ns(p \wedge q)) \geq \log(2)$, we have

$$N = (K+M)L = \frac{C'\log n}{\varepsilon}\left(1 + \frac{C''}{\log(ns(p \wedge q))}\right) \leq \frac{C'(1 + \frac{C''}{\log 2})}{\varepsilon}\log n,$$

for some constants $C'$ and $C''$. Hence, there exists some constant $C$ depending only on $\varepsilon$ such that the total time complexity of Algorithm 4 is $O(n^C)$.

# G   Proof of Theorem 5: Exact graph matching by seeded graph matching

For a pair of correlated SBMs $(G_1, G_2) \sim \mathrm{CSBM}(n, p, q, s)$, where $p = \frac{a\log n}{n}$ and $q = \frac{b\log n}{n}$, Algorithm 4 efficiently matches $(1 - o(1))n$ vertices correctly with high probability. Our next step is to finish the matching on those remaining vertices.

To do this, we use the seeded graph matching algorithm[7] (Algorithm 2): Starting with an initial partial matching on at least $(1 - \varepsilon/16)n$ vertices which is correct on whatever it matches, we form new matches between vertex $i$ in $G_1$ and vertex $j$ in $G_2$ if the common neighbors of those two vertices under the current partial matching sufficiently large. We then update the partial matching and repeat this rule until we get a complete matching, which will be shown happens with high probability. Define $h(x) = x\log x - x + 1$. The sufficiently large common neighbors threshold is set as $\gamma\frac{p^2+q^2}{2}s^2(n + 2n^{\frac{3}{4}})$, where $\gamma \in (1, \infty)$ such that $h(\gamma) = \frac{3\log n}{(n-2)pqs^2}$.

Denote $\mathrm{N}_\pi(i, j)$ as the number of common neighbors of $i$ and $j$ under correspondence $\pi$. In another word, $\mathrm{N}_\pi(i, j)$ is the number of vertex $v \in I$ such that $v$ is a neighbor of $i$ in $A$ and $\pi(v)$ is a neighbor of $j$ in $B$. If $\pi = \pi_*$, we also write it as $\mathrm{N}(i, j)$.

The following lemma for SBM is an analogy to Lemma 13 in [41], which studied the property of Erdős–Rényi graph.

**Lemma 17.** *Fix $\varepsilon > 0$ and $a, b > 0$ such that $\frac{a+b}{2} \geq 1 + \varepsilon$. Let $p = a\frac{\log n}{n}, q = b\frac{\log n}{n}$ and $G \sim \mathrm{SBM}(n, p, q)$. Let $I \subset [n]$ be a subset of vertices, $e_G(I, I^c)$ denotes the number of edges between vertices in $I$ and vertices in $I^c = [n] \setminus I$. With probability $1 - n^{-\frac{\varepsilon}{8}}$, for any $I$ such*

---

[7]Relevant seeded graph matching algorithms also occur in [69, 5, 40].

*that* $|I| \leq \frac{\varepsilon}{16}n$, $e_G(I, I^c) \geq \eta|I||I^c|p \wedge q$, *where $\eta$ is the unique solution in $(0, 1)$ such that* $h(\eta) = \frac{(1 + \frac{\varepsilon}{8}) \log n}{(p+q)(\frac{n}{2} - n^{3/4} - \frac{\varepsilon}{16}n)}$. *In particular,*

$$\eta \geq 1 - \sqrt{\frac{1 + \frac{\varepsilon}{8}}{(1 + \varepsilon)(1 - \frac{\varepsilon}{7})}}.$$

*Proof.* If $I = \emptyset$, $e_G(I, I^c) \geq \eta|I||I^c|\frac{p+q}{2}$ holds trivially. Let $I$ be an non-empty set with $1 \leq |I| \leq \frac{\varepsilon}{16}n$. We denote $k$ as $|I|$. The number of edges between $I$ and $I^c$ is the summation of $k(n - k)$ independent Bernoulli trails. For arbitrary vertex $i \in I$ and vertex $j \in I^c$, if they have the same community label, then the Bernoulli trail between them has mean $p$. Otherwise, the Bernoulli trail has mean $q$. We assume that there are $N_1$ trails with mean $p$ and $N_2$ trails with mean $q$, where $N_1 + N_2 = k(n - k)$. We can write out the distribution as $e_G(I, I^c) \sim \mathrm{Binom}(N_1, p) + \mathrm{Binom}(N_2, q)$.

Under the balanced community event $\mathcal{H}$, $\frac{n}{2} - n^{3/4} \leq |V^+|, |V^-| \leq \frac{n}{2} + n^{3/4}$. Assume that there are $k_+$ vertices in $I$ with label $+1$ and the remaining $k_-$ vertices with label $-1$.

$$\frac{N_1}{k} = \frac{1}{k}\left(k_+(|V^+| - k_+) + k_-(|V^-| - k_-)\right) \geq \frac{n}{2} - n^{3/4} - \frac{k_+^2 + k_-^2}{k} \geq \frac{n}{2} - n^{3/4} - \frac{\varepsilon}{16}n, \tag{36}$$

$$\frac{N_2}{k} = \frac{1}{k}\left(k_-(|V^+| - k_+) + k_+(|V^-| - k_-)\right) \geq \frac{n}{2} - n^{3/4} - \frac{2k_1 k_2}{k} \geq \frac{n}{2} - n^{3/4} - \frac{\varepsilon}{32}n. \tag{37}$$

We are interested in the probability of $e_G(I, I^c)$ being less than $n|I||I^c|p \wedge q$:

$$\mathbb{P}(e_G(I, I^c) \leq \eta k(n-k)p \wedge q) \leq \mathbb{P}(e_G(I, I^c) \leq \eta(N_1 p + N_2 q)) \leq \exp(-(N_1 p + N_2 q)h(\eta)),$$

where the first inequality holds because $N_1 + N_2 = k(n - k)$ and $p \wedge q \leq p, q$ and the second inequality holds because of the multiplicative Chernoff bound (Lemma 4).

Let $h(\eta) = \frac{(1 + \frac{\varepsilon}{8}) \log n}{(p+q)(\frac{n}{2} - n^{3/4} - \frac{\varepsilon}{16}n)}$, we can show that

$$\frac{k(1 + \frac{\varepsilon}{8}) \log n}{N_1 p + N_2 q} < \frac{(1 + \frac{\varepsilon}{8}) \log n}{(p+q)(\frac{n}{2} - n^{3/4} - \frac{\varepsilon}{16}n)} \leq \frac{1 + \frac{\varepsilon}{8}}{(1 + \varepsilon)(1 - 2n^{-1/4} - \frac{\varepsilon}{8})} < \frac{1 + \frac{\varepsilon}{8}}{(1 + \varepsilon)(1 - \frac{\varepsilon}{7})} < 1, \tag{38}$$

where the first inequality holds because of (36) and (37), the second inequality holds because $\frac{a+b}{2} \geq 1 + \varepsilon$ and the third inequality holds for sufficiently large $n$. Since $h(\eta) \in (0, 1)$, there is an unique solution of $\eta \in (0, 1)$ due to the monotocity (decreasing) of the function $h(\cdot)$.

Applying the fact that $(N_1 p + N_2 q)h(\eta) > (1 + \frac{\varepsilon}{8}) \log n$ (38) and using an union bound over all subsets $I \subset [n]$ with size $1 \leq |I| \leq \frac{\varepsilon}{16}$, we have

$$\mathbb{P}(\exists I \subset [n] \text{ s.t. } 1 \leq |I| \leq \frac{\varepsilon}{16}, e_G(I, I^c) \leq \eta|I|(n - |I|)p \wedge q)$$

$$\leq \sum_{k=1}^{\frac{\varepsilon}{16}} n^k \mathbb{P}(e_G(I, I^c) \leq \eta k(n-k)p \wedge q) \leq \sum_{k=1}^{\frac{\varepsilon}{16}} n^{k - k(1 + \frac{\varepsilon}{8})} = O(n^{-\frac{\varepsilon}{8}}).$$

Because $h(x) \geq \frac{(x-1)^2}{2}$ for $x \in (0, 1)$, we have $\eta \geq 1 - \sqrt{2h(\eta)} > 1 - \sqrt{\frac{1 + \frac{\varepsilon}{8}}{(1 + \varepsilon)(1 - \frac{\varepsilon}{7})}}$. $\qquad\square$

Now, we are ready to prove Theorem 5.

*Proof of Theorem 5.* **Firstly, we study the size of common neighbors.** For an arbitrary vertex $u \in [n]$ in $G_1$ and $v \in [n]$ in $G_2$ such that $v$ is not the true correspondence of $u$, we study the number of common neighbors $\mathrm{N}(u, v)$ in the intersection graph corresponding to the true permutation.

*Case a: These two vertices come from different communities.* If $v \neq \pi_*(u)$, $\boldsymbol{\sigma}(u)\boldsymbol{\sigma}(v) = -1$, then we have $\mathrm{N}(u, v) \sim \mathrm{Binom}(n - 2, pqs^2)$. By the multiplicative Chernoff bound (Lemma 4) for Binomial distributions, for $\gamma \in (1, \infty)$, we have

$$\mathbb{P}(\mathrm{N}(u, v) \geq \gamma \frac{p^2 + q^2}{2}s^2(n + 2n^{\frac{3}{4}})) < \mathbb{P}(\mathrm{N}(u, v) \geq \gamma(n - 2)pqs^2) \leq e^{-(n-2)pqs^2 h(\gamma)} = n^{-3},$$

where $h(\gamma) = \frac{3\log n}{(n-2)pqs^2} > 1, \gamma \in (1,\infty)$. The first inequality holds because $\frac{p^2+q^2}{2}(n+2n^{\frac{3}{4}}) > (n-2)pq$.

By a union bound over all $u,v$ such that $v \neq \pi_*(u), \boldsymbol{\sigma}(u)\boldsymbol{\sigma}(v) = -1$, we have

$$\mathbb{P}\{\exists v \neq \pi_*(u), \boldsymbol{\sigma}(u)\boldsymbol{\sigma}(v) = -1, \text{ s.t. } \mathrm{N}(u,v) \geq \gamma\frac{p^2+q^2}{2}s^2 n(1+2n^{-\frac{1}{4}})\} = O(\frac{1}{n}).$$

*Case b: These two vertices come from the same community.* If $v \neq \pi_*(u), \boldsymbol{\sigma}(u)\boldsymbol{\sigma}(v) = 1$, then we have $\mathrm{N}(u,v) \sim \mathrm{Binom}(|V^{\boldsymbol{\sigma}(v)}| - 2, p^2s^2) + \mathrm{Binom}(|V^{-\boldsymbol{\sigma}(v)}|, q^2s^2)$ because there are $|V^{\boldsymbol{\sigma}(v)}| - 2$ vertices left from the same community as $u,v$ and $|V^{-\boldsymbol{\sigma}(v)}|$ vertices from the different community. Denote $n_1$ as $|V^{\boldsymbol{\sigma}(v)}| - 2$ and $n_2$ as $|V^{-\boldsymbol{\sigma}(v)}|$, $n_1 + n_2 = n - 2$.

$$\mathbb{P}\{\mathrm{N}(u,v) \geq \gamma\frac{p^2+q^2}{2}s^2(n+2n^{\frac{3}{4}})\} < \mathbb{P}\{\mathrm{N}(u,v) \geq \gamma(n_1p^2 + n_2q^2)s^2\}$$
$$\leq \exp(-(n_1p^2 + n_2q^2)s^2 h(\gamma)) < n^{-3}.$$

The first inequality holds because $\frac{p^2+q^2}{2}(n+2n^{\frac{3}{4}}) > n_1p^2 + n_2q^2$. The last inequality holds because $\frac{3\log n}{(n-2)pq} > \frac{3\log n}{\frac{p^2+q^2}{2}n(1-2n^{-\frac{1}{4}})} > \frac{3\log n}{n_1p^2+n_2q^2}$ for sufficiently large $n$.

Then, with an union bound over all $u,v$ such that $v \neq \pi_*(u), \boldsymbol{\sigma}(u)\boldsymbol{\sigma}(v) = 1$, we have

$$\mathbb{P}\{\exists v \neq \pi_*(u), \boldsymbol{\sigma}(u)\boldsymbol{\sigma}(v) = 1, \text{ s.t. } \mathrm{N}(u,v) \geq \gamma\frac{p^2+q^2}{2}s^2 n(1+2n^{-\frac{1}{4}})\} = O(\frac{1}{n}).$$

Combining the above two cases, we have

$$\mathbb{P}\{\exists v \neq \pi_*(u), \mathrm{N}(u,v) \geq \gamma\frac{p^2+q^2}{2}s^2 n(1+2n^{-\frac{1}{4}})\} = o(1). \tag{39}$$

**Secondly, we show that the algorithm is working properly.** From (39), we assume that for all $v \neq \pi_*(u), \mathrm{N}(u,v) < \gamma\frac{p^2+q^2}{2}s^2(n+2n^{\frac{3}{4}})$. We want to show $\widetilde{\pi} = \pi_*|_J$ in every step of Algorithm 2 by induction. This is true for the initialization of $\widetilde{\pi}$ as a base case, from our assumption for Theorem 5. Suppose that this is true for $t$-th round of the algorithm, then at the $(t+1)$-th round, we have that for all $v \neq \pi_*(u)$,

$$\mathrm{N}_{\widetilde{\pi}}(u,v) \leq \mathrm{N}(u,v) < \gamma\frac{p^2+q^2}{2}s^2(n+2n^{\frac{3}{4}}).$$

Therefore, as the $(t+1)$-th round, the algorithm will still not add an fake correspondence. Either the algorithm terminates, or a new vertex $i$ is added to $J$ such that $\widetilde{\pi}(i) = i$, which preserves the property $\widetilde{\pi} = \pi_*|_J$.

Next, we show that Algorithm 2 always ends with $J = [n]$ by contradiction. Assume that the algorithm terminates with $|J| < n$, then $J^c \neq \emptyset$. Then, by definition, for al $i \in J^c$, $\mathrm{N}_{\widetilde{\pi}}(i, \pi_*(i)) < \gamma\frac{p^2+q^2}{2}(n+2n^{\frac{3}{4}})$. Therefore,

$$e_{G_1 \wedge_{\pi_*} G_2}(J, J^c) = \sum_{i \in J^c} \mathrm{N}_{\widetilde{\pi}(i,\pi_*(i))} \leq |J^c|\gamma\frac{p^2+q^2}{2}s^2(n+2n^{\frac{3}{4}}). \tag{40}$$

On the other side, $G_1 \wedge_{\pi_*} G_2 \sim \mathrm{SBM}(n, s^2p, s^2q)$. If $s^2\frac{p+q}{2} \geq (1+\varepsilon)\frac{\log n}{n}$, then from Lemma 17 (in view of $J^c$ as $I$), with probability of $1 - O(n^{-\frac{\varepsilon}{8}})$,

$$e_{G_1 \wedge_{\pi_*} G_2}(J, J^c) \geq \eta|J||J^c|s^2\frac{p+q}{2} \geq \eta(1-\frac{\varepsilon}{16})n|J^c|s^2(p \wedge q). \tag{41}$$

To reach a contradiction between (40) and (41), the remaining is to prove that

$$\gamma \leq \frac{\eta(1-\frac{\varepsilon}{16})ns(p \wedge q)}{\frac{p^2+q^2}{2}(n+2n^{\frac{3}{4}})}. \tag{42}$$

We observe that $h(\gamma) = \frac{3 \log n}{(n-2)pqs^2} = \Theta(1)$, while $\frac{\eta(1-\frac{\varepsilon}{16})ns(p \wedge q)}{\frac{p^2+q^2}{2}(n+2n^{\frac{3}{4}})} = \Theta(\frac{n}{\log n})$. Therefore (42) holds for sufficiently large $n$.

**Finally, we analyze the time complexity of Algorithm 2.** For each $u \in [n]$ to be added into the seeded set $I$, we update the number of common neighbors $N_{\widetilde{\pi}}(i,j)$ for all $i,j \in I$. This step takes $O(\deg_{G_1}(u) \deg_{G_2}(u))$, where $\deg_G(u)$ denotes the degree of $u$ in graph $G$. With probability $1 - o(\frac{1}{n^2})$, $\deg_{G_1}(u) = O(n(p+q))$. By summing up for all possible $u$ to be added, the time complexity of Algorithm 2 is $O(n^3(p+q)^2)$. $\qquad\square$

# H   Proof of Proposition 1

In this section, we calculate the expectation of similarity score for correlated SBM. Recall that we have defined $\sigma_+^2 := sp(1-sp), \sigma_-^2 := sq(1-sq)$ and that $\sigma_{\text{eff}}^2 := (\frac{\sigma_+^2 + \sigma_-^2}{2})$ throughout this paper.

*Proof of Proposition 1.* By definition of the similarity score,

$$\mathbb{E}[\Phi_{ij}\mathbf{1}_{\mathcal{H}}] = \sum_{H \in \mathcal{T}} \text{aut}(H) \sum_{S(i) \cong H} \sum_{S(j) \cong H} \mathbb{E}\left[\overline{A}_{S(i)}\overline{B}_{S(j)}\mathbf{1}_{\mathcal{H}}\right].$$

The expectation is zero if $S(i) = S(j)$, which implies considering $\pi_*(i) = j$ suffices:

$$\mathbb{E}[\Phi_{ij}\mathbf{1}_{\mathcal{H}}] = (1+o(1)) \sum_{H \in \mathcal{T}} \text{aut}(H) \sum_{S(i) \cong H} \mathbb{E}\left[\overline{A}_{S(i)}\overline{B}_{S(\pi_*(i))} \mid \mathcal{H}\right].$$

Define $X_1$ as the random variable for the number of in-community edges of $S(i)$, which takes possible value from 0 to $N$. Define $|\{S(i) : S(i) \cong H\}|$ as the total number of $S$ rooted at $i$ and isomorphic to $H$, we have

$$\mathbb{E}[\Phi_{i\pi_*(i)}\mathbf{1}_{\mathcal{H}}] = (1+o(1)) \sum_{H \in \mathcal{T}} \text{aut}(H)|\{S(i) : S(i) \cong H\}|\mathbb{E}\left[\mathbb{E}\left[\overline{A}_{S(i)}\overline{B}_{S(\pi_*(i))} \mid X_1, \mathcal{H}\right]\right]$$

$$\leq (1+o(1)) \sum_{H \in \mathcal{T}} n^N \sum_{N_1=0}^{N} \mathbb{P}(X_1 = N_1 \mid \mathcal{H})\rho^N \sigma_+^{2N_1}\sigma_-^{2N-2N_1}.$$

The second inequality holds by $|\{S(i) : S(i) \cong H\}| = \frac{\binom{n}{N}N!}{\text{aut}(H)} = (1+o(1))\frac{n^N}{\text{aut}(H)}$ since $\binom{n}{N}N!$ is the number of possible vertex embedding of the chandelier onto the random graph and also $\mathbb{E}\left[\overline{A}_{S(i)}\overline{B}_{S(\pi_*(i))}|\{X_1 = N_1\} \cap \mathcal{H}\right] = \rho_+^{N_1}\rho_-^{N-N_1}\sigma_+^{2N_1}\sigma_-^{2(N-N_1)} = (1+o(1))\rho^N\sigma_+^{2N_1}\sigma_-^{2(N-N_1)}$.

Then, according to Lemma 9 and the binomial theorem, we have $\mathbb{P}(X_1 = N_1 \mid \mathcal{H}) = (1+o(1))\binom{N}{N_1}2^{-N}$. Therefore,

$$\mathbb{E}[\Phi_{i\pi_*(i)}\mathbf{1}_{\mathcal{H}}] = (1+o(1))|\mathcal{T}|n^N\rho^N \sum_{N_1=0}^{N} \binom{N}{N_1}\frac{1}{2^N}\sigma_+^{2N_1}\sigma_-^{2N-2N_1}$$

$$= (1+o(1))|\mathcal{T}|n^N\rho^N(\frac{\sigma_+^2 + \sigma_-^2}{2})^N = (1+o(1))|\mathcal{T}|n^N\rho^N\sigma_{\text{eff}}^{2N}. \qquad\square$$

# I   Proof of Proposition 2

## I.1   Decorated union graph and union graph partition

The analysis of the first moment involves two chandeliers, while the second moment analysis requires four chandeliers. Before delving into the analysis, we introduce the notation for the decorated union graph and establish a rule for union graph partition. We adopt some notations and definitions from [41] and further introduce concepts that are particularly useful for correlated stochastic block models.

- **(Decorated graph)** For any graph $G$, define $\dot{G}$ as a decorated graph $\dot{G} := (G, D_G)$, where $D_G$ is a decoration mapping from the edge set to a decoration set. Define the edge set of decorated graph $E(\dot{G}) := E(G)$ and the vertex set of decorated graph $V(\dot{G}) := V(G)$.

- **(Decorated union graph)** For any pair of chandeliers $H, I \in \mathcal{T}$, let $S_1(i), S_2(j) \cong H$ and $T_1(i), T_2(j) \cong I$. The union graph is defined as $U = S_1 \cup S_2 \cup T_1 \cup T_2$. Now, let us define the proper decoration for $\dot{U}$:

$$D_U(e) = \begin{cases} \text{The subset of } \{S_1, S_2, T_1, T_2\} \text{ where } e \text{ occurs}, & \text{if } e \in E(U), \\ \emptyset, & \text{if } e \notin E(U). \end{cases} \quad (43)$$

  We call an edge $e \in E(U)$ $t$-decorated if $|D_U(e)| = t$, $t \in \{0, 1, 2, 3, 4\}$.

- **(Decorated set operation)** Assume that $\dot{U} = (U, D_U)$ and $\dot{U}' = (U', D_{U'})$ are two decorated graphs. We define the union, intersection, and difference operations, from which the complement and symmetric difference naturally follow.

  - Union. $\dot{U} \cup \dot{U}' := (U \cup U', D_{U''} : D_{U''}(e) = D_U(e) \cup D_{U'}(e))$.
  - Intersection. $\dot{U} \cap \dot{U}' := (U \cap U', D_{U''} : D_{U''}(e) = D_U(e) \cap D_{U'}(e))$.
  - Difference. $\dot{U} \setminus \dot{U}' := (U \setminus U', D_{U''} : D_{U''}(e) = D_U(e) \setminus D_{U'}(e))$.

- **(Union graph partition)** Assume that $i$ is the root of union graph. Consider the graph $U$ with edge $(i, a)$ removed for all neighbors $a$ of $i$. Let $\dot{\mathcal{C}}(i, a)$ be the connected component therein that contains $a$. Let $\dot{\mathcal{G}}(i, a)$ be the graph union of $\dot{\mathcal{C}}(i, a)$ and the edge $(i, a)$.

  Then, we divide the set of root neighbors $\mathcal{N}(i)$ into the following sets depending on whether $\dot{\mathcal{G}}(i, a)$ is a tree: $\mathcal{N}_T = \{a : (i, a) \in E(U), \dot{\mathcal{G}}(i, a) \text{ is a tree}\}, \mathcal{N}_N = \{a : (i, a) \in E(\dot{U})\} \setminus \mathcal{N}_T$. Furthermore, we breakdown $\mathcal{N}_T$ into two sets depending on whether there are at least $M$ edges 3-decorated: $\mathcal{N}_M = \{a \in \mathcal{N}_T, |\{e \in E(\dot{\mathcal{G}}(i, a)) : |D_U(e)| \geq 3\}| \geq M\}$, $\mathcal{N}_L = \mathcal{N}_T \setminus \mathcal{N}_M,$ .

  Next, we define the decomposition of decorated union graph $\dot{U}$:

$$\dot{U}_L := \bigcup_{a \in \mathcal{N}_L} \dot{\mathcal{G}}(i, a), \quad \dot{U}_M := \bigcup_{a \in \mathcal{N}_M} \dot{\mathcal{G}}(i, a), \quad \dot{U}_N := \dot{U} \setminus (\dot{U}_L \cup \dot{U}_M). \quad (44)$$

  If any of these become an empty set, we define it as the graph consisting of the single vertex $i$. To provide an intuitive understanding: $\dot{U}_N$ contains those chandelier branches that form cycles in the union graph, while $\dot{U}_L$ and $\dot{U}_M$ are the collection of those chandelier branches that do not tangle with other branches from bottom and remain a part of tree rooted at $i$ (or $j$) in the union graph. As a characteristic of trees, the decorations on edges within $\dot{U}_M$ and $\dot{U}_N$ are monotonically decreasing, meaning that the decoration set of an edge at depth $d \geq 1$ is always a subset of the decoration set of the preceding edge at depth $d - 1$ that connects to it. For each node $v \in \dot{U}_M$ that is connected to the root, the connected component $\dot{\mathcal{G}}(i, a)$ should have at list $M$ vertices that are at least 3-decorated. $\dot{U}_L$ is the union of all connected components remained that are trees. The definition of $\dot{U}_L$ implies that the branches cannot be fully 3-decorated, otherwise this branch would fall in $\dot{U}_M$.

For every union graph, we can decompose the decorated union graph based on its decoration sets. Specifically, we want to keep track of how many times each vertex appears on the chandeliers that are rooted at $i$ and the chandeliers that are rooted at $j$. We present the definition as follows.

**Definition 3** (Decorated union graph decomposition by decorations)**.**

$$K_{\ell m} := (V(U), \{e \in E(U) : \ell = \mathbf{1}_{\{e \in S_1\}} + \mathbf{1}_{\{e \in T_1\}}, k = \mathbf{1}_{\{e \in S_2\}} + \mathbf{1}_{\{e \in T_2\}}\}). \quad (45)$$

**Definition 4** (Edge counts on 3 and 4-decorated edges)**.** *Based on the definition of union graph partition and $K_{\ell m}$, we further define*

$$e_L := \frac{1}{2}[e((K_{22} \cup K_{21}) \cap U_L) + e((K_{22} \cup K_{12}) \cap U_L)], \quad (46)$$

$$e_M := \frac{1}{2}[e((K_{22} \cup K_{21}) \cap U_M) + e((K_{22} \cup K_{12}) \cap U_M)], \quad (47)$$

$$e_N := \frac{1}{2}[e((K_{22} \cup K_{21}) \cap U_N) + e((K_{22} \cup K_{12}) \cap U_N)], \quad (48)$$

*where $2e_L$ (resp. $2e_M$, $2e_N$) is the counts of 3-decorated edges and two times the 4-decorated edges on $U_L$ (resp. $U_M$, $U_N$).*

We use $\mathcal{U}_L(v_L, e_L, \ell)$ to denote the collection of all possible $\dot{U}_L$ with $v_L$ vertices except for $i$ and $j$, $e_L$ counts of special edges as defined, and $\ell$ edges in $K_{11} \cap \dot{U}_L$. $\mathcal{U}_M(v_M, e_M)$ denotes the family of all possible $\dot{U}_M$ with $v_M$ vertices except for $i$ and $j$ and $e_L$ counts of special edges as defined. $\mathcal{U}_N(v_N, e_N, k)$ denotes the family of all possible $\dot{U}_N$ with $v_N$ vertices except for $i$ and $j$, $e_N$ counts of special edges as defined, and excess $k$.

We further define the weights for of decorated graphs $\dot{U}_L$, $\dot{U}_M$, and $\dot{U}_N$.

**Definition 5.** *Let $\dot{G}$ be an arbitrary $\dot{U}_L$, $\dot{U}_M$, or $\dot{U}_N$, we define the weight of $\dot{G}$ with regard to a chandelier $S$ as*

$$w_S(\dot{G}) := \prod_{\mathcal{B} \in \mathcal{K}(S), \mathcal{B} \subset G} \mathrm{aut}(\mathcal{B})^{\frac{1}{2}}. \tag{49}$$

*We set $w_S(\dot{G}) = 1$ if $\dot{G}$ contains no bulbs in $\mathcal{K}(S)$. We define the weight of $\dot{G}$ as the multiplication of its weights over $S_1, S_2, T_1, T_2$:*

$$w(\dot{G}) := w_{S_1}(\dot{G}) w_{S_2}(\dot{G}) w_{T_1}(\dot{G}) w_{T_2}(\dot{G}). \tag{50}$$

We have the following observation with regard to the concepts introduced above:

- **(Counting edges)** $2(v_L + v_M + v_N + k + 1 + \mathbf{1}_{\{i \neq j\}}) + 2(e_L + e_M + e_N) = 4N$. This holds because $2(v_L + v_M + v_N + k + 1 + \mathbf{1}_{\{i \neq j\}})$ counts all the edges on union graph twice and $2(e_L + e_M + e_N)$ makes up for an additional count for 3-decorated edges and two counts for 4-decorated edges.

- **(Automorphism as decorated graph weights)** From the definition of $\mathrm{aut}(\cdot)$ and chandelier,

$$(\mathrm{aut}(S_1)\mathrm{aut}(S_2)\mathrm{aut}(T_1)\mathrm{aut}(T_2))^{\frac{1}{2}} = w(\dot{U}_L)w(\dot{U}_M)w(\dot{U}_N). \tag{51}$$

- **(Trivial upper bound on the decorated graph weights)** Since $\mathrm{aut}(\mathcal{B})$ is upper bounded by $R$ for arbitrary bulb $\mathcal{B}$, this inequality follows from the definition of decorated union graph weights

$$w(\dot{U}_L) \leq (\sqrt{R}^{|\mathcal{N}_L|})^4 = R^{2|\mathcal{N}_L|}. \tag{52}$$

  The same holds for $\dot{U}_M$ and $\dot{U}_N$.

Specific to the moment calculation for correlated stochastic block models, we present the following two definitions.

**Definition 6** $(g_{\dot{U}}(\sigma_+, \sigma_-))$**.** *Fix chandeliers $(S_1, S_2, T_1, T_2)$ on the complete graph, conditioned on the ground-truth labels of correlated SBMs, we define a function as follows:*

$$g_{\dot{U}}(\sigma_+, \sigma_-) := \sigma_+^{h(S_1)+h(S_2)+h(T_1)+h(T_2)} \sigma_-^{\overline{h}(S_1)+\overline{h}(S_2)+\overline{h}(T_1)+\overline{h}(T_2)},$$

*where $h(\cdot)$ is the counts of edges connecting two points from the same community, $\overline{h}(\cdot)$ is the counts of edges connecting two points from different communities on the complete graph $K_n$.*

**Definition 7** (Extension of $\sigma_{\mathrm{eff}}^2$)**.** *We define $\gamma_2 := \left(\frac{\sigma_+^4 + \sigma_-^4}{2}\right)/\sigma_{\mathrm{eff}}^4$ and $\gamma_1 := \left(\frac{\sigma_+^3 + \sigma_-^3}{2}\right)/\sigma_{\mathrm{eff}}^3$. Thus, $\left(\frac{\sigma_+^4 + \sigma_-^4}{2}\right)^k = (1 + o(1)) \left(\frac{\sigma_+^2 + \sigma_-^2}{2}\right)^{2k} \gamma_2^k$ and $\left(\frac{\sigma_+^3 + \sigma_-^3}{2}\right)^k = (1 + o(1)) \left(\frac{\sigma_+^2 + \sigma_-^2}{2}\right)^{\frac{3k}{2}} \gamma_1^k$.*

**Remark 5.** *We observe that $\gamma_2 < 2$ and $\gamma_1 \leq \gamma_2$. The first holds simply because $\sigma_+^4 + \sigma_-^4 \geq 2\sigma_+^2 \sigma_-^2$. By Cauchy–Schwarz inequality, for an arbitrary random variable $X$, $\mathbb{E}[X^3]\mathbb{E}[X^2]^2 \leq \mathbb{E}[X^4]\mathbb{E}[X^2]^{3/2}$. Let $X$ takes $\sigma_+^2$ and $\sigma_-^2$ uniformly at random, then $\frac{\mathbb{E}[X^3]}{\mathbb{E}[X^2]^{3/2}} \leq \frac{\mathbb{E}[X^4]}{\mathbb{E}[X^2]^2}$ and thus $\gamma_1 \leq \gamma_2$ is implied.*

## I.2 Proof of the Proposition

In regime $sD_+(a, b) > 1$, we can recover the correct community label with high probability [2, 47, 1]. Therefore we can effectively work on the correct centralized adjacency matrices. Recall that $\mathcal{H} = \{\frac{n}{2} - n^{\frac{3}{4}} \leq |V^+|, |V^-| \leq \frac{n}{2} + n^{\frac{3}{4}}\}$, as defined in Section D.

*Proof of Proposition 2.* From the definition of variance,

$$\text{Var}[\Phi_{ij}\mathbf{1}_{\mathcal{H}}] = \sum_{H,I \in \mathcal{T}} \text{aut}(H)\text{aut}(I) \sum_{S_1,S_2 \cong H, T_1,T_2 \cong I} \text{Cov}(\overline{A}_{S_1}\overline{B}_{S_2}\mathbf{1}_{\mathcal{H}}, \overline{A}_{T_1}\overline{B}_{T_2}\mathbf{1}_{\mathcal{H}}).$$

If $S_1 = S_2, T_1 = T_2$, and $V(S_1) \cap V(T_1) = \{i\}$, the covariance becomes zero. Also, we know from Proposition 1 that $\mathbb{E}[\overline{A}_{S_1}\overline{B}_{S_2}]\mathbb{E}[\overline{A}_{T_1}\overline{B}_{T_2}]$ is either 0 or $(1 + o(1))(\rho\sigma_{\text{eff}})^{2N}$. Thus,

$$\text{Var}[\Phi_{ij}\mathbf{1}_{\mathcal{H}}] \leq \sum_{H,I \in \mathcal{T}} \text{aut}(H)\text{aut}(I) \sum_{S_1,S_2 \cong H, T_1,T_2 \cong I} \mathbb{E}[\overline{A}_{S_1}\overline{B}_{S_2}\overline{A}_{T_1}\overline{B}_{T_2}\mathbf{1}_{\mathcal{H}}].$$

$\mathbb{E}[\overline{A}_{S_1}\overline{B}_{S_2}\overline{A}_{T_1}\overline{B}_{T_2}]$ is non-zero if and only if each edge $e \in U$ is at least 2-decorated. Let $\mathcal{W}_{ij}$ denote the collection of decorated union graph $U_D$, where $S_1(i), S_2(j) \cong H, T_1(i), T_2(j) \cong I$ for some $H, I \in \mathcal{T}$, such that each edge is at least 2-decorated and satisfies $S_1 \neq S_2$ or $T_1 \neq T_2$ or $V(S_1) \cap V(T_1) \neq \{i\}$.

$$\text{Var}[\Phi_{ij}\mathbf{1}_{\mathcal{H}}] \leq \sum_{\dot{U} \in \mathcal{W}_{ij}} (\text{aut}(S_1)\text{aut}(S_2)\text{aut}(T_1)\text{aut}(T_2))^{\frac{1}{2}} \mathbb{E}[\overline{A}_{S_1}\overline{B}_{S_2}\overline{A}_{T_1}\overline{B}_{T_2}\mathbf{1}_{\mathcal{H}}]. \quad (53)$$

Conditioned on the ground-truth labeling $\boldsymbol{\sigma}_*$,

$$g_{\dot{U}}(\sigma_+, \sigma_-) = \prod_{2 \leq \ell+m \leq 4} \prod_{(u,v) \in K_{\ell m}} \sigma_{c(u,v)}^{(\ell+m)},$$

where $c(u, v) = +$ if $\boldsymbol{\sigma}_*(u) = \boldsymbol{\sigma}_*(v)$ and $c(u, v) = -$ if $\boldsymbol{\sigma}_*(u) \neq \boldsymbol{\sigma}_*(v)$.

Conditioned on $\boldsymbol{\sigma}_*$, $\mathcal{H}$ is determined,

$$\mathbb{E}[\overline{A}_{S_1}\overline{B}_{S_2}\overline{A}_{T_1}\overline{B}_{T_2}\mathbf{1}_{\mathcal{H}}] = \mathbb{E}\left[\mathbb{E}[\overline{A}_{S_1}\overline{B}_{S_2}\overline{A}_{T_1}\overline{B}_{T_2} \mid \boldsymbol{\sigma}_*]\mathbf{1}_{\mathcal{H}}\right]. \quad (54)$$

Conditioned on $\boldsymbol{\sigma}_*$, $g_{\dot{U}}(\sigma_+, \sigma_-)$ is fixed, so as $g_{\dot{U}}^{-1}(\sigma_+, \sigma_-)$.

$$g_{\dot{U}}(\sigma_+, \sigma_-)^{-1}\mathbb{E}[\overline{A}_{S_1}\overline{B}_{S_2}\overline{A}_{T_1}\overline{B}_{T_2} \mid \boldsymbol{\sigma}_*] = \prod_{2 \leq \ell+m \leq 4} \prod_{(u,v) \in K_{\ell m}} \sigma_{c(u,v)}^{-(\ell+m)}\mathbb{E}[\overline{A}_{uv}^\ell \overline{B}_{uv}^m \mid \boldsymbol{\sigma}_*].$$

We define $\beta_{\ell m}(e)$ as $\sigma_{c(u,v)}^{-(\ell+m)}\mathbb{E}[\overline{A}_{uv}^\ell \overline{B}_{uv}^m \mid \boldsymbol{\sigma}_*]$, for $e = (u, v) \in K_{\ell,m}$. Thus,

$$g_{\dot{U}}(\sigma_+, \sigma_-)^{-1}\mathbb{E}[\overline{A}_{S_1}\overline{B}_{S_2}\overline{A}_{T_1}\overline{B}_{T_2} \mid \boldsymbol{\sigma}_*] = g_{\dot{U}}(\sigma_+, \sigma_-)^{-1} \prod_{2 \leq \ell+m \leq 4} \prod_{(u,v) \in K_{\ell m}} \beta_{\ell m}(e).$$

Then, we apply Lemma 13 to bound each $\beta_{\ell m}(e)$. Since the upper bound of $\beta_{\ell m}(e)$ only depends on $\ell$ and $m$, we have the following:

$$g_{\dot{U}}(\sigma_+, \sigma_-)^{-1}\mathbb{E}[\overline{A}_{S_1}\overline{B}_{S_2}\overline{A}_{T_1}\overline{B}_{T_2} \mid \boldsymbol{\sigma}_*] \leq |\max\{\rho_+, \rho_-\}|^{e(K_{11})} \frac{(sp \wedge sq)^{-e(K_{22})}}{\sqrt{(sp \wedge sq)}^{(e(K_{21})+e(K_{12}))}}$$

$$= (1 + o(1))\rho^{e(K_{11})}(sp \wedge sq)^{-2N+e(U)}, \quad (55)$$

where the last equality holds because $2\left[\frac{1}{2}(e(K_{12}) + e(K_{21})) + e(K_{22})\right] = (2-2)e(K_{20}) + (2 - 2)e(K_{02}) + (3 - 2)e(K_{12}) + (3 - 2)e(K_{21}) + (4 - 2)e(K_{22}) = 4N - e(U)$. From (54) and (55), and the fact that $\mathcal{H}$ happens with high probability, we have

$$\mathbb{E}[\overline{A}_{S_1}\overline{B}_{S_2}\overline{A}_{T_1}\overline{B}_{T_2}\mathbf{1}_{\mathcal{H}}] = (1 + o(1))\rho^{e(K_{11})}(sp \wedge sq)^{-2N+e(U)}\mathbb{E}[g_{\dot{U}}(\sigma_+, \sigma_-) \mid \mathcal{H}]. \quad (56)$$

It remains to compute $\mathbb{E}\left[g_{\dot{U}}(\sigma_+, \sigma_-) \mid \mathcal{H}\right]$. Here, $\dot{U}$ is fixed and we assume that there are $d_2$ 2-decorated edges, $d_3$ 3-decorated edges, and $d_4$ 4-decorated edges without loss of generality. Let $X^{(2)}, X^{(3)}$, and $X^{(4)}$ be the number of in-community edges for 2, 3, and 4-decorated edges on the complete graph respectively. In $\dot{U}$ is a tree, we can apply Lemma 10 to determine the distribution of $X^{(2)}, X^{(3)}$, and $X^{(4)}$.

Generally, $\dot{U}$ contains cycles. $\dot{U}$ can be decomposed into a tree with an additional set of edges connecting vertices on the tree. We assume that there are $A_i$ $i$-decorated edges on the tree and $B_i$ $i$-decorated edges in the additional edge set of size $e(\dot{U}) - v(\dot{U}) + 1$, for $i \in \{2, 3, 4\}$. We apply the Corollary 1 and have that $\mathbb{P}(X^{(2,a)} = a_2, X^{(3,a)} = a_3, X^{(4,a)} = a_4, X^{(2,b)} = b_2, X^{(3,b)} = b_3, X^{(4,b)} = b_4 \mid \mathcal{H}) \leq (1 + o(1))\binom{A_2}{a_2}\binom{A_3}{a_3}\binom{A_4}{a_4}\frac{1}{2^{A_2+A_3+A_4}}$, where $X^{(i,a)}$ is the number of $i$-decorated in-community edges on the tree-part of $\dot{U}$ and $X^{(i,b)}$ is the number of $i$-decorated in-community edges among the additional edge set.

$A_i$ and $B_i$ are fixed but summed up to $d_i$ for each $\dot{U}$. $a_i$ ($b_i$) takes possible values from $0$ to $A_i$ ($B_i$), for $i \in [4]$. The number of $i$-decorated in-community edges is $a_i + b_i$, and the number of $i$-decorated cross-community edges is $d_i - a_i - b_i = (A_i - a_i) + (B_i - b_i)$.

$$
\begin{aligned}
\mathbb{E}\left[g_{\dot{U}}(\sigma_+, \sigma_-) \mid \mathcal{H}\right] &= \sum_{a_2}^{A_2}\sum_{a_3}^{A_3}\sum_{a_4}^{A_4}\sum_{b_2}^{B_2}\sum_{b_3}^{B_3}\sum_{b_4}^{B_4} \sigma_+^{2(a_2+b_2)}\sigma_-^{2(d_2-a_2-b_2)}\sigma_+^{3(a_3+b_3)}\sigma_-^{3(d_3-a_3-b_3)} \\
&\quad \times \sigma_+^{4(a_4+b_4)}\sigma_-^{4(d_4-a_4-b_4)}\binom{A_2}{a_2}\binom{A_3}{a_3}\binom{A_4}{a_4}\frac{1}{2^{A_2+A_3+A_4}} \\
&= \left(\frac{\sigma_+^2+\sigma_-^2}{2}\right)^{A_2}\left(\frac{\sigma_+^3+\sigma_-^3}{2}\right)^{A_3}\left(\frac{\sigma_+^4+\sigma_-^4}{2}\right)^{A_4} \\
&\quad \times \sum_{b_2=0}^{B_2}\sigma_+^{2b_2}\sigma_-^{2(B_2-b_2)}\sum_{b_3=0}^{B_3}\sigma_+^{3b_3}\sigma_-^{3(B_3-b_3)}\sum_{b_4=0}^{B_4}\sigma_+^{4b_4}\sigma_-^{4(B_4-b_4)} \\
&\leq \left(\frac{\sigma_+^2+\sigma_-^2}{2}\right)^{d_2}\left(\frac{\sigma_+^3+\sigma_-^3}{2}\right)^{d_3}\left(\frac{\sigma_+^4+\sigma_-^4}{2}\right)^{d_4} 2^{k+1},
\end{aligned}
\tag{57}
$$

where the last inequality holds because we can upper bound by multiplying a binomial coefficient $\binom{B_2}{b_2}\binom{B_3}{b_3}\binom{B_4}{b_4}$ and that $B_2 + B_3 + B_4 = k + 1$. By the definition of $\gamma_2$ and $\gamma_1$,

$$
\mathbb{E}\left[g_{\dot{U}}(\sigma_+, \sigma_-) \mid \mathcal{H}\right] \leq \sigma_{\text{eff}}^{2d_2+3d_3+4d_4}\gamma_1^{d_3}\gamma_2^{d_4}2^{k+1} \leq \sigma_{\text{eff}}^{4N}\gamma_2^{2(e_L+e_M+e_N)}2^{k+1}.
\tag{58}
$$

The last inequality holds because $\gamma_1 < \gamma_2$ and $d_3 + d_4 = e(K_{12}) + e(K_{21}) + e(K_{22}) \leq 2(e_L + e_M + e_N)$.

Denote $\mathcal{W}_{ij}(v, k)$ as the subset of $\mathcal{W}_{ij}$ that contains all the elements that have exactly $v$ vertices except for $i$ and $j$ and excess $k$. By applying the definition of union graph partitions, decorated graph weights, and (53),

$$
\begin{aligned}
\text{Var}[\Phi_{ij}\mathbf{1}_{\mathcal{H}}] &\leq (1 + o(1))\sum_{k \geq -1}\sum_v (sp \wedge sq)^{-2N+v+k+1}\sum_{\dot{U} \in \mathcal{W}_{ij}(v,k)}\rho^{e(K_{11})}w(\dot{U}_L)w(\dot{U}_M)w(\dot{U}_N) \\
&\quad \times \sigma_{\text{eff}}^{4N}\gamma_2^{2(e_L+e_M+e_L)}2^{k+1},
\end{aligned}
\tag{59}
$$

For the $\dot{U}_L, \dot{U}_M, \dot{U}_N$ partition, we define

$$
P_L(v_L, e_L, \ell) := \sum_{\dot{U}_L \in \mathcal{U}_L(v_L,e_L,\ell)} w(\dot{U}_L),
\tag{60}
$$

$$
P_M(v_M, e_M) := \sum_{\dot{U}_M \in \mathcal{U}_L(v_M,e_M)} w(\dot{U}_M),
\tag{61}
$$

$$
P_N(v_N, e_N, k) := \sum_{\dot{U}_N \in \mathcal{U}_N(v_N,e_N)} w(\dot{U}_N).
\tag{62}
$$

Then, we can write out the upper bound as

$$\text{Var}[\Phi_{ij}\mathbf{1}_{\mathcal{H}}] \lesssim \sum_k \sum_v (sp \wedge sq)^{-2N+v+k+1} \sum_{v_L,v_M,v_N \geq 0} \sum_{e_L,e_M,e_N,\ell \geq 0} \sigma_{\text{eff}}^{4N} \gamma_2^{2(e_L+e_M+e_L)} 2^{k+1}$$
$$\times \rho^\ell P_L(v_L, e_L, \ell) P_M(v_M, e_M) P_N(v_N, e_N, k).$$

Applying Proposition 1, Lemma 18, Lemma 19, and Lemma 20, we can upper bound the variance by:

$$\text{Var}[\Phi_{ij}\mathbf{1}_{\mathcal{H}}] \lesssim \sum_k \sum_v (sp \wedge sq)^{-2N+v+k+1} \sum_{v_i \geq 0} \sum_{e_i,\ell \geq 0} \rho^\ell (11\beta n)^{v_M} R^{\frac{4e_M}{M}} \mathbf{1}_{\{v_M \leq e_M \frac{4K+4M}{M}\}}$$
$$\times \sigma_{\text{eff}}^{4N} \gamma_2^{2(e_L+e_M+e_L)} \times 2n^{v_L}(1+2L^2)^{e_L} f_{\widetilde{t},t} \mathbf{1}_{\{K+M|e_L+v_L\}} \mathbf{1}_{\{v_L+e_L \leq 2N\}}$$
$$\times n^{v_N}(11\beta)^{v_N}(22R^4(v_N+1)^2)^{k+1} \mathbf{1}_{\{v_N \leq 2(K+M)(k+1)\}}$$

Note that $v_M \leq e_M \frac{4K+4M}{M}$, we have $\sum_{v_M \geq 0}(11\beta)^{v_M} \leq 2(11\beta)^{e_M \frac{4K+4M}{M}}$. Also, we know that $v_N \leq 2N, v_N \leq 2(K+M)(k+1)$. Then, $\sum_{v_N \geq 0}(11\beta)^{v_M} \leq 2(11\beta)^{2(K+M)(k+1)}$. Thus,

$$\text{Var}[\Phi_{ij}\mathbf{1}_{\mathcal{H}}] \lesssim \sum_{k,v}(sp \wedge sq)^{-2N+v+k+1} \sum_{e_L+e_M+e_N=2N-(v+k+1)} 2^3 n^v \left(R^{\frac{4}{M}}(11\beta)^{\frac{4K+4M}{M}}\right)^{e_M}$$
$$\times (22R^4(2N+1)(11\beta)^{2(K+M)})^{k+1}(1+2L^2)^{e_L} \sigma_{\text{eff}}^{4N} \gamma_2^{2(e_L+e_M+e_L)}$$
$$\times \sum_{v_L \geq 0, l \geq 0} \left(\rho^\ell f_{\widetilde{t},t} \mathbf{1}_{\{K+M|e_L+v_L\}} \mathbf{1}_{\{v_L+e_L \leq 2N\}}\right).$$

Regarding to the Lemma 4 in [41], if $\frac{12L^2}{\rho^{2(K+M)}(|\mathcal{J}|-2L)} \leq \frac{1}{2}$, then

$$\text{Var}[\Phi_{ij}\mathbf{1}_{\mathcal{H}}] \lesssim \sum_{k,v}(sp \wedge sq)^{-2N+v+k+1} \sum_{v_L+v_M+v_N+e_L+e_M+e_N=2N} 2^3 n^v \left(R^{\frac{4}{M}}(11\beta)^{\frac{4K+4M}{M}}\right)^{e_M}$$
$$(22R^4(2N+1)(11\beta)^{2(K+M)})^{k+1}(1+2L^2)^{e_L} \sigma_{\text{eff}}^{4N} \gamma_2^{2(e_L+e_M+e_N)}$$
$$8\rho^{2N}\left(\rho^{-2e_L}\mathbf{1}_{e_L \neq 0} + \frac{12L^2}{\rho^{2(K+M)}|\mathcal{J}|}\mathbf{1}_{e_L=0}\right)|\mathcal{T}|^2.$$

Note that $\frac{12L^2}{\rho^{2(K+M)}(|\mathcal{J}|-2L)} \leq \frac{1}{2}$ is guaranteed by the first condition in (22) $\frac{14L^2}{\rho^{2(K+M)}(|\mathcal{J}|)} \leq \frac{1}{2}$ for large enough $n$.

In the following step, we divide $\text{Var}[\Phi_{ij}\mathbf{1}_{\mathcal{H}}]$ by $\mathbb{E}[\Phi_{i\pi_*(i)}\mathbf{1}_{\mathcal{H}}]^2$ and use the fact that $e_N = 2N - (v + k + 1 + e_L + e_M)$:

$$\frac{\text{Var}[\Phi_{ij}\mathbf{1}_{\mathcal{H}}]}{\mathbb{E}[\Phi_{i\pi_*(i)}\mathbf{1}_{\mathcal{H}}]^2} \leq (1+o(1))2^6 \sigma_{\text{eff}}^{4N-4N} \sum_{k \geq -1} \left(\frac{22R^4(2N+1)(11\beta)^{2(K+M)}}{n}\right)^{k+1}$$
$$\sum_{e_M \geq 0}\left(\gamma_2^2 \frac{R^{\frac{4}{M}}(11\beta)^{\frac{4K+4M}{M}}}{ns(p \wedge q)}\right)^{e_M}$$
$$\sum_{e_L \geq 0}\left(\gamma_2^2 \frac{1+2L^2}{ns(p \wedge q)}\right)^{e_L}\left(\rho^{-2e_L}\mathbf{1}_{\{e_L > 0\}} + \frac{12L^2}{\rho^{2(K+M)}|\mathcal{J}|}\mathbf{1}_{\{e_L=0\}}\right)$$
$$\sum_{e_M \geq 0}^{2N}\left(\gamma_2^2(\frac{n}{sp \wedge sq})\right)^{2N-(v+k+1+e_L+e_M)} \mathbf{1}_{\{e_L+e_M \leq 2N-(v+k+1)\}}$$

In view of $\gamma_2 < 2$, the last three conditions in (22) guarantee that

$$\sum_{k \geq -1}\left(\frac{22R^4(2N+1)(11\beta)^{2(K+M)}}{n}\right)^{k+1} \leq 2,$$
$$\sum_{e_M \geq 0}\left(\gamma_2^2 \frac{R^{\frac{4}{M}}(11\beta)^{\frac{4K+4M}{M}}}{ns(p \wedge q)}\right)^{e_M} \leq 2, \quad \sum_{e_L \geq 0}\left(\gamma_2^2 \frac{1+2L^2}{ns(p \wedge q)}\right)^{e_L} \leq 2.$$

Also, for sufficiently large $n$, $\frac{\gamma_2^2}{ns(p\wedge q)} \leq \frac{1}{2}$, such that

$$\sum_{v=0}^{2N-k-1} \left(\frac{\gamma_2^2}{ns(p\wedge q)}\right)^{2N-(v+k+1+e_L+e_M)} \leq 2.$$

In conclusion,

$$\frac{\text{Var}[\Phi_{ij}\mathbf{1}_{\mathcal{H}}]}{\mathbb{E}[\Phi_{i\pi_*(i)}\mathbf{1}_{\mathcal{H}}]^2} \leq O\left(\frac{\gamma_2^2(2+4L^2)}{\rho^2 ns(p\wedge q)} + \frac{12L^2}{\rho^{2(K+M)}|\mathcal{J}|}\right) = O\left(\frac{L^2}{\rho^2 ns(p\wedge q)} + \frac{L^2}{\rho^{2(K+M)}|\mathcal{J}|}\right). \quad \square$$

### I.3   Proof of auxiliary Lemmas

To complete the proof of Proposition 2, it remains to prove Lemma 18, Lemma 19, and Lemma 20. In this case, those upper bounds have been shown in [41]. We briefly re-state the proof idea.

**Lemma 18** (Upper bound of $P_L(v_L, e_L, \ell)$ (True pairs: $j = \pi_*(i)$, $sD_+(a,b) > 1$))**.**

$$P_L(v_L, e_L, \ell) \leq (1+o(1))2n^{v_L}(1+2L^2)^{e_L} f_{\widetilde{t},t}\mathbf{1}_{\{K+M|e_L+v_L\}}\mathbf{1}_{\{v_L+e_L\leq 2N\}}.$$

*Proof of Lemma 18.* Firstly, we define unlabeled union graph class $\widetilde{\mathcal{U}}(v_L, e_L, \ell)$ and set of labeled union graphs isomorphic to $\dot{U}_L \in \widetilde{\mathcal{U}}(v_L, e_L, \ell)$ as $H(\dot{U}_L)$.

$$P_L(v_L, e_L, \ell) = \sum_{\dot{U}_L \in \mathcal{U}_L(v_L, e_L, \ell)} w(\dot{U}_L) = \sum_{\dot{U}_L \in \widetilde{\mathcal{U}}_L(v_L, e_L, \ell)} w(\dot{U}_L)|H(\dot{U}_L)|.$$

The number of ways to label $\dot{U}_L$ is $|H(\dot{U}_L)| \leq \binom{n}{v_L}\frac{v_L!}{\text{aut}(\dot{U}_L)} \leq \frac{n^{v_L}}{\text{aut}(\dot{U}_L)}$, $\binom{n}{v_L}v_L! \leq n^{v_L}$. In addition, $w(\dot{U}_L) \leq \text{aut}(\dot{U}_L)$. This is true because every bulbs are exactly 2-decorated so for the union graph, the automorphism number coming from the orbits in this bulb is the same as the weights of this bulb. However, there can be other orbits outside the bulbs, for example, some bulbs are isomorphic so the vertices on the wires can be in the same orbit, thereby increasing the automorphism number.

From Lemma 7 in [41], we know that

$$|\widetilde{\mathcal{U}}_L(v_L, e_L, l)| \leq 2(1+2L^2)^{e_L} f_{\widetilde{t},t}\mathbf{1}_{\{K+M|e_L+v_L\}}\mathbf{1}_{\{v_L+e_L\leq 2N\}},$$

where $f_{\widetilde{t},t}$ counts the possible structures of chandeliers before merging edges to form a union graph and $\widetilde{t}$ is depending on $\ell$.

Combining these pieces together, we derived the upper bound:

$$P_L(v_L, e_L, \ell) \leq 2n^{v_L}(1+2L^2)^{e_L} f_{\widetilde{t},t}\mathbf{1}_{\{K+M|e_L+v_L\}}\mathbf{1}_{\{v_L+e_L\leq 2N\}}.$$

See Lemma 3 in [41] for more detailed discussion on $t, \widetilde{t}$ and $f_{\widetilde{t},t}$. $\quad\square$

**Lemma 19** (Upper bound of $P_M(v_M, e_M)$ (True pairs: $j = \pi_*(i)$, $sD_+(a,b) > 1$))**.**

$$P_M(v_M, e_M) \leq (11\beta n)^{v_M} R^{\frac{4e_M}{M}} \sigma_{\text{eff}}^{2v_M+2e_M} \gamma_2^{e_M} \mathbf{1}_{\{v_M \leq e_M \frac{4K+4M}{M}\}}.$$

*Proof.* We have $|\mathcal{N}_M|M \leq 2e_M$ because each connected component $\dot{\mathcal{G}}(i,a)$ contains at least $M$ edges that are 3 or 4-decorated and $\mathcal{N}_M$ is the number of neighbors $a$ in this set. This gives the constraint on $v_M \leq \frac{2e_M}{M}(2K+2M)$ and from the definition of $w(\cdot)$, $w(\dot{U}_M) \leq R^{2|\mathcal{N}_M|} \leq R^{\frac{4e_M}{M}}$. Define $\widetilde{\mathcal{U}}_M(v_M, e_M)$ as the unlabeled union graph class and $H(\dot{U}_M)$ as the set of labeled $\dot{U}_M$ for $\dot{U}_M \in \widetilde{\mathcal{U}}_M(v_M, e_M)$.

$$P_M(v_M, e_M) = \sum_{\dot{U}_M \in \mathcal{U}_M(v_M, e_M)} w(\dot{U}_M) \leq R^{\frac{4e_M}{M}} \sum_{\dot{U}_M \in \widetilde{\mathcal{U}}_M(v_M, e_M)} |H(\dot{U}_M)|. \quad (63)$$

We have $|H(\dot{U}_M)| \leq (n)^{v_M}$. There are at most $\beta^{v_M}$ rooted unlabeled undercoated trees, where $\beta = \frac{1}{(1+o(1))\alpha}$ [55] and at most $11^{v_M}$ decorations for each tree. Thus, $|\widetilde{\mathcal{U}}_M(v_M, e_M)| \leq (11\beta)^{v_M}$. Finally, we obtain that

$$P_M(v_M, e_M) \leq (11\beta n)^{v_M} R^{\frac{4e_M}{M}} \mathbf{1}_{\{v_M \leq (2K+2M)\frac{2e_M}{M}\}}. \quad \square$$

**Lemma 20** (Upper bound of $P_N(v_N, e_N, k)$ (True pairs: $j = \pi_*(i)$, $sD_+(a, b) > 1$))**.**

$$P_N(v_N, e_N, k) \leq n^{v_N}(11\beta)^{v_N}(11R^4(v_N + 1)^2)^{k+1}\mathbf{1}_{\{v_N \leq 2(K+M)(2k+2)\}}.$$

*Proof.* From Lemma 2 in [41], we know that $|\mathcal{N}_N| \leq 2k + 2$ and thus $w(\dot{U}_N) \leq R^{|\mathcal{N}_N|} \leq R^{4(K+1)}$. Briefly, this is because the excess is $k$ and whenever two branches tangle with each other, the excess of this graph increases by one. To maximally involving branches in this $k + 1$ times of branch tangles, we never re-tangle a branch after it has tangled with the other branch. In this way, we see that there are at most $2(k + 1)$ branches in $\dot{U}_N$. This immediately gives the condition of $v_N \leq (2k + 2)(M + K)$. Define $\widetilde{\mathcal{U}}_N(v_N, e_N, k)$ as the unlabeled union graph class and $H(\dot{U}_N)$ as the set of labeled $\dot{U}_N$ for $\dot{U}_N \in \widetilde{\mathcal{U}}_N(v_N, e_N, k)$.

$$P_N(v_N, e_N, k) = \sum_{\dot{U}_N \in \mathcal{U}_N(v_N, e_N, k)} w(\dot{U}_N) \leq R^{4(k+1)} \sum_{\dot{U}_N \in \widetilde{\mathcal{U}}_N(v_N, e_N, k)} |H(\dot{U}_N)|.$$

$\dot{U}_L$ consists of a tree with $v_N$ vertices and additional $k + 1$ edges connecting vertices on the tree. What's more, each edge can be associated with at most 11 possible decoration sets.

$$|\widetilde{\mathcal{U}}_N(v_N, e_N, k)| \leq \beta^{v_N}\binom{\binom{v_N+1}{2}}{k+1}11^{v_N+k+1} \leq (11\beta)^{v_N}(11(v_N + 1)^2)^{(k+1)}.$$

Therefore,

$$P_N(v_N, e_N, k) \leq n^{v_N}(11\beta)^{v_N}(11R^4(v_N + 1)^2)^{k+1}\mathbf{1}_{\{v_N \leq 2(K+M)(2k+2)\}}. \qquad \square$$

# J  Proof of Proposition 3

In this section, we show the Proposition 3 to complete the analysis on variance of similarity score. In this case, as $j \neq \pi_*(i)$, the decorated union graph structure is different from the case when $j = \pi_*(i)$ because the roots of $S_1$ and $S_2$ are different on the complete graph.

## J.1  Graph partition

The definitions in Section I still apply, except that we need to re-define the union graph partition.

- **(Union graph partition)** We first decompose $\dot{U}$ into three edge-disjoint subgraphs. Specifically, for any neighbor $a$ of $i$ in $\dot{U}$, consider the graph $\dot{U}$ with the edge $(i, a)$ removed and let $\dot{\mathcal{C}}(i, a)$ be the connected component that contains $a$. Denote $\dot{\mathcal{G}}(i, a)$ as the union of $\dot{\mathcal{C}}(i, a)$ and the edge $(i, a)$. Let

$$\mathcal{N}_T(i) = \{a : D_U((i, a)) \cap \{S_1, T_1\} \neq \emptyset, \dot{\mathcal{G}}(i, a) \text{ is a tree not containing } j\},$$
$$\mathcal{N}_N(i) = \{a : D_U((i, a)) \cap \{S_1, T_1\} \neq \emptyset\} \setminus \mathcal{N}_T(i).$$

  Symmetrically, let

$$\mathcal{N}_T(j) = \{a : D_U((j, a)) \cap \{S_2, T_2\} \neq \emptyset, \dot{\mathcal{G}}(j, a) \text{ is a tree not containing } i\},$$
$$\mathcal{N}_N(j) = \{a : D_U((j, a)) \cap \{S_2, T_2\} \neq \emptyset\} \setminus \mathcal{N}_T(j).$$

  Next, we further decompose $\dot{U}_T(i)$ into two edge-disjoint subtrees and similarly for $\dot{U}_T(j)$. In particular, define

$$\mathcal{N}_M(i) = \{a \in \mathcal{N}_T(i) : |\{e \in E(\dot{\mathcal{G}}(i, a)) : |D_e| \geq 3\}| \geq M\},$$
$$\mathcal{N}_L(i) = \mathcal{N}_T(i) \setminus \mathcal{N}_M(i).$$

  Then, we decompose $\dot{U}$ according to the node set $\mathcal{N}$ into two tree-parts on root $i$ (resp. $j$):

$$\dot{U}_L(i) := \bigcup_{a \in \mathcal{N}_L(i)} \dot{\mathcal{G}}(i, a), \quad \dot{U}_M(i) := \bigcup_{a \in \mathcal{N}_M(i)} \dot{\mathcal{G}}(i, a).$$

and the non-tree part

$$\dot{U}_N := \dot{U} \setminus (\dot{U}_L(i) \cup \dot{U}_M(i) \cup \dot{U}_L(j) \cup \dot{U}_M(j)). \tag{64}$$

For convenience, we denote

$$\dot{U}_L := \dot{U}_L(i) \cup \dot{U}_L(j), \quad \dot{U}_M := \dot{U}_M(i) \cup \dot{U}_M(j). \tag{65}$$

We again assign weights to each of these decorated union graph partitions.

$$w(\dot{U}_L) := w_{S_1}(\dot{U}_L(i))w_{T_1}(\dot{U}_L(i))w_{S_2}(\dot{U}_L(j))w_{T_2}(\dot{U}_L(j)), \tag{66}$$

$$w(\dot{U}_M) := w_{S_1}(\dot{U}_M(i))w_{T_1}(\dot{U}_M(i))w_{S_2}(\dot{U}_M(j))w_{T_2}(\dot{U}_M(j)), \tag{67}$$

$$w(\dot{U}_N) := w_{S_1}(\dot{U} \setminus \dot{U}_T(i))w_{T_1}(\dot{U} \setminus \dot{U}_T(i))w_{S_2}(\dot{U} \setminus \dot{U}_T(j))w_{T_2}(\dot{U} \setminus \dot{U}_T(j)). \tag{68}$$

## J.2   Proof of the Proposition

*Proof of Proposition 3.* In the case of $j \neq \pi_*(i)$, the excess $k$ of the union graph starts from $-2$. This is because the minimum number of edges of a decorated union graph for $j \neq \pi_*(i)$ with $v$ vertices except for $i$ and $j$ is $v$, when $S_1, T_1$ and $S_2, T_2$ form two disjoint trees rooted at $i$ and $j$ respectively. Therefore, $k \geq v - v(\dot{U}) = -2$.

The proof of Proposition 2 in Section I holds for $j \neq \pi_*(i)$ until (59), which holds with placing $k+1$ with $k+2$. This is because the decorated union graph can be viewed as two disjoint trees with $v+2$ vertices plus $k+2$ edges connecting vertices on these two trees. We have,

$$\mathrm{Var}[\Phi_{ij}\mathbf{1}_{\mathcal{H}}] \leq (1+o(1)) \sum_{k \geq -2}^{2N-k-2} \sum_{v=0} (sp \wedge sq)^{-2N+v+k+2} \sum_{\dot{U} \in \mathcal{W}_{ij}(v,k)} \rho^{e(K_{11})}\mathrm{aut}(H)\mathrm{aut}(I)$$
$$\times \sigma_{\mathrm{eff}}^{4N} \gamma_2^{e(K_{12})+e(K_{21})+e(K_{22})} 2^{k+2}.$$

We define $P_{ij}(v,k) := \sum_{\dot{U} \in \mathcal{W}_{ij}(v,k)} \rho^{e(K_{11})}\mathrm{aut}(H)\mathrm{aut}(I)\gamma_2^{e(K_{12})+e(K_{21})+e(K_{22})}$.

$$\mathrm{Var}[\Phi_{ij}\mathbf{1}_{\mathcal{H}}] \leq (1+o(1)) \sum_{k \geq -2}^{2N-k-2} \sum_{v=0} (sp \wedge sq)^{-2N+v+k+2}\sigma_{\mathrm{eff}}^{4N} 2^{k+2} P_{ij}(v,k)$$

**Case a:** $k = -2$.   We first consider the special case when $k = -2$. When $k = -2$, there are two disjoint trees in the decorated union graph. Then, it must be the case of $S_1 = T_1$, $S_2 = T_2$, $v = 2N$, and $H = I$. Therefore, $e(K_{12}) = e(K_{21}) = e(K_{22}) = 0$. We have,

$$P_{ij}(2N, -2) \leq \sum_{H \in \mathcal{T}} \mathrm{aut}(H)^2 |S_1(i) : S_1 \cong H||S_2(j) : S_2 \cong H| = |\mathcal{T}|n^{2N}.$$

Under this parameterization, $-2N + v + k + 2$ is also zero, we have

$$\mathrm{Var}[\Phi_{ij}\mathbf{1}_{\mathcal{H}}] \leq (1+o(1)) \left( |\mathcal{T}|n^{2N}\sigma_{\mathrm{eff}}^{4N} + \sum_{k > -2}^{2N-k-2} \sum_{v=0} (sp \wedge sq)^{-2N+v+k+2}\sigma_{\mathrm{eff}}^{4N} 2^{k+2} P_{ij}(v,k) \right).$$

It turns out that the summation over $k > -2$ would have the same order as the first special case.

**Case b:** $k > -2$.   We enumerate through three parts of the union graph. Recall that $e(K_{12}) + e(K_{21}) + e(K_{22}) \leq 2(e_L + e_M + e_N) = 4N - 2(v+k+2)$, therefore

$$P_{ij}(v,k) \leq \sum_{e_M} \sum_{v_L, v_M, v_N} \sum_{\dot{U} \in \mathcal{W}(v,k)} w(\dot{U}_L)w(\dot{U}_M)w(\dot{U}_N)\gamma_2^{e(K_{12})+e(K_{21})+e(K_{22})}$$
$$\leq \sum_{e_M} \gamma_2^{4N-2(v+k+2)} \sum_{v_L, v_M, v_N} P_L(v_L)P_M(v_M, e_M)P_N(v_N, k), \tag{69}$$

where $P_M(v_M, e_M)$ is defined as before, while we let $P_L(v_L)$ and $P_N(v_N, k)$ have no constraints on the number of 3 and 4-decorated edges for $\dot{U}_L$ and $\dot{U}_N$.

We can plug those upper bounds from Lemma 21, Lemma 22, and Lemma 23 into (69):

$$P_{ij}(v,k) \leq \sum_{e_M} R^{2\frac{e_M}{M}} n^v |\mathcal{T}| 4^L L^{2L \wedge (4K+2)} (6\beta)^{4(K+M)-2} \mathbf{1}_{\{e_M \leq 2N-(v+k+2)\}}$$

$$\times \sum_{v_L+v_M+v_N=v} v_M (11\beta)^{v_M} \beta (11\beta)^{v_N} (22R^4 (v_N+2)^2)^{k+2} \sigma_{\text{eff}}^{4N} \gamma_2^{4N-2(v+k+2)}$$

$$\times \mathbf{1}_{\{v_M \leq 2(K+M)\frac{2e_M}{M}\}} \mathbf{1}_{\{v_N \leq 4(K+M)(k+2)\}}.$$

Note that $v_M, v_N \leq v \leq 2N - k - 2 \leq 2N + 1$ as $k \geq -1$. Therefore $v_M (v_N+2)^{2(k+2)} \leq (2N+1)^{3(k+2)}$. Applying $e_M \leq 2N - (v+k+2)$, $v_N \leq 4(K+M)(k+2)$ and $v_M \leq 2(K+M)\frac{e_M}{M}$ we can get the following upper bound:

$$\frac{\text{Var}[\Phi_{ij}\mathbf{1}_{\mathcal{H}}]}{\mathbb{E}[\Phi_{i\pi_*(i)}\mathbf{1}_{\mathcal{H}}]^2} \leq (1+o(1))\frac{1}{|\mathcal{T}|\rho^{2N}} + (1+o(1))\frac{1}{|\mathcal{T}|\rho^{2N}} \left\{ 4^L L^{2L \wedge (4K+2)} (6\beta)^{4K+4M} \right.$$

$$\times \sum_{k \geq -1} \left( \frac{(11\beta)^{4(K+M)} 22R^4 (2N+1)^3}{n} \right)^{k+2}$$

$$\times \left. \sum_{v=0}^{2N-k-2} \left( \gamma_2^2 \frac{R^{\frac{2}{M}} (11\beta)^{\frac{4(K+M)}{M}}}{ns(p \wedge q)} \right)^{2N-(v+k+2)} \right\}.$$

From the first condition in (23), because $\gamma_2 < 2$, we have

$$\sum_{v=0}^{2N-k-2} \left( \gamma_2^2 \frac{R^{\frac{2}{M}} (11\beta)^{\frac{4(K+M)}{M}}}{ns(p \wedge q)} \right)^{2N-(v+k+2)} \leq 2.$$

From the second condition in (23), we have $\frac{(11\beta)^{4(K+M)} 22R^4 (2N+1)^3}{n} \leq \frac{1}{2}$, therefore,

$$\sum_{k \geq -1} \left( \frac{(11\beta)^{4(K+M)} 22R^4 (2N+1)^3}{n} \right)^{k+2} \leq 2 \left( \frac{(11\beta)^{4(K+M)} 22R^4 (2N+1)^3}{n} \right).$$

From the second condition in (23), we know that

$$4^L L^{2L \wedge (4K+2)} (6\beta)^{4K+4M} 2^2 \left( \frac{(11\beta)^{4(K+M)} 22R^4 (2N+1)^3}{n} \right) \leq \frac{1}{2}.$$

In summary,

$$\frac{\text{Var}[\Phi_{ij}\mathbf{1}_{\mathcal{H}}]}{\mathbb{E}[\Phi_{i\pi_*(i)}\mathbf{1}_{\mathcal{H}}]^2} = O\left( \frac{1}{|\mathcal{T}|\rho^{2N}} \right). \qquad \square$$

### J.3 Proof of auxiliary Lemmas

The following Lemma 21, Lemma 22, and Lemma 23, have been shown by Mao et al. [41]. We briefly re-state those results for completeness.

**Lemma 21** (Upper bound of $P_L(v_L)$ (Fake pairs: $j \neq \pi_*(i)$, $sD_+(a,b) > 1$)).

$$P_L(v_L) \leq n^{v_L} |\mathcal{T}| 4^L L^{2L \wedge (4K+2)} (6\beta)^{4(K+M)-2}.$$

*Proof.* Firstly, we introduce unlabeled union graph sets $\widetilde{\mathcal{U}}(v_L, e_L)$ and the set of labeled isomorphic members as $H(\dot{U}_L)$, for $\dot{U}_L \in \mathcal{U}(v_L, e_L, l)$. From definition (60),

$$P_L(v_L) = \sum_{\dot{U}_L \in \mathcal{U}_L(v_L)} w(\dot{U}_L) = \sum_{\dot{U}_L \in \widetilde{\mathcal{U}}_L(v_L)} w(\dot{U}_L) |H(\dot{U}_L)|.$$

As we have repeatedly seen, $|H(\dot{U}_L)| \leq \frac{n^{v_L}}{\text{aut}(\dot{U}_L)}$. From the Lemma 10 and Claim 5-(v) in [41] we have the following takeaways: (1) $|\widetilde{\mathcal{U}}_L(v_L)| \leq |\mathcal{T}| 4^L L^{2L \wedge (4K+2)} (6\beta)^{4(K+M)-2}$; (2) $w(\dot{U}_L) \leq \text{aut}(\dot{U}_L(i))\text{aut}(\dot{U}_L(j)) = \text{aut}(\dot{U}_L)$. Putting pieces together, we complete the proof. $\qquad \square$

**Lemma 22** (Upper bound of $P_M(v_M, e_M)$ (Fake pairs: $j \neq \pi_*(i)$)).
$$P_M(v_M, e_M) \leq v_M R^{\frac{2e_M}{M}} n^{v_M} (11\beta)^{v_M} \mathbf{1}_{\{e_M \leq 2N-(v+k+2)\}} \mathbf{1}_{\{v_M \leq 2(K+M)\frac{2e_M}{M}\}}.$$

*Proof.* Define $\widetilde{\mathcal{U}}_N(v_M, e_M)$ as the collection of unlabeled decorated union graphs and $|H(\dot{U}_M)|$ be the set of labeled isomorphic members for each $\dot{U}_M \in \widetilde{\mathcal{U}}_N(v_M, e_M)$.

In $\dot{U}_M$, there are $2e_M$ edges that are 3 or 4-decorated. Since for each branch connecting to the root, there are at least $M$ edges on it being at least 3-decorated, $|\mathcal{N}_M(i)| + |\mathcal{N}_M(j)| \leq 2\frac{e_M}{M}$. Thus, this directly gives $w(\dot{U}_M) \leq R^{\frac{1}{2} \times 2(|\mathcal{N}_M(i)|+|\mathcal{N}_M(j)|)} \leq R^{2\frac{e_M}{M}}$ and
$$P_M(v_M, e_M) = \sum_{\dot{U}_M \in \mathcal{U}_M(v_M, e_M)} w(\dot{U}_M) \leq R^{2\frac{e_M}{M}} \sum_{\dot{U}_M \in \widetilde{\mathcal{U}}_M(v_M, e_M)} |H(\dot{U}_M)|.$$

Since $\dot{U}_M(i)$ and $\dot{U}_M(j)$ are two vertex-disjoint trees with $v_M$ edges. There are at most $v_M \beta^{v_M}$ unlabeled non-decorated because we can allocate $v_M$ edges to 2 trees in at most $v_M$ ways and under each way the number of rooted unlabeled trees are bounded by $\beta^{v_M}$. There are at most $11^{v_M}$ ways of decorating each edge. Therefore, $|\widetilde{\mathcal{U}}_M(v_M, e_M)| \leq v_M (11\beta)^{v_M}$. In addition, the number of labeled isomorphic members $|H(\dot{U}_M)| \leq n^{v_M}$ and the number of 3-decorated edges plus two times of 4-decorated edges, $e_M$, is upper bounded by $4N - 2(v+k+2)$. Putting things together,
$$P_M(v_M, e_M) \leq v_M R^{\frac{2e_M}{M}} n^{v_M} (11\beta)^{v_M} \mathbf{1}_{\{e_M \leq 2N-(v+k+2)\}}.$$

Lastly, because each connected component $\dot{\mathcal{C}}(i,a)$ contains at most $2(K+M) - 1$ edges, for all $a \in \mathcal{N}_T(i)$ (same holds for $a \in \mathcal{N}_T(j)$). This is because there are at most four wires, each from $S_1, T_1, S_2, T_2$ go through vertex $a$ and every edge is at least 2-decorated. Therefore, $v_M \leq (K+M)\frac{2e_M}{M}$. □

**Lemma 23** (Upper bound of $P_N(v_N, k)$ (Fake pairs: $j \neq \pi_*(i)$)).
$$P_N(v_N, k) \leq n^{v_N} \beta (11\beta)^{v_N} (11R^4 (v_N+2)^2)^{k+2} \mathbf{1}_{\{v_N \leq 4(K+M)(k+2)\}}.$$

*Proof.* From the Lemma 9 in [41] we know that $|\mathcal{N}_N(i)| + |\mathcal{N}_N(j)| \leq 4(k+2)$. The intuition behind this bound is that $\dot{U}_N$ can be viewed as a bunch of branches (wires plus bulbs) coming from two different roots $i, j$ tangled together. Whenever two branches intersect with each other, the excess grow by one. Since the excess of decorated union graph is $k$ and the starting point is two separate trees (with excess $-2$), there must be $k + 2$ times of intersection between different branches. Each time of intersection involves at most four branches.

Thus $w(\dot{U}_N) \leq R^{|\mathcal{N}_N(i)|+|\mathcal{N}_N(j)|} \leq R^{4(k+2)}$. As usual, we define $\widetilde{\mathcal{U}}_N(v_N, k)$ as the collection of unlabeled decorated union graphs and $|H(\dot{U}_L)|$ be the set of labeled isomorphic members for each $\dot{U}_L \in \widetilde{\mathcal{U}}_N(v_N, k)$.
$$P_N(v_N, k) = \sum_{\dot{U}_N \in \mathcal{U}_N(v_N, k)} w(\dot{U}_N) \leq R^{4(k+2)} \sum_{\dot{U}_N \in \widetilde{\mathcal{U}}_N(v_N, k)} |H(\dot{U}_L)|.$$

By $|H(\dot{U}_N)| \leq \binom{n}{v_N} \frac{v_N!}{\mathrm{aut}(\dot{U}_N)} \leq n^{v_N}$, we have:
$$P_N(v_N, k) \leq R^{4(k+2)} |\widetilde{\mathcal{U}}_N(v_N, k)|.$$

In this part, the total number of unlabeled non-decorated graphs $\dot{U}_N$ with $v_N + 2$ vertices and excess $k$ is bounded by $\beta^{v_N+1}\binom{\binom{v_N+2}{2}}{k+1} \leq \beta^{v_N+1}(v_N+2)^{2(k+1)}$. This is because for a unlabeled connected graph with $v_N + 2$ vertices, we can construct a spanning tree with $v_N + 1$ vertices first and than add additional $k + 1$ edges connecting some of the $v_N + 2$ vertices. We can also bound the number of unlabeled $\dot{U}_N$ as $v_N \beta^{v_N} \binom{\binom{v_N+2}{2}}{k+2}$, which is constructing two trees rooted at $i$ and $j$, with a total of $v_N$ ways of vertex number allocation, and then adding an additional $k + 2$ edges. The latter bound is looser. Also, there are at most $11^{v_N+k+2}$ ways of decoration. Thus,
$$|\widetilde{\mathcal{U}}_N(v_N, k)| \leq \beta^{v_N+1}(v_N+2)^{2(k+1)} 11^{v_N+k+2} \leq \beta^{v_N+1}(v_N+2)^{2(k+2)} 11^{v_N+k+2}.$$
Lastly, $v_N$ is upper bounded by $(K+M)(|\mathcal{N}_N(i)| + |\mathcal{N}_N(j)|)$ by Claim 4 in [41]. □

## K   Proof of Proposition 4

In this section, we prove Proposition 4, which extends the Proposition 1 to the case when we can only perform almost exact community recovery with a single graph. We denote $\widehat{\boldsymbol{\sigma}} := (\widehat{\boldsymbol{\sigma}}_A, \widehat{\boldsymbol{\sigma}}_B)$, which is the combination of the community label estimates for both graphs.

*Proof of Proposition 4.* By definition of the similarity score and $\mathcal{H}$,

$$
\mathbb{E}[\Phi_{ij}^{\widehat{\boldsymbol{\sigma}}} \mathbf{1}_{\mathcal{H}}] = \sum_{H \in \mathcal{T}} \mathrm{aut}(H) \sum_{S(i) \cong H} \sum_{T(j) \cong H} \mathbb{E}[\overline{A}_S^{\widehat{\boldsymbol{\sigma}}_A} \overline{B}_T^{\widehat{\boldsymbol{\sigma}}_B} \mathbf{1}_{\mathcal{H}}]
$$

$$
= (1 + o(1)) \sum_{H \in \mathcal{T}} \mathrm{aut}(H) \sum_{S(i) \cong H} \sum_{T(j) \cong H} \mathbb{E}\left[\mathbb{E}[\overline{A}_S^{\widehat{\boldsymbol{\sigma}}_A} \overline{B}_T^{\widehat{\boldsymbol{\sigma}}_B} \mid \boldsymbol{\sigma}_*, \widehat{\boldsymbol{\sigma}}] \mathbf{1}_{\mathcal{H}}\right].
$$

Define $\mathcal{C}$ as the edge collection of the intersection graph of $S$ and $T$: $\mathcal{C} := E(S) \cap E(T)$.

$$
\mathbb{E}[\Phi_{ij}^{\widehat{\boldsymbol{\sigma}}} \mathbf{1}_{\mathcal{H}}]
$$

$$
\sim \sum_{H \in \mathcal{T}} \mathrm{aut}(H) \sum_{S(i) \cong H} \sum_{T(j) \cong H} \mathbb{E}\left[\mathbb{E}[\prod_{e \in \mathcal{C}} \overline{A}_e^{\widehat{\boldsymbol{\sigma}}_A} \overline{B}_e^{\widehat{\boldsymbol{\sigma}}_B} \prod_{e' \in E(S) \setminus \mathcal{C}} \overline{A}_{e'}^{\widehat{\boldsymbol{\sigma}}_A} \prod_{e'' \in E(T) \setminus \mathcal{C}} \overline{B}_{e'}^{\widehat{\boldsymbol{\sigma}}_B} \mid \boldsymbol{\sigma}_*, \widehat{\boldsymbol{\sigma}}] \mathbf{1}_{\mathcal{H}}\right]
$$

$$
\sim \sum_{H \in \mathcal{T}} \mathrm{aut}(H) \sum_{S(i) \cong H} \sum_{T(j) \cong H} \mathbb{E}\left[\prod_{e \in \mathcal{C}} \mathbb{E}[\overline{A}_e^{\widehat{\boldsymbol{\sigma}}_A} \overline{B}_e^{\widehat{\boldsymbol{\sigma}}_B} \mid \boldsymbol{\sigma}_*, \widehat{\boldsymbol{\sigma}}] \prod_{e' \in E(S) \setminus \mathcal{C}} \mathbb{E}[\overline{A}_{e'}^{\widehat{\boldsymbol{\sigma}}_A} \mid \boldsymbol{\sigma}_*, \widehat{\boldsymbol{\sigma}}] \right.
$$

$$
\times \left. \prod_{e'' \in E(T) \setminus \mathcal{C}} \mathbb{E}[\overline{B}_{e'}^{\widehat{\boldsymbol{\sigma}}_B} \mid \boldsymbol{\sigma}_*, \widehat{\boldsymbol{\sigma}}] \mathbf{1}_{\mathcal{H}}\right].
$$

We perform case studies for $\mathbb{E}[\Phi_{ij}^{\widehat{\boldsymbol{\sigma}}} \mathbf{1}_{\mathcal{H}}]$ based on the structure of $S, T$ as a union graph and then later sum each case up.

**(a) $S = T$, that is, all edges appear in pairs (this case is only possible for $i = j$).**   There are $(1 + o(1))n^N / \mathrm{aut}(H)$ labeled union graphs of $S_1$ and $T_1$ satisfying this condition. In this case, every edge is 2-decorated and it no longer matters whether $\widehat{\boldsymbol{\sigma}}$ gives a correct output or not, as we can show the following upper bound.

$$
\mathbb{E}[\Phi_{ij}^{\widehat{\boldsymbol{\sigma}}} \mathbf{1}_{\mathcal{H}}]^{(A)} = (1 + o(1)) \sum_{H \in \mathcal{T}} \mathrm{aut}(H) \frac{n^N}{\mathrm{aut}(H)} \mathbb{E}\left[\prod_{e \in \mathcal{C}} \mathbb{E}[\overline{A}_e^{\widehat{\boldsymbol{\sigma}}_A} \overline{B}_e^{\widehat{\boldsymbol{\sigma}}_B} \mid \boldsymbol{\sigma}_*, \widehat{\boldsymbol{\sigma}}] \mathbf{1}_{\mathcal{H}}\right]
$$

$$
\leq (1 + o(1))|\mathcal{T}| n^N \sum_{N_+} \mathbb{P}(\zeta = N_+) \sum_{N'} \mathbb{P}(\eta_{c+} = N_{c+}, \eta_{c-} = N_{c-} \mid \zeta = N_+)
$$

$$
\times (\rho_+ \sigma_+^2)^{N_{c+}} (\rho_- \sigma_-^2)^{N_{c-}} (\rho_+ \sigma_+^2 + \Delta^2)^{N_+ - N_{c+}} (\rho_- \sigma_-^2 + \Delta^2)^{N - N_+ - N_{c-}}
$$

$$
\leq (1 + o(1))|\mathcal{T}| n^N \sum_{N_+ = 0}^{N} \binom{N}{N-1} \frac{1}{2^N} (\rho \sigma_+^2 + \Delta^2)^{N_+} (\rho \sigma_-^2 + \Delta^2)^{N - N_+}
$$

$$
= (1 + o(1))|\mathcal{T}| n^N (\rho \sigma_{\mathrm{eff}}^2 + \Delta^2)^N,
$$

where $\zeta$ is the number of in-community edges out of the $N$ edges in $S$, $\eta_{c+}$ is the number of in-community edges that are centralized incorrectly, $\eta_{c-}$ is the number of cross-community edges that are centralized incorrectly, and $\Delta := |p - q|$. (i) The first equality holds by definition and counting cases. (ii) The second inequality holds because there are $N_{c+}(N_{c-})$ in(cross)-community edges centralized correctly, each of which contributes the same as $\mathbb{E}[\overline{A}_e \overline{B}_e \mid \boldsymbol{\sigma}_*] = (1 + o(1))\rho_+ \sigma_+^2 (\rho_- \sigma_-^2)$ (Lemma 13). For the remaining edges, $N_+ - N_{c+}$ ($N - N_+ - N_{c-}$) of them has $\mathbb{E}[\overline{A}_e^{\widehat{\boldsymbol{\sigma}}_A} \overline{B}_e^{\widehat{\boldsymbol{\sigma}}_B} \mid \boldsymbol{\sigma}_*, \widehat{\boldsymbol{\sigma}}] = (1 + o(1))(\rho_+ \sigma_+^2) + \mathbb{E}[\overline{A}_e^{\widehat{\boldsymbol{\sigma}}_A} \mid \boldsymbol{\sigma}_*, \widehat{\boldsymbol{\sigma}}] \mathbb{E}[\overline{B}_e^{\widehat{\boldsymbol{\sigma}}_B} \mid \boldsymbol{\sigma}_*, \widehat{\boldsymbol{\sigma}}] \leq (1 + o(1))(\rho_+ \sigma_+^2 + \Delta^2)$. (iii) The third inequality holds because $\Delta^2 > 0$, Lemma 9 gives the distribution of $\zeta$, and $\rho = (1 + \Theta(\frac{\log n}{n}))\rho_+, \rho = (1 + \Theta(\frac{\log n}{n}))\rho_-$. (iv) The last equality holds from the binomial theorem.

Observe that $\Delta^2 = (1 + \Theta(\frac{\log n}{n}))(\rho \sigma_{\mathrm{eff}}^2)$. Assume that $N = O(\log n)$,

$$
\mathbb{E}[\Phi_{ij}^{\widehat{\boldsymbol{\sigma}}} \mathbf{1}_{\mathcal{H}}]^{(A)} = (1 + o(1))|\mathcal{T}| n^N (\rho \sigma_{\mathrm{eff}})^N.
$$

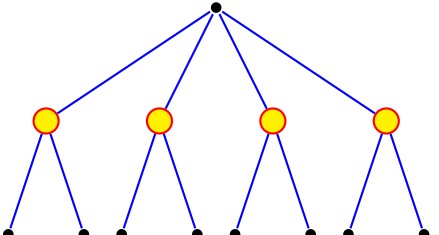 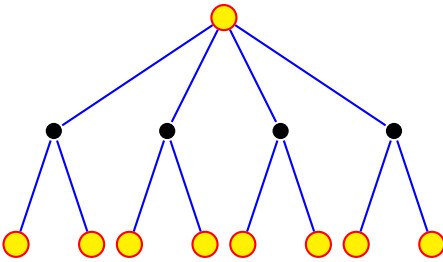

Figure 8: *Left*: One possible labeling such that all edges are centralized incorrectly, with the colored vertices indicating those that are labeled incorrectly and the black vertices indicating those that are labeled correctly. *Right*: Another possible labeling such that all edges are centralized incorrectly.

**(b) $S$ and $T$ have no common edges, that is, $E(S) \cap E(T) = \emptyset$.** We can use the trivial bound on the labeled $S$ and $T$ as $\frac{n^{2N}}{\text{aut}(H)^2}$.

$$\mathbb{E}[\Phi_{ij}^{\widehat{\sigma}} \mathbf{1}_{\mathcal{H}}]^{(B)}$$

$$\sim \sum_{H \in \mathcal{T}} \text{aut}(H) \sum_{S(i) \cong H} \sum_{T(j) \cong H: \mathcal{C} = \emptyset} \mathbb{E}\left[ \prod_{e' \in E(S)} \mathbb{E}[\overline{A}_{e'}^{\widehat{\sigma}_A} \mid \boldsymbol{\sigma}_*, \widehat{\boldsymbol{\sigma}}] \prod_{e'' \in E(T)} \mathbb{E}[\overline{B}_{e'}^{\widehat{\sigma}_B} \mid \boldsymbol{\sigma}_*, \widehat{\boldsymbol{\sigma}}] \mathbf{1}_{\mathcal{H}} \right]$$

$$\lesssim |\mathcal{T}| \frac{n^{2N}}{\text{aut}(H)} \mathbb{P}(E) \Delta^{2N}, \tag{70}$$

where $E$ denotes the event $\{\widehat{\boldsymbol{\sigma}} : \prod_{e \in E(S)} \mathbb{E}[\overline{A}_e^{\widehat{\sigma}_A}] \prod_{e \in E(T)} \mathbb{E}[\overline{B}_e^{\widehat{\sigma}_B}] = \Theta(\Delta^{e(S)+e(T)})\} \cap \mathcal{H}$, that is, every edge is centralized correctly and $\mathcal{H}$ happens. This inequality is true because conditioned on $\mathcal{H}$ and $\widehat{\boldsymbol{\sigma}}$ being correct, $\mathbb{E}[\overline{A}_e^{\widehat{\sigma}_A} \mid \boldsymbol{\sigma}_*, \widehat{\boldsymbol{\sigma}}] \mathbf{1}_{\mathcal{H}} = o(n^{-D_+(a,b,s,\varepsilon)})$, the upper bound of the probability that one vertex on this edge being labeled incorrectly.

We further denote $E'$ as $\{\widehat{\boldsymbol{\sigma}}_A : \prod_{e \in E(S)} \mathbb{E}[\overline{A}_e^{\widehat{\sigma}_A}] = \Theta(\Delta^{e(S)})\} \cap \mathcal{H}$. It is obvious that $\mathbb{P}(E) \leq \mathbb{P}(E')$. Then, we need to upper bound the probability of $\widehat{\boldsymbol{\sigma}}_A$ giving incorrect centralization for all edges in $S \cup T$.

We observe that *there are only two situations*, that is, no vertices in $S$ that has a neighbor with the same label correctness as itself. See Figure 8 for an illustration. We denote those two possible outcomes constraint on $S \cup T$ as $\widehat{\boldsymbol{\sigma}}_1$ and $\widehat{\boldsymbol{\sigma}}_2$. $\mathbb{P}(E') = \mathbb{P}(\{\widehat{\boldsymbol{\sigma}}_A = \widehat{\boldsymbol{\sigma}}_1\} \cap \mathcal{H}) + \mathbb{P}(\{\widehat{\boldsymbol{\sigma}}_A = \widehat{\boldsymbol{\sigma}}_2\} \cap \mathcal{H})$.

By the label correctness, we separate $V(S)$ into two disjoint sets: $V(S)_c^{\widehat{\sigma}}$ for the correctly labeled vertices and $V(S)_{ic}^{\widehat{\sigma}}$ for the incorrectly labeled vertices. The deterministic relationship between $\widehat{\boldsymbol{\sigma}}_1$ and $\widehat{\boldsymbol{\sigma}}_2$ is: $V(S)_{ic}^{\widehat{\sigma}_1} = V(S)_c^{\widehat{\sigma}_2}$. without loss of generality, we assume $|V(S)_c^{\widehat{\sigma}_1}| > |V(S)_{ic}^{\widehat{\sigma}_1}|$. Observe that event $\{\widehat{\boldsymbol{\sigma}}_A = \widehat{\boldsymbol{\sigma}}_1\}$ (resp. $\{\widehat{\boldsymbol{\sigma}}_A = \widehat{\boldsymbol{\sigma}}_2\}$) is equivalent as saying the set of vertices on odd (resp. even) levels are labeled incorrectly (falling in the bad vertex set $I_\varepsilon$ as defined in Section D).

Denote $p_{a,b,s,\varepsilon,\delta,V} := n^{-V \times D_+(a,b,s,\varepsilon)} + n^{-\varepsilon\delta(1-o(1))\log n}$. We can apply Lemma 8,

$$\mathbb{P}(\{\widehat{\boldsymbol{\sigma}}_A = \widehat{\boldsymbol{\sigma}}_1\} \cap \mathcal{H}) \leq (1 - p_{a,b,s,\varepsilon,\delta,|V(S)_c^{\widehat{\sigma}_1}|})(p_{a,b,s,\varepsilon,\delta,|V(S)_{ic}^{\widehat{\sigma}_1}|}),$$

$$\mathbb{P}(\{\widehat{\boldsymbol{\sigma}}_A = \widehat{\boldsymbol{\sigma}}_2\} \cap \mathcal{H}) \leq (1 - p_{a,b,s,\varepsilon,\delta,|V(S)_{ic}^{\widehat{\sigma}_1}|})(p_{a,b,s,\varepsilon,\delta,|V(S)_c^{\widehat{\sigma}_1}|}),$$

$$\mathbb{P}(E') \leq 2(1 - p_{a,b,s,\varepsilon,\delta,|V(S)_c^{\widehat{\sigma}_1}|})(p_{a,b,s,\varepsilon,\delta,|V(S)_{ic}^{\widehat{\sigma}_1}|}).$$

Consider each chandelier has $L$ branches and each has a $M$-wire (assume that $M = \Theta(\frac{\log n}{\log \log n})$ as in condition (3)). Even if we don't know the structure of bulbs, we have a coarse lower bound $|V(S)_{ic}^{\widehat{\sigma}_1}| = \Omega(\frac{\log n}{\log \log n})$, because at least half of the vertices on wires should be labeled incorrectly.

For some constant $c_1 > 0$ such that $|V(S)_{ic}^{\widehat{\sigma}_1}| \geq c_1 \frac{\log n}{\log \log n}$,

$$\mathbb{P}(E) \leq \mathbb{P}(E') \leq O(p_{a,b,s,\varepsilon,\delta,|V(S)_{ic}^{\widehat{\sigma}_1}|}) \leq O(n^{-D_+(a,b,s,\varepsilon)c_1\frac{\log n}{\log \log n}}).$$

By substituting $\mathbb{P}(E)$ and $\Delta^{2N}$ into (70), we have

$$\mathbb{E}[\Phi_{ij}^{\widehat{\sigma}}\mathbf{1}_{\mathcal{H}}]^{(B)} \leq O\left(|\mathcal{T}|\frac{n^{2N}}{\operatorname{aut}(H)}n^{-[D_+(a,b,s,\varepsilon)]c_1\frac{\log n}{\log\log n}}(\rho\sigma_{\text{eff}}^2)^{2N}(\nu)^{2N}\right).$$

Recall that $\mu = |\mathcal{T}|n^N(\rho\sigma_{\text{eff}}^2)^N$. Denote $\nu^2 = \frac{\Delta^2}{(\rho\sigma_{\text{eff}}^2)^2}(= (1+o(1))\frac{2(a-b)}{\rho(a+b)})$. For some constant $c_2 > 0$ such that $\rho\sigma_{\text{eff}}^2 \leq c_2\frac{\log n}{n}$,

$$\mathbb{E}[\Phi_{ij}^{\widehat{\sigma}}\mathbf{1}_{\mathcal{H}}]^{(B)} \leq O\left(\mu c_2^N\nu^{2N}(\frac{\log n}{n})^N n^N n^{-D_+(a,b,s,\varepsilon)c_1\frac{\log n}{\log\log n}}\right)$$

$$= O\left(\mu c_2^N\nu^{2N}\frac{(\log n)^N}{n^{c_1 D_+(a,b,s,\varepsilon)\frac{\log n}{\log\log n}}}\right).$$

For some constant $c_3 > 0$ such that $N = c_3\log n$, as in assumption (3),

$$= O\left(\mu(\frac{(c_2\nu^2\log n)^{c_3\log\log n}}{n^{c_1 D_+(a,b,s,\varepsilon)}})^{\frac{\log n}{\log\log n}}\right)$$

$$= o(\mu/n^2).$$

(c) $S$ **and** $T$ **have some common edges, that is,** $E(S) \cap E(T) \neq \emptyset$. There are at most $[n^{2N} - n^N(n-N)^N]/\operatorname{aut}(H)^2 = o(n^N)n^N/\operatorname{aut}(H)^2$ cases in the enumeration of $S$ and $T$.

$$\mathbb{E}[\Phi_{ij}^{\widehat{\sigma}}\mathbf{1}_{\mathcal{H}}]^{(C)} \lesssim \sum_{H\in\mathcal{T}}\operatorname{aut}(H)\sum_{M=1}^{N-1}\sum_{S(i)\cong H,T(j)\cong H:\mathcal{C}\neq\emptyset,X_d=M}(\rho\sigma_{\text{eff}}^2 + \Delta^2)^{N-M}\Delta^{2M}\mathbb{P}(E''),$$

where we denote by $E''$ the event

$$\{\widehat{\sigma} : \prod_{e'\in E(S)\backslash\mathcal{C}}\mathbb{E}[\overline{A}_{e'}^{\widehat{\sigma}_A}]\prod_{e''\in E(T)\backslash\mathcal{C}}\mathbb{E}[\overline{B}_{e'}^{\widehat{\sigma}_B}] = \Theta(\Delta)^{|E(S)\backslash\mathcal{C}|+|E(T)\backslash\mathcal{C}|}\}\cap\mathcal{H}.$$

Let $X_d$ denote the number of different edges between $S$ and $T$ under the true permutation. The structures of $S$ and $T$ such that $\{X_d = M\}$ happens, $\sum_{S(i)\cong H}\sum_{T(j)\cong H}\mathbf{1}_{\{X_d=M\}}$, is upper bounded by $\frac{n^N}{\operatorname{aut}(H)}\times\frac{n^M}{\operatorname{aut}(H)}$ as changing $M$ edges from $S$ to $T$ allows changing at most $M$ vertices for a tree. Then, we have

$$\mathbb{E}[\Phi_{ij}^{\widehat{\sigma}}\mathbf{1}_{\mathcal{H}}]^{(C)} \lesssim |\mathcal{T}|n^N\sum_{M=1}^{N-1}(\rho\sigma_{\text{eff}}^2 + \Delta^2)^{N-M}n^M\Delta^{2M}\mathbb{P}(E'')$$

$$\lesssim |\mathcal{T}|n^N\sum_{M=1}^{N-1}(\rho\sigma_{\text{eff}}^2)^{N-M}n^M(\rho\sigma_{\text{eff}}^2)^{2M}(\frac{2(a-b)}{\rho(a+b)})^{2M}\mathbb{P}(E'')$$

$$= (1+o(1))\mu\sum_{M=1}^{N-1}(n\rho\sigma_{\text{eff}}^2)^M\nu^{2M}\mathbb{P}(E''). \tag{71}$$

We observe the following:

$$\mathbb{P}(E'') \leq n^{-D_+(a,b,s,\varepsilon)(2M/2D)}.$$

This inequality holds because there are $2M$ edges centralized incorrectly (required from event $E''$) in $S$ and $T$ and each incorrect labeling of a vertex can lead to incorrect centralization on at most $D$ edges. Therefore, at least $2M/2D$ vertices should be wrongly labeled for $E''$ to happen.

Substitute $\mathbb{P}(E'')$ into (71). Assuming that $D = o(\frac{\log n}{\log\log n})$ (3),

$$\mathbb{E}[\Phi_{ij}^{\widehat{\sigma}}\mathbf{1}_{\mathcal{H}}]^{(C)} \lesssim \mu\sum_{M=1}^{N-1}(\frac{n\rho\sigma_{\text{eff}}^2\nu^2}{n^{\frac{1}{D}D_+(a,b,s,\varepsilon)}})^M \lesssim \mu\sum_{M=1}^{N-1}(\frac{\nu^2 c_1\log n}{n^{\frac{1}{D}D_+(a,b,s,\varepsilon)}})^M = o(\mu).$$

In summary,

$$\mathbb{E}[\Phi_{ij}^{\widehat{\sigma}}\mathbf{1}_{\mathcal{H}}] = \begin{cases} \mathbb{E}[\Phi_{ij}^{\widehat{\sigma}}\mathbf{1}_{\mathcal{H}}]^{(A)} + \mathbb{E}[\Phi_{ij}^{\widehat{\sigma}}\mathbf{1}_{\mathcal{H}}]^{(B)} + \mathbb{E}[\Phi_{ij}^{\widehat{\sigma}}\mathbf{1}_{\mathcal{H}}]^{(C)} = (1+o(1))\mu & \text{if } j = \pi_*(i), \\ \mathbb{E}[\Phi_{ij}^{\widehat{\sigma}}\mathbf{1}_{\mathcal{H}}]^{(B)} + \mathbb{E}[\Phi_{ij}^{\widehat{\sigma}}\mathbf{1}_{\mathcal{H}}]^{(C)} = o(\mu) & \text{if } j \neq \pi_*(i). \end{cases}$$

$\square$

**Remark 6.** *(Denser regime) If we are not restricting the sparse regime $p = \frac{a \log n}{n}$ and $q = \frac{b \log n}{n}$, we are interested in what general $p, q$ conditions are for Lemma 4 to hold. Assume that $p \vee q = O(n^{-c(n)})$, then the geometric series $\sum_{M=1}^{N-1} \frac{n \rho \sigma_{\mathrm{eff}}^2 \nu^2}{n^{\frac{1}{D} D_+(a,b,s,\varepsilon)}}$ converges if and only if $c(n) > 1 - \frac{D_+(a,b,s,\varepsilon)}{D}$.*

## L    Proof of Proposition 5

### L.1    Proof of the Proposition

In this section, we analyze the second moment of similarity score. We expect the variance of $\Phi_{ij}$ to be infinitely small in comparison with the squared expectation of true pair's similarity score.

*Proof of Proposition 5.*  Recall that $S_1, T_1$ are rooted on $i$ and $S_2, T_2$ are rooted on $j$.

$$
\mathrm{Var}[\Phi_{ij}^{\widehat{\sigma}} \mathbf{1}_{\mathcal{H}}] = \sum_{H,I \in \mathcal{T}} \mathrm{aut}(H)\mathrm{aut}(I) \sum_{S_1,S_2 \cong H, T_1,T_2 \cong I} \mathrm{Cov}(\overline{A}_{S_1}^{\widehat{\sigma}_A} \overline{B}_{S_2}^{\widehat{\sigma}_B} \mathbf{1}_{\mathcal{H}}, \overline{A}_{T_1}^{\widehat{\sigma}_A} \overline{B}_{T_2}^{\widehat{\sigma}_B} \mathbf{1}_{\mathcal{H}})
$$

$$
= \underbrace{\sum_{H,I \in \mathcal{T}} \mathrm{aut}(H)\mathrm{aut}(I) \sum_{S_1,S_2 \cong H, T_1,T_2 \cong I} \mathbb{E}[\overline{A}_{S_1}^{\widehat{\sigma}_A} \overline{B}_{S_2}^{\widehat{\sigma}_B} \overline{A}_{T_1}^{\widehat{\sigma}_A} \overline{B}_{T_2}^{\widehat{\sigma}_B} \mathbf{1}_{\mathcal{H}}]}_{V_1} - \tag{72}
$$

$$
\underbrace{\sum_{H,I \in \mathcal{T}} \mathrm{aut}(H)\mathrm{aut}(I) \sum_{S_1,S_2 \cong H, T_1,T_2 \cong I} \mathbb{E}[\overline{A}_{S_1}^{\widehat{\sigma}_A} \overline{B}_{S_2}^{\widehat{\sigma}_B} \mathbf{1}_{\mathcal{H}}]\mathbb{E}[\overline{A}_{T_1}^{\widehat{\sigma}_A} \overline{B}_{T_2}^{\widehat{\sigma}_B} \mathbf{1}_{\mathcal{H}}]}_{V_2}
$$

**(a) Analyzing $V_2$.**    We first give the upper bound of the latter part. When analyzing with correct centralization, we ignore this part as it is non-negative in the correct centralization case. However, in this case, it is possible to be negative because there can be odd number of edges occurring once and also being incorrectly centralized.

After factorizing $V_2$,

$$
V_2 = \left( \sum_{H \in \mathcal{T}} \mathrm{aut}(H) \sum_{S_1,S_2 \cong H} \mathbb{E}[\overline{A}_{S_1}^{\widehat{\sigma}_A} \overline{B}_{S_2}^{\widehat{\sigma}_B} \mathbf{1}_{\mathcal{H}}] \right) \left( \sum_{I \in \mathcal{T}} \mathrm{aut}(I) \sum_{T_1,T_2 \cong I} \mathbb{E}[\overline{A}_{T_1}^{\widehat{\sigma}_A} \overline{B}_{T_2}^{\widehat{\sigma}_B} \mathbf{1}_{\mathcal{H}}] \right),
$$

we can see that $V_2 \geq 0$, and thus we have $\mathrm{Var}[\Phi_{ij}^{\widehat{\sigma}} \mathbf{1}_{\mathcal{H}}] \leq V_1$ for $j = \pi_*(i)$.

**(b) Analyzing $V_{11}$.**    The main challenge here is that we do not have the condition that every edge occurs at least twice in the union graph as in the analysis of Regime I. However, we can put union graphs into two categories based on whether every edge is at least 2-decorated or not. We keep the notation of $\mathcal{W}_{ij}$ as the collection of decorated union graphs $\dot{U}$ that are at least 2-decorated and we decompose (72) as follows

$$
V_1 = \sum_{H,I \in \mathcal{T}} \mathrm{aut}(H)\mathrm{aut}(I) \sum_{S_1,S_2 \cong H, T_1,T_2 \cong I} \mathbb{E}[\overline{A}_{S_1}^{\widehat{\sigma}_A} \overline{B}_{S_2}^{\widehat{\sigma}_B} \overline{A}_{T_1}^{\widehat{\sigma}_A} \overline{B}_{T_2}^{\widehat{\sigma}_B} \mathbf{1}_{\mathcal{H}}]
$$

$$
= \underbrace{\sum_{\dot{U} \in \mathcal{W}_{ij}} (\mathrm{aut}(S_1)\mathrm{aut}(S_2)\mathrm{aut}(T_1)\mathrm{aut}(T_2))^{\frac{1}{2}} \mathbb{E}[\overline{A}_{S_1}^{\widehat{\sigma}_A} \overline{B}_{S_2}^{\widehat{\sigma}_B} \overline{A}_{T_1}^{\widehat{\sigma}_A} \overline{B}_{T_2}^{\widehat{\sigma}_B} \mathbf{1}_{\mathcal{H}}]}_{V_{11}} +
$$

$$
+ \underbrace{\sum_{\dot{U} \notin \mathcal{W}_{ij}} (\mathrm{aut}(S_1)\mathrm{aut}(S_2)\mathrm{aut}(T_1)\mathrm{aut}(T_2))^{\frac{1}{2}} \mathbb{E}[\overline{A}_{S_1}^{\widehat{\sigma}_A} \overline{B}_{S_2}^{\widehat{\sigma}_B} \overline{A}_{T_1}^{\widehat{\sigma}_A} \overline{B}_{T_2}^{\widehat{\sigma}_B} \mathbf{1}_{\mathcal{H}}]}_{V_{12}}.
$$

We first show that $V_{11}/\mu^2 = O\left( \frac{L^2}{\rho^2 ns(p \wedge q)} + \frac{L^2}{\rho^{2(K+M)}|\mathcal{J}|} \right)$.

The incorrect centralization affects the upper bound of moments as the following:

$$\mathbb{E}[\overline{A}_{S_1}^{\widehat{\sigma}_A}\overline{B}_{S_2}^{\widehat{\sigma}_B}\overline{A}_{T_1}^{\widehat{\sigma}_A}\overline{B}_{T_2}^{\widehat{\sigma}_B}\mathbf{1}_{\mathcal{H}}]$$

$$= \mathbb{E}\left[g_{\dot{U}}(\sigma_+,\sigma_-)\prod_{2\leq \ell+m\leq 4}\prod_{(u,v)\in K_{\ell m}}\sigma_{c(u,v)}^{-(\ell+m)}\mathbb{E}[\overline{A}_{uv}^{\widehat{\sigma}_A,\ell}\overline{B}_{uv}^{\widehat{\sigma}_B,m}\mid \sigma_*,\widehat{\sigma}]\mathbf{1}_{\mathcal{H}}\right]$$

$$= \mathbb{E}\left[g_{\dot{U}}(\sigma_+,\sigma_-)\prod_{2\leq \ell+m\leq 4}\beta_{l,m}^{e(K_{\ell m})}\mathbf{1}_{\mathcal{H}}\right]$$

$$\leq \rho^{e(K_{11})}(sp\wedge sq)^{-2N+e(U)}\sigma_{\text{eff}}^{4N}(1+O(\frac{\log n}{n}))^{e(U)}$$

$$\leq \rho^{e(K_{11})}(sp\wedge sq)^{-2N+e(U)}\sigma_{\text{eff}}^{4N}(1+O(\frac{\log n}{n}))^{4N}. \tag{73}$$

The first two equalities follow from definitions. The third inequality holds because of the upper bounds of $\beta_{\ell,m}^{e(K_{\ell m})}$ from Lemma 14 holds for all $\widehat{\sigma}$.

Then, the structures of the union graph, alone with the assigned weights are bounded the same as in Section I. This implies that $\frac{V_{11}}{\mathbb{E}[\Phi_{i\pi_*(i)}\mathbf{1}_{\mathcal{H}}]^2}\leq O\left(\frac{L^2}{\rho^2 ns(p\wedge q)}+\frac{L^2}{\rho^{2(K+M)}|\mathcal{J}|}\right)$ under the same condition as (22).

**(c) Analyzing $V_{12}$.** To conclude $\frac{\text{Var}[\Phi_{ij}^{\widehat{\sigma}}\mathbf{1}_{\mathcal{H}}]}{\mathbb{E}[\Phi_{i\pi_*(i)}\mathbf{1}_{\mathcal{H}}]^2}\leq \frac{V_{11}+V_{12}+V_2}{\mathbb{E}[\Phi_{i\pi_*(i)}\mathbf{1}_{\mathcal{H}}]^2}=O\left(\frac{L^2}{\rho^2 ns(p\wedge q)}+\frac{L^2}{\rho^{2(K+M)}|\mathcal{J}|}\right)$, it remains to show $V_{12}=o\left(\frac{L^2}{\rho^2 ns(p\wedge q)}+\frac{L^2}{\rho^{2(K+M)}|\mathcal{J}|}\right)$. Lemma 24 gives a even stronger result, because as assumed in Proposition 5, $L=o(\log n)$), giving $\frac{L^2}{\rho^2 ns(p\wedge q)}+\frac{L^2}{\rho^{2(K+M)}|\mathcal{J}|}\gg n^{-\varepsilon'}$ for all $\varepsilon'>0$. $\qquad\square$

**Lemma 24.** *Under the same conditions as Proposition 5, for some $\varepsilon'>0$,*

$$\frac{V_{12}}{\mathbb{E}[\Phi_{i\pi_*(i)}\mathbf{1}_{\mathcal{H}}]^2}=o(n^{-\varepsilon'}).$$

*Proof.* For the expectation inside $V_1$ part, it can be separated as eight sets $K_{\ell m}$:

$$\mathbb{E}[\overline{A}_{S_1}^{\widehat{\sigma}_A}\overline{B}_{S_2}^{\widehat{\sigma}_B}\overline{A}_{T_1}^{\widehat{\sigma}_A}\overline{B}_{T_2}^{\widehat{\sigma}_B}\mathbf{1}_{\mathcal{H}}]=\mathbb{E}\left[g_{\dot{U}}g_{\ddot{U}}^{-1}\mathbb{E}[\prod_{\ell\in[2],m\in[2],\ell+m\geq 1}\prod_{e\in K_{\ell m}}\overline{A}_e^{\widehat{\sigma}_A,\ell}\overline{B}_e^{\widehat{\sigma}_B,m}\mid \sigma_*,\widehat{\sigma}]\mathbf{1}_{\mathcal{H}}\right],$$

where $g_{\dot{U}}$ is the abbreviation for $g_{\dot{U}}(\sigma_+,\sigma_-)$.

Conditioned on $\sigma$ and $\widehat{\sigma}$, approximately centered edges are still independent with each other,

$$\mathbb{E}[\overline{A}_{S_1}^{\widehat{\sigma}_A}\overline{B}_{S_2}^{\widehat{\sigma}_B}\overline{A}_{T_1}^{\widehat{\sigma}_A}\overline{B}_{T_2}^{\widehat{\sigma}_B}\mathbf{1}_{\mathcal{H}}]=\mathbb{E}\left[g_{\dot{U}}g_{\ddot{U}}^{-1}\prod_{\ell\in[2],m\in[2],\ell+m\geq 1}\prod_{e\in K_{\ell m}}\mathbb{E}[\overline{A}_e^{\widehat{\sigma}_A,\ell}\overline{B}_e^{\widehat{\sigma}_B,m}\mid \sigma_*,\widehat{\sigma}]\mathbf{1}_{\mathcal{H}}\right],$$

Lemma 14 gives the upper bound over $\eta_{\ell,m}:=\sigma_{c(e)}^{-(\ell+m)}\mathbb{E}[\overline{A}_e^{\widehat{\sigma}_A,\ell}\overline{B}_e^{\widehat{\sigma}_B,m}\mid \sigma_*,\widehat{\sigma}]$ and thus by enumerating through the products, we have

$$g_{\ddot{U}}^{-1}\prod_{\ell\in[2],m\in[2],\ell+m\geq 1}\prod_{e\in K_{\ell m}}\mathbb{E}[\overline{A}_e^{\widehat{\sigma}_A,\ell}\overline{B}_e^{\widehat{\sigma}_B,m}\mid \sigma_*,\widehat{\sigma}]$$

$$\leq (1+\Theta(\frac{\log n}{n}))^{4N}\rho^{e(K_{11})}(sp\wedge sq)^{-\frac{1}{2}(e(K_{12})+e(K_{21}))-e(K_{22})}\mathbb{P}(\mathcal{E})\Delta^{z/2},$$

$$\leq (1+o(1))\rho^{e(K_{11})}(sp\wedge sq)^{v+k+1-2N}\mathbb{P}(\mathcal{E})(\frac{|a-b|}{a\wedge b})^{z/2},$$

where $\mathcal{E} := \{$Every 1-decorated edges are centralized incorrectly$\} \cap \mathcal{H}$ and $z$ is the number of 1-decorated edges, namely, $z := e(K_{01}) + e(K_{10})$. The second inequality holds because $-\frac{1}{2}(e(K_{12}) + e(K_{21})) - e(K_{22}) = (v + k + 1) - 2N - z/2$.

According to Lemma 8 and condition (24), we have that for any $\varepsilon > 0$,

$$\mathbb{P}(\mathcal{E}) \leq n^{-D_+(a,b,s,\varepsilon)\frac{z}{D}}.$$

Putting things together,

$$\mathbb{E}[\overline{A}_{S_1}^{\widehat{\sigma}_A}\overline{B}_{S_2}^{\widehat{\sigma}_B}\overline{A}_{T_1}^{\widehat{\sigma}_A}\overline{B}_{T_2}^{\widehat{\sigma}_B}\mathbf{1}_\mathcal{H}] \leq (1+o(1))n^{-D_+(a,b,s,\varepsilon)\frac{z}{D}}(sp \wedge sq)^{v+k+1-2N}\rho^{e(K_{11})}\mathbb{E}[g_{\dot{U}}(\sigma_+,\sigma_-) \mid \mathcal{H}].$$

It remains to calculate $\mathbb{E}[g_{\dot{U}}(\sigma_+,\sigma_-) \mid \mathcal{H}]$.

Similar as in the calculation in Section I, where we only consider $\dot{U}$ being at least 2-decorated. Here, we need to generalize it into the case when $\dot{U}$ has 1-decorated edges. $\dot{U}$ can be decomposed into a tree with an additional set of edges connecting vertices on the tree. We assume that there are $A_i$ $i$-decorated edges on the tree and $B_i$ $i$-decorated edges in the additional edge set of size $e(\dot{U}) - v(\dot{U}) + 1$, for $i \in [4]$. We apply the Corollary 1 and have that $\mathbb{P}(X^{(1,a)} = a_1, X^{(2,a)} = a_2, X^{(3,a)} = a_3, X^{(4,a)} = a_4, X^{(1,b)} = b_1, X^{(2,b)} = b_2, X^{(3,b)} = b_3, X^{(4,b)} = b_4 \mid \mathcal{H}) \leq (1+o(1))\binom{A_1}{a_1}\binom{A_2}{a_2}\binom{A_3}{a_3}\binom{A_4}{a_4}\frac{1}{2^{A_1+A_2+A_3+A_4}}$, where $X^{(i,a)}$ is the number of $i$-decorated in-community edges on the tree-part of $\dot{U}$ and $X^{(i,b)}$ is the number of $i$-decorated in-community edges among the additional edge set.

$A_i$ and $B_i$ are fixed but summed up to $d_i$ for each $\dot{U}$. $a_i$ ($b_i$) takes possible values from 0 to $A_i$ ($B_i$), for $i \in [4]$. The number of $i$-decorated in-community edges is $a_i + b_i$, and the number of $i$-decorated cross-community edges is $d_i - a_i - b_i = (A_i - a_i) + (B_i - b_i)$.

$$\mathbb{E}\left[g_{\dot{U}}(\sigma_+,\sigma_-) \mid \mathcal{H}\right]$$

$$\leq (1+o(1))\sum_{a_1}^{A_1}\sum_{a_2}^{A_2}\sum_{a_3}^{A_3}\sum_{a_4}^{A_4}\sum_{b_1}^{B_1}\sum_{b_2}^{B_2}\sum_{b_3}^{B_3}\sum_{b_4}^{B_4}\sigma_+^{(a_1+b_1)}\sigma_+^{2(a_2+b_2)}\sigma_-^{2(d_2-a_2-b_2)}\sigma_+^{3(a_3+b_3)}\sigma_-^{3(d_3-a_3-b_3)}$$

$$\times \sigma_+^{4(a_4+b_4)}\sigma_-^{4(d_4-a_4-b_4)}\binom{A_1}{a_1}\binom{A_2}{a_2}\binom{A_3}{a_3}\binom{A_4}{a_4}\frac{1}{2^{A_1+A_2+A_3+A_4}}$$

$$= (1+o(1))\left(\frac{\sigma_+ + \sigma_-}{2}\right)^{A_1}\left(\frac{\sigma_+^2 + \sigma_-^2}{2}\right)^{A_2}\left(\frac{\sigma_+^3 + \sigma_-^3}{2}\right)^{A_3}\left(\frac{\sigma_+^4 + \sigma_-^4}{2}\right)^{A_4}$$

$$\times \sum_{b_1=0}^{B_1}\sigma_+^{b_1}\sigma_-^{(B_1-b_1)}\sum_{b_2=0}^{B_2}\sigma_+^{2b_2}\sigma_-^{2(B_2-b_2)}\sum_{b_3=0}^{B_3}\sigma_+^{3b_3}\sigma_-^{3(B_3-b_3)}\sum_{b_4=0}^{B_4}\sigma_+^{4b_4}\sigma_-^{4(B_4-b_4)}$$

$$\leq (1+o(1))\left(\frac{\sigma_+ + \sigma_-}{2}\right)^{d_1}\left(\frac{\sigma_+^2 + \sigma_-^2}{2}\right)^{d_2}\left(\frac{\sigma_+^3 + \sigma_-^3}{2}\right)^{d_3}\left(\frac{\sigma_+^4 + \sigma_-^4}{2}\right)^{d_4}2^{k+1},$$

where the last inequality holds because the upper bound holds with multiplying a binomial coefficient $\binom{B_1}{b_1}\binom{B_2}{b_2}\binom{B_3}{b_3}\binom{B_4}{b_4}$ and that $\sum_i B_i = k+1$. By the definition of $\gamma_2$ and $\gamma_1$,

$$\mathbb{E}\left[g_{\dot{U}}(\sigma_+,\sigma_-) \mid \mathcal{H}\right] \leq (1+o(1))\left(\frac{\sigma_+ + \sigma_-}{2}\right)^{d_1}\sigma_{\text{eff}}^{2d_2+3d_3+4d_4}\gamma_1^{d_3}\gamma_2^{d_4}2^{k+1}.$$

Define $\gamma_0 := (\frac{\sigma_+ + \sigma_-}{2})/\sigma_{\text{eff}}$ and we can see that $\gamma_0 < 1$. So, we can upper bound $(\frac{\sigma_+ + \sigma_-}{2})^{d_1}$ as $\sigma_{\text{eff}}^{d_1}$. Because $\gamma_1 < \gamma_2$ and $d_3 + d_4 = e(K_{12}) + e(K_{21}) + e(K_{22})$,

$$\mathbb{E}\left[g_{\dot{U}}(\sigma_+,\sigma_-) \mid \mathcal{H}\right] \leq (1+o(1))\sigma_{\text{eff}}^{4N}\gamma_2^{e(K_{12})+e(K_{21})+e(K_{22})}2^{k+1}.$$

In summary,

$$\mathbb{E}[\overline{A}_{S_1}\overline{B}_{S_2}\overline{A}_{T_1}\overline{B}_{T_2}\mathbf{1}_\mathcal{H}] \leq (1+o(1))n^{-D_+(a,b,s,\varepsilon)\frac{z}{D}}(sp \wedge sq)^{v+k+1-2N}\rho^{e(K_{11})}$$

$$\times \sigma_{\text{eff}}^{4N}\gamma_2^{e(K_{12})+e(K_{21})+e(K_{22})}2^{k+1}. \tag{74}$$

After plugging the upper bound on cross-moments to the ratio, it remains to bound the number of different union graph structures and their corresponding weights. In parallel to $\mathcal{W}_{ij}$, we define $\mathcal{S}_{ij}$ as the collection of decorated union graphs that have at least one 1-decorated edge. $\mathcal{S}_{ij}(v, k)$ denotes those with $v + 1 + \mathbf{1}_{j \neq \pi_* i}$ vertices and excess $k$.

$$
\frac{V_{12}}{\mathbb{E}[\Phi_{i\pi_*(i)}\mathbf{1}_{\mathcal{H}}]^2}
\leq \frac{\sum_{v+k+1=0}^{4N} \sum_{\dot{U} \in \mathcal{S}_{ij}(v,k)} \operatorname{aut}(H)\operatorname{aut}(I)\rho^{e(K_{11})}(sp \wedge sq)^{v+k+1-2N}\gamma_2^{e(K_{12})+e(K_{21})+e(K_{22})}2^{k+1}}{(1+o(1))n^{2N}\rho^{2N}|\mathcal{T}|^2 n^{zD_+(a,b,s,\varepsilon)/D}}
$$

We define the $(\dot{U}_L, \dot{U}_M, \dot{U}_N)$ partition of decorated union graph as (44) in Section I. We define $\mathcal{U}_L(v_L, z, \ell)$ as the collection of $\dot{U}_L$ that has $v_L$ vertices, $\ell$ edges belonging to set $e(K_{11})$, and no more than $z$ 1-decorated edges. We define $\mathcal{U}_M(v_M)$ as the collection of $\dot{U}_M$ that has $v_M$ vertices. We define $\mathcal{U}_N(v_N, k)$ as the collection of $\dot{U}_N$ that has $v_N$ vertices and excess $k$. We also keep the notation of $\widetilde{\mathcal{U}}$ as the corresponding unlabeled decorated union graph sets. In addition,

$$
P_L(v_L, z, \ell) := \sum_{\dot{U}_L \in \mathcal{U}_L(v_L, z, \ell)} w(\dot{U}_L), \tag{75}
$$

$$
P_M(v_M) := \sum_{\dot{U}_L \in \mathcal{U}_L(v_M)} w(\dot{U}_M), \tag{76}
$$

$$
P_N(v_N, k) := \sum_{\dot{U}_L \in \mathcal{U}_L(v_N, k)} w(\dot{U}_N). \tag{77}
$$

From the above partition,

$$
\frac{V_{12}}{\mathbb{E}[\Phi_{i\pi_*(i)}\mathbf{1}_{\mathcal{H}}]^2}
\leq \frac{\sum_{v=N}^{4N} \sum_{k+1=0}^{4N-v} (sp \wedge sq)^{v+k+1-2N}2^{k+1} \sum_z \sum_{v_L, v_M, v_N} \sum_\ell \rho^\ell P_L(v_L, z, \ell)P_M(v_M)P_L(v_N, k)}{(1+o(1))n^{2N}\rho^{2N}|\mathcal{T}|^2 n^{zD_+(a,b,s,\varepsilon)/D}\gamma_2^{2(v+k+2)-4N}}.
$$

We show upper bounds for $P_L(v_L, z, \ell)$ in Lemma 25, which is

$$
\sum_{\ell=0}^{2N} \rho^\ell P_L(v_L, z, \ell) \leq (4N)^{2z+1}n^{v_L}L(4LM)^{6L}|\mathcal{T}|^2\rho^{2N}\rho^{-\frac{z}{2}}.
$$

The upper bounds for $P_M$ and $P_N$ trivially follows from Lemma 19 and Lemma 20, with a replacement of 11 to 15 as the possible decorations of each vertex increase by 4 for 1-decoration and a different maximum value of $e_M$, parameterized by $v, k, z$.

$$
P_M(v_M) \leq R^{\frac{2e_M}{M}}n^{v_M}(15\beta)^{(K+M)\frac{2e_M}{M}}\mathbf{1}_{\{e_M \leq 2N-(v+k+2)+z/2\}}
$$
$$
P_N(v_N, k) \leq n^{v_N}(15\beta)^{v_N}(15R^4(v_N+1)^2)^{k+1}\mathbf{1}_{\{v_N \leq 2(K+M)(2k+2)\}}.
$$

Putting all the pieces together, we have

$$
\frac{V_{12}}{\mathbb{E}[\Phi_{i\pi_*(i)}\mathbf{1}_{\mathcal{H}}]^2} \leq \sum_{k \geq -1} \left( \frac{30R^4(2N+1)^2(15\beta)^{4(K+M)}}{n} \right)^{k+1}
$$
$$
\times \sum_{v=N}^{4N-k-1} \left( \frac{R^{\frac{2}{M}}(15\beta)^{2\frac{K+M}{M}}}{ns(p \wedge q)}\gamma_2^2 \right)^{2N-v-k-1}
$$
$$
\times \sum_{z=1\vee(2(v+k+1)-4N)}^{4N} \left( \frac{(4N)^2 R^{\frac{1}{M}}(15\beta)^{\frac{K+M}{M}}}{\sqrt{\rho}n^{\frac{D_+(a,b,s,\varepsilon)}{D}}} \right)^z \times 2NL(4LM)^{6L}.
$$

From condition (24), $2NL(4LM)^{6L} \leq \log^3 n$. Also, since with (24) and $\gamma_2 < 2$,

$$\frac{R^{\frac{2}{M}}(15\beta)^{2\frac{K+M}{M}}}{ns(p \wedge q)}\gamma_2^2 \leq \frac{1}{2}, \quad \frac{15R^4(2N+1)^2(15\beta)^{4(K+M)}}{n} \leq \frac{1}{2}.$$

For $v \leq 2N - k - 1$, we have

$$\sum_{v=N}^{2N-k-1}\left(\frac{R^{\frac{2}{M}}(15\beta)^{2\frac{K+M}{M}}}{ns(p \wedge q)}\gamma_2^2\right)^{2N-v-k-1} \leq 2.$$

For the first summation, we always have

$$\sum_{k\geq-1}\left(\frac{30R^4(2N+1)^2(15\beta)^{4(K+M)}}{n}\right)^{k+1} \leq 2.$$

For the last summation with those additional terms, from condition (24), we have

$$\sum_{z=1}^{4N}\left(\frac{(4N)^2R^{\frac{1}{M}}(15\beta)^{\frac{K+M}{M}}}{\sqrt{\rho}n^{\frac{D_+(a,b,s,\varepsilon)}{D}}}\right)^z \times 2NL(4LM)^{6L} = o(n^{-\varepsilon'}),$$

for some $\varepsilon' > 0$ because $n^{D_+(a,b,s,\varepsilon)/D}$ is the only term being polynomial.

If $v > 2N - k - 1$, then we know that $z > 2(v + k + 1 - 2N)$. The summation over $v$ and $z$ together is upper bounded by

$$\sum_{v=2N-k}^{4N-k-1}\left(\frac{R^{\frac{2}{M}}(15\beta)^{2\frac{K+M}{M}}}{ns(p \wedge q)}\gamma_2^2 \times \frac{(4N)^2R^{\frac{1}{M}}(15\beta)^{\frac{K+M}{M}}}{\sqrt{\rho}n^{\frac{D_+(a,b,s,\varepsilon)}{D}}}\right)^{v+k+1-2N} 2NL(4LM)^{6L} = o(n^{-\varepsilon'}),$$

because, again $n^{D_+(a,b,s,\varepsilon)/D}$ is the only term being polynomial.

In summary, we have $\frac{V_{12}}{\mathbb{E}[\Phi_{i\pi_*(i)}\mathbf{1}_{\mathcal{H}}]^2} = o(n^{-\varepsilon'})$ for some $\varepsilon' > 0$, which completes the proof. $\square$

## L.2  Proof of auxiliary Lemmas

**Lemma 25.** *For true pairs,*

$$\sum_{\ell=0}^{2N}\rho^\ell P_L(v_L, z, \ell) \leq (4N)^{2z+1}n^{v_L}L(4LM)^{6L}|\mathcal{T}|^2\rho^{2N}\rho^{-\frac{z}{2}}.$$

*Proof.* We define the unlabeled union graph sets corresponding to $\mathcal{U}(v_L, z, \ell)$ as $\widetilde{\mathcal{U}}(v_L, z, \ell)$. From definition (60),

From the definition (75) and Claim 1,

$$\sum_{\ell=0}^{2N}\rho^\ell P_L(v_L, z, \ell) \leq \sum_{\ell=0}^{2N}\rho^\ell\sum_{\dot{U}_L \in \widetilde{\mathcal{U}}_L(v_L,z,\ell)}\frac{n^{v_L}w(\dot{U}_L)}{\text{aut}(\dot{U}_L)} \leq \sum_{\ell=0}^{2N}n^{v_L}\rho^\ell|\widetilde{\mathcal{U}}_L(v_L, z, \ell)|(2K)^z. \quad (78)$$

Recall that $e_L = \frac{1}{2}(e(K_{12} \cup K_{22} \cap \dot{U}_L) + e(K_{21} \cup K_{22} \cap \dot{U}_L))$. The total number of edges on $S_1, T_1, S_2$ and $T_2$ involved in $\dot{U}_L$ is $(2(v_L + e_L) - z)$ and thus the total number of bulbs on $S_1, T_1, S_2$ and $T_2$ involved in $\dot{U}_L$ is $b := \frac{2(v_L+e_L)-z}{K+M} < 4L$. Those $b$ bulbs can be partly or fully overlapped (namely, *tangled*) with another stay on their own. From the definition of $\dot{U}_L$ (44), it is impossible to have three or more bulbs tangling with each other. If two bulbs are tangling with each other, we put them into a pair. If a bunch of bulbs are all not tangling with any other bulbs, we pair them up arbitrarily. We denote $t_1$ as the number of pairs of bulbs that have decorations being a subset of $\{S_1, S_2\}$ or $\{T_1, T_2\}$. For all 2-decorated edges among these pairs, they are in $K_{11}$. Since $\dot{U}_L$ has at most $z$ 1-decorated edges, we have

$$\ell \geq t_1K - z/2. \quad (79)$$

Next, we introduce three types of bulbs. The first type is called *effective non-isomorphic* bulbs, which is a selection of bulbs that are not isomorphic to each other and always pair with a bulb that are not of the same type. The selection is not unique and we take the largest possible set of bulbs satisfying those rules as the set of effective non-isomorphic bulbs. Fixed the effective bulb set, for bulbs that are isomorphic to those effective non-isomorphic bulbs, we name them as *shadow effective bulbs*. For the remaining bulbs, we name them as *non-effective bulbs*. We have the following Claim 2:

$$\frac{t_1}{2} \le a \le L + \frac{t_1}{2}. \tag{80}$$

From definition, there is at most one non-effective bulb and at most one effective non-isomorphic bulb in each pair of bulbs, while two shadow bulbs can pair up.

We call those effective non-isomorphic bulbs as effective because when enumerating through the chandelier structures, we let them having the priority of taking any possible structure from $\mathcal{J}$ and serving the base of that pair. Shadow effective bulbs are named so because they mirror the structure of effective non-isomorphic bulbs and thus will not increase the union graph richness too much. For non-effective bulbs, we let them take any possible structures with the constrain that there are at most $z$ 1-decorated edges in $\dot{U}_L$.

We denote the number of effective non-isomorphic bulbs as $a$. For all pairs, we upper bound the number of different non-isomorphic tangled bulbs as following combinatorial factor

$$\binom{|\mathcal{J}|}{a}\binom{b}{a}\binom{2N}{z/2}(4N)^{\frac{z}{2}},$$

where $\binom{|\mathcal{J}|}{a}$ comes from the structure of $a$ effective non-isomorphic bulbs, $\binom{b}{a}$ is the upper bound of choosing $a$ effective non-isomorphic bulbs from $b$ bulbs, $\binom{2N}{z/2}$ is the upper bound on the selection of which edges on effective non-isomorphic bulbs and shadow effective bulbs are overlapped as there are at most $2K$ ($< 2N$) 2-decorated edges on bulbs if all bulbs are perfectly overlapping with one another, and $(4N)^{\frac{z}{2}}$ bounds the placement of those remaining 1-decorated edges from the non-effective bulbs as each of them has at most $4N$ possible vertices to attach to on the union graph.

Since wires cannot tangle with bulbs (otherwise, it is not a tree), we bound the possible structures separately. There can be at most 4 wires tangling with each other, from top to bottom. We apply a very loose bound even without using this fact, which is $(b-1)!M^{b-1}$. This is because there are at most $b$ wires and we determine the structure of wires on the union graph one by one. When the $t$-th wire comes in, it can determine which of the $t-1$ wires to tangle with and the length of overlap, from $0$ to $M$.

Putting together (79) and (80) with the combinatorial observations, we have

$$\rho^\ell |\widetilde{\mathcal{U}}_L(v_L, z, \ell)| \le \sum_{a=0}^{2L}\sum_{b=2a}^{4L}\sum_{t_1=0}^{a} \rho^{t_1 K - \frac{z}{2}} \mathbf{1}_{\{a \le L + \frac{t_1}{2}\}} |\mathcal{J}|^a \binom{b}{a}(2N)^{3z/2}(b-1)!M$$

$$\le |\mathcal{T}|(4N)^z \rho^{-\frac{z}{2}} \sum_{t_1=0}^{2L}(|\mathcal{J}|^{\frac{t_1}{2}}\rho^{t_1 K}) \sum_{a=0}^{2L}\sum_{b=2a}^{4L}\binom{b}{a}(b-1)!M^{b-1}$$

$$\le (4LM)^{6L}|\mathcal{T}|(4N)^z \rho^{-\frac{z}{2}} \sum_{t_1=0}^{2L}(|\mathcal{J}|\rho^{2K})^{\frac{t_1}{2}}$$

$$\le L(4LM)^{6L}|\mathcal{T}|^2 \rho^{2N}(4N)^z \rho^{-\frac{z}{2}}. \tag{81}$$

In the second inequality, we loose the upper bound of $t_1$ from $a$ to $2L$ and change the order of summation. In the third inequality, we bound the summation over $a$ and $b$. Lastly, $\sum_{t_1=0}^{2L}(|\mathcal{J}|\rho^{2K})^{\frac{t_1}{2}} \le L(|\mathcal{J}|\rho^{2K})^L = L|\mathcal{T}|\rho^{2N}$.

Plugging (81) back to (78), after a summation over $\ell$, we complete the proof. $\qquad \square$

**Claim 1.** *Assume that $j = \pi_*(i)$. For any arbitrary $\dot{U}_L \in \mathcal{U}_L(v_L, z, \ell)$, we have*

$$w(\dot{U}_L) \le \mathrm{aut}(\dot{U}_L)(2K)^z.$$

*Proof.* Denote all bulbs contained in $\dot{U}_L$ as $\mathcal{B}_1, \mathcal{B}_2, \ldots, \mathcal{B}_w$. (1) Some of them can be fully overlapped to form a 2-decorated bulbs in the union graph. (2) Some of them can be partly overlapped. (3) And the remaining of them are fully 1-decorated. There cannot be three or more bulbs overlapping with each other thanks to the definition of $\dot{U}_L$ (65).

Each bulb $\mathcal{B}_i$ contributes to the $w(\dot{U}_L)$ by $\text{aut}(\mathcal{B}_i)$ independently from definition (66) and (50). For any vertex on the bulbs, it will not be at the same orbit as any vertex on the wire, so studying the automorphism of the overlapped bulbs gives a lower bound on the automorphism of the whole decorated graph. Since each bulb occurs in at most one overlapped bulb, to prove the claim, it suffices to examine the relationship between weights and automorphism for each of the three cases aforementioned.

For $i, j \in [w]$, if bulbs $\mathcal{B}_i$ is partly overlapping with $\mathcal{B}_j$. From Corollary 2, $\sqrt{\text{aut}(\mathcal{B}_i)\text{aut}(\mathcal{B}_j)} \leq \text{aut}(\mathcal{B}_i \cup \mathcal{B}_j)(2K)^{|E(\mathcal{B}_i)\triangle E(\mathcal{B}_j)|}$. If $\mathcal{B}_i$ is fully overlapping with $\mathcal{B}_j$, then $\sqrt{\text{aut}(\mathcal{B}_i)\text{aut}(\mathcal{B}_j)} = \text{aut}(\mathcal{B}_i \cup \mathcal{B}_j)$. If $\mathcal{B}_i$ is fully 1-decorated, then $\sqrt{\text{aut}(\mathcal{B}_i)} \leq \text{aut}(\mathcal{B}_i)$.

Denote $I_1$ and $I_2$ as the collections of index pairs that the corresponding bulbs fall in case (1) or (2). Denote $I_3$ as the collection of indices corresponding to the bulbs falling in case (3). Therefore,

$$
\begin{aligned}
w(\dot{U}_L) &\leq \prod_{(i,j)\in I_1} \text{aut}(\mathcal{B}_i \cup \mathcal{B}_j)(2K)^{|E(\mathcal{B}_i)\triangle E(\mathcal{B}_j)|} \prod_{(i,j)\in I_2} \text{aut}(\mathcal{B}_i \cup \mathcal{B}_j) \prod_{i\in I_3} \text{aut}(\mathcal{B}_i) \\
&\leq \text{aut}(\dot{U}_L)(2K)^{\sum_{(i,j)\in I} |E(\mathcal{B}_i)\triangle E(\mathcal{B}_j)|}.
\end{aligned}
\tag{82}
$$

The union graph has at least $2 \times \sum_{(i,j)\in I} |E(\mathcal{B}_i)\triangle E(\mathcal{B}_j)|$ 1-decorated edges and we know that $\dot{U}_L$ has at most $z$ 1-decorated edges. Therefore, $\sum_{(i,j)\in I} |E(\mathcal{B}_i)\triangle E(\mathcal{B}_j)| \leq z$. Substituting this into (82) completes the proof. $\qquad \square$

**Claim 2.** *Let $a$ be the number of effective non-isomorphic bulbs and $t_1$ be the number of pairs of tangled bulbs that are decorated by a subset of either $\{S_1, S_2\}$ or $\{T_1, T_2\}$. We have*

$$
\frac{t_1}{2} \leq a \leq L + \frac{t_1}{2}.
\tag{83}
$$

*Proof.* For an arbitrary bulb, it occurs at most one time in $S_1, S_2, T_1, T_2$ each and they are paired up into two sets, $a \geq \frac{t_1}{2}$. Assume that there are $t'$ non-isomorphic bulbs among the $t_1$ pairs, they can all be assigned as effective non-isomorphic bulbs. Then, without loss of generality, $S_1, S_2$ have at most $L - \frac{t'}{2}$ bulbs unspecified. By assumption, they cannot pair up with each other, so every one bulb from $S_1, S_2$ remaining will pair up with another bulb from $T_1, T_2$. Therefore, the remaining bulbs have at most $L - \frac{t'}{2}$ effective non-isomorphic bulbs.

Together, we have $a \leq t' + (L - \frac{t'}{2}) \leq L + \frac{t_1}{2}$, as $t' \leq t_1$. $\qquad \square$

## M   Proof of Proposition 6

### M.1   Proof of the Proposition

*Proof of Proposition 6.* Recall that $S_1$ and $T_1$ are rooted on $i$, $S_2$ and $T_2$ are rooted on $j$. The minimum value of $k$ is $-2$ when the union graph consists of two disconnected trees, $S_1 \cup T_1$ and $S_2 \cup T_2$. We use the same notation for different parts of the variance as in Section L.

$$\mathrm{Var}[\Phi^{\widehat{\sigma}}_{ij}\mathbf{1}_\mathcal{H}] = \sum_{H,I\in\mathcal{T}}\mathrm{aut}(H)\mathrm{aut}(I)\sum_{S_1(i),S_2(i)\cong H, T_1(j),T_2(j)\cong I}\mathrm{Cov}(\overline{A}^{\widehat{\sigma}_A}_{S_1}\overline{B}^{\widehat{\sigma}_B}_{S_2}\mathbf{1}_\mathcal{H},\overline{A}^{\widehat{\sigma}_A}_{T_1}\overline{B}^{\widehat{\sigma}_B}_{T_2}\mathbf{1}_\mathcal{H})$$

$$= \underbrace{\sum_{U\in\mathcal{W}_{ij}}(\mathrm{aut}(S_1)\mathrm{aut}(S_2)\mathrm{aut}(T_1)\mathrm{aut}(T_2))^{\frac{1}{2}}\mathbb{E}[\overline{A}^{\widehat{\sigma}_A}_{S_1}\overline{B}^{\widehat{\sigma}_B}_{S_2}\overline{A}^{\widehat{\sigma}_A}_{T_1}\overline{B}^{\widehat{\sigma}_B}_{T_2}\mathbf{1}_\mathcal{H}]}_{V_{11}}$$

$$+ \underbrace{\sum_{U\notin\mathcal{W}_{ij}}(\mathrm{aut}(S_1)\mathrm{aut}(S_2)\mathrm{aut}(T_1)\mathrm{aut}(T_2))^{\frac{1}{2}}\mathbb{E}[\overline{A}^{\widehat{\sigma}_A}_{S_1}\overline{B}^{\widehat{\sigma}_B}_{S_2}\overline{A}^{\widehat{\sigma}_A}_{T_1}\overline{B}^{\widehat{\sigma}_B}_{T_2}\mathbf{1}_\mathcal{H}]}_{V_{12}} -$$

$$\underbrace{\sum_{H,I\in\mathcal{T}}\mathrm{aut}(H)\mathrm{aut}(I)\sum_{S_1,S_2\cong H,T_1,T_2\cong I}\mathbb{E}[\overline{A}^{\widehat{\sigma}_A}_{S_1}\overline{B}^{\widehat{\sigma}_B}_{S_2}\mathbf{1}_\mathcal{H}]\mathbb{E}[\overline{A}^{\widehat{\sigma}_A}_{T_1}\overline{B}^{\widehat{\sigma}_B}_{T_2}\mathbf{1}_\mathcal{H}]}_{V_2}. \tag{84}$$

Since $V_2 \geq 0$, it suffices to bound the first two summations. The same argument in Section L to bound the $V_{11}$ for true pairs works for this case with an additional fluctuation coming from using Lemma 14 to bound the cross moments rather than Lemma 13. We have

$$V_{11}/\mu^2 \leq (1+\Theta(\frac{\log n}{n}))^{4N}\frac{\mathrm{Var}[\Phi^{\widehat{\sigma}}_{ij}\mathbf{1}_\mathcal{H}]}{\mu^2}|_{sD_+(a,b)>1} = O(\frac{1}{|\mathcal{T}|\rho^{2N}})$$

under conditions (23). The additional fluctuation comes from the cross moment bounds.

Lemma 26 shows that $V_{12}/\mu^2 = o(\frac{1}{|\mathcal{T}|\rho^{2N}})$. In summary, we have $\frac{\mathrm{Var}[\Phi^{\widehat{\sigma}}_{ij}\mathbf{1}_\mathcal{H}]}{\mathbb{E}[\Phi_{i\pi_*(i)}\mathbf{1}_\mathcal{H}]^2} = O(\frac{1}{|\mathcal{T}|\rho^{2N}})$. □

**Lemma 26.** *Under the same conditions as Proposition 6,*

$$\frac{V_{12}}{\mu^2} = o(\frac{1}{|\mathcal{T}|\rho^{2N}}).$$

*Proof.* First, we apply the upper bound on $\mathbb{E}[\overline{A}^{\widehat{\sigma}_A}_{S_1}\overline{B}^{\widehat{\sigma}_B}_{S_2}\overline{A}^{\widehat{\sigma}_A}_{T_1}\overline{B}^{\widehat{\sigma}_B}_{T_2}\mathbf{1}_\mathcal{H}]$ as derived in (74), with replacing $k+1$ to $k+2$ everywhere as from definition $v$ is the number of vertices except for $i$ and $j$. When $j\neq\pi_*(i)$, the minimum value of $j$ starts from $-2$ when the union graph consists of two disjoint trees.

$$\mathbb{E}[\overline{A}^{\widehat{\sigma}_A}_{S_1}\overline{B}^{\widehat{\sigma}_B}_{S_2}\overline{A}^{\widehat{\sigma}_A}_{T_1}\overline{B}^{\widehat{\sigma}_B}_{T_2}\mathbf{1}_\mathcal{H}] \leq (1+o(1))n^{-D_+(a,b,s,\varepsilon)\frac{z}{D}}(sp\wedge sq)^{v+k+2-2N}\rho^{e(K_{11})}$$
$$\times \sigma^{4N}_{\mathrm{eff}}\gamma_2^{e(K_{12})+e(K_{21})+e(K_{22})}2^{k+2}. \tag{85}$$

After plugging the upper bound on cross-moments to the ratio, it remains to bound the number of different union graph structures and their corresponding weights.

$$\frac{V_{12}}{\mathbb{E}[\Phi_{i\pi_*(i)}\mathbf{1}_\mathcal{H}]^2}$$
$$\leq \frac{\sum_{v+k+2=0}^{4N}\sum_{\dot{U}\in\mathcal{S}_{ij}(v,k)}\mathrm{aut}(H)\mathrm{aut}(I)\rho^{e(K_{11})}(sp\wedge sq)^{v+k+2-2N}\gamma_2^{e(K_{12})+e(K_{21})+e(K_{22})}2^{k+2}}{(1+o(1))n^{2N}\rho^{2N}|\mathcal{T}|^2 n^{zD_+(a,b,s,\varepsilon)/D}}.$$

**(a) Case $k=-2$.** We first consider the special case when $k=-2$. When $k=-2$, there are two disjoint trees in the decorated union graph and all edges are decorated by a subset of either $\{S_1,T_1\}$ or $\{S_2,T_2\}$. Then, $e(K_{11})=e(K_{12})=e(K_{21})=e(K_{22})=0$. Also $v\geq 2N$,

$$V_{12} = \sigma^{4N}_{\mathrm{eff}}\sum_{\dot{U}\in\mathcal{S}_{ij}(v\geq 2N)}\mathrm{aut}(H)\mathrm{aut}(I)n^{-\frac{zD_+(a,b,s,\varepsilon)}{D}} \leq 2\sigma^{4N}_{\mathrm{eff}}F_{ij},$$

where

$$F_{ij} := \sum_{H\in\mathcal{T}}\sum_{I\in\mathcal{T}:\mathrm{aut}(I)\leq\mathrm{aut}(H)}\mathrm{aut}(H)^2\sum_{\dot{U}\in\mathcal{S}_{ij}(v\geq 2N,H,I)}n^{-\frac{zD_+(a,b,s,\varepsilon)}{D}},$$

and $\mathcal{S}_{ij}(v \geq 2N, H, I)$ is the collection of decorated union graphs that have at least $2N + 2$ vertices, excess $-2$, at least 1 edge 1-decorated, and that $S_1, S_2 \cong H, T_1, T_2 \cong I$.

For a specific decorated graph $\dot{U}$, we denote $t_1$ (resp. $t_2$) as the number of different edges vetween $S_1$ and $T_2$ (resp. $S_2$ and $T_2$). There are $z = 2(t_1 + t_2)$ 1-decorated edges and the remaining edges are in set $K_{02}$ or $K_{20}$. We write out $F_{ij}$ under the summation over $t_1$ and $t_2$.

$$
\begin{aligned}
F_{ij} &= \sum_{H \in \mathcal{T}} \sum_{S_1, S_2 \cong H} \sum_{\exists I \in \mathcal{T}, \mathrm{aut}(H) > \mathrm{aut}(I), T_1, T_2 \cong I} \sum_{t_1 = 0}^{N} \sum_{t_2 = 0}^{N} n^{-\frac{zD_+(a,b,s,\varepsilon)}{D}} \mathbf{1}_{\{t_1 + t_2 \geq 1\}} \\
&= \sum_{t_1 + t_2 \geq 1} \sum_{H \in \mathcal{T}} |\mathcal{S}(v \geq 2N, H, t_1, t_2)| n^{-\frac{zD_+(a,b,s,\varepsilon)}{D}},
\end{aligned}
$$

where $\mathcal{S}(v \geq 2N, H, t_1, t_2)$ collects all the possible decorated union graph that have $S_1, S_2 \cong H$, $T_1, T_2 \cong I$ for some $I \in \mathcal{T}$ such that $\mathrm{aut}(I) < \mathrm{aut}(H)$, and $S_1$ (resp. $S_2$) differ in $t_1$ (resp. $t_2$) edges with $T_1$ (resp. $T_2$).

Next, we bound $|\mathcal{S}(v \geq 2N, H, t_1, t_2)|$ by the following way: First, enumerate through all $S_1, S_2 \cong H$ on the complete graph with all possible structure, this gives $\frac{n^{2N}}{\mathrm{aut}(H)^2}$. Then, we choose which edges that are overlapped with $T_1$ on $S_1$ (resp. overlapped with $T_2$ on $S_2$). This is at most $\binom{N}{t_1}\binom{N}{t_2} \leq N^{t_1+t_2}$. After this, we draw $t_1 + t_2$ new vertices for $T_1$ and $T_2$ and allow them arbitrarily connecting edges among those $N$ vertices on its chandelier, which is upper bounded by $\binom{N}{2}^{t_1+t_2}$ (ignoring the constraint that $T_1 \cong T_2 \cong I$ for some chandelier $I$ having less automorphism number than $H$). Altogether,

$$
|\mathcal{S}_{ij}(v \geq 2N, H, I)| \leq \frac{n^{2N}}{\mathrm{aut}(H)^2} \left( \frac{N^3}{n^{2D_+(a,b,s,\varepsilon)/D}} \right)^{t_1+t_2}.
$$

Therefore,

$$
F_{ij} = n^{2N} |\mathcal{T}| \sum_{t_1 + t_2 \geq 1} \left( \frac{N^3}{n^{2D_+(a,b,s,\varepsilon)/D}} \right)^{t_1+t_2}.
$$

As assumed in Proposition 6, $N = \Theta(\log n)$ and $D = o(\frac{\log n}{\log \log n})$. Therefore,

$$
V_{12}/\mu^2 = o(\frac{1}{|\mathcal{T}|\rho^{2N}}).
$$

**(b) Case $k > -2$.** In general, we define $\dot{U}_L, \dot{U}_M$, and $\dot{U}_N$ partition the same as (64) and (65). We also define the weights of each part the same as (66), (67), and (68). We define $P_L(v_L, z) = \sum_{\dot{U} \in \mathcal{U}_L(v_L, z)} w(\dot{U})$. The definition of $P_M(v_M)$ and $P_N(v_N, k)$ follow. All union graph class should not have more than $z$ 1-decorated edges, but specifically we only need this constraint for $\dot{U}_L$.

Note that $e(K_{12}) + e(K_{21}) + e(K_{22}) \leq 4N - 2(v + k + 1) + z$,

$$
\begin{aligned}
V_{12}|_{k>-2} &\leq \sigma_{\mathrm{eff}}^{4N} \sum_{v=N}^{4N} \sum_{k+2=1}^{4N-v} \sum_{z=1}^{4N} \gamma_2^{4N-2(v+k+2)} 2^{k+2} (sp \wedge sq)^{v-2N+k+2} \\
&\quad \times \sum_{v_L, v_M, v_N} P_L(v_L, z) P_M(v_M) P_L(v_N, k) \left( \frac{\gamma_2}{n^{D_+(a,b,s,\varepsilon)/D}} \right)^z.
\end{aligned}
$$

We show the upper bound for $\dot{U}_L$ part in Lemma 27. The upper bound for $P_M$ and $P_N$ trivially follows from Lemma 22 and Lemma 23 as they do not use any assumption on edges are all at least 2-decorated, except for the number 11, the possible ways of decoration. So, we change 11 to 15 and then every thing follows. When $z \neq 0$, $e_M$ as defined before in an arbitrary $\dot{U}_M$ has maximum value $2N - (v + k + 2) - z/2$ (same holds for $e_N, e_L$ but we do not need to use them in our bound). In

summary,

$$P_L(v_L, z) \le n^{v_L} |\mathcal{T}| \beta^{4K} (4LM)^{4L} (4L)! (2\beta)^{4(K+M)} \left( (4N)^2 \beta^{\frac{K}{K+M}} \right)^z$$

$$P_M(v_M) \le R^{\frac{2e_M}{M}} n^{v_M} (15\beta)^{(K+M)\frac{2e_M}{M}} \mathbf{1}_{\{e_M \le 2N-(v+k+2)+z/2\}}$$

$$\le n^{v_M} \left( R^{\frac{2}{M}} (15\beta)^{\frac{2(K+M)}{M}} \right)^{2N-v-k-2} \left( R^{\frac{1}{M}} (15\beta)^{\frac{K+M}{M}} \right)^z$$

$$P_N(v_N, k) \le n^{v_N} \beta (15\beta)^{(K+M)2(k+2)} (15R^4(v_N+2)^2)^{k+2}$$

$$\le n^{v_N} \beta \left( (15\beta)^{2(K+M)} 15R^4 (4N+1)^2 \right)^{k+2}.$$

The last inequality holds because $v_N \le v \le 4N - (k+2)$ and $k \ge -1$.

Putting every pieces together,

$$\frac{V_{12}|_{k>-2}}{\mu^2} \le \sum_{k \ge -1} \left( \frac{(15\beta)^{2(K+M)} 30R^4(4n+1)^2}{n} \right)^{k+2} \sum_{v=N}^{4N-k-2} \left( \frac{\gamma_2^2 (15\beta)^{2\frac{K+M}{M}} R^{\frac{2}{M}}}{ns(p \wedge q)} \right)^{2N-v-k-2}$$

$$\times \sum_{z=1 \vee (2N-v-k-2)}^{4N} \left( \frac{\gamma_2 (4N)^2 R^{\frac{1}{M}} (15\beta)^{\frac{K+M}{M}} \beta^{\frac{K}{K+M}}}{n^{\frac{D_+(a,b,s,\varepsilon)}{D}}} \right)^z$$

$$\times \frac{\beta^{4K+1} (2\beta)^{4(K+M)} (4LM)^{4L} (4L)!}{|\mathcal{T}| \rho^{2N}}.$$

From the second condition in (25), we first look at the summation of $v$ from $N$ to $2N-k-2$,

$$\sum_{z=1}^{4N} \left( \frac{\gamma_2 (4N)^2 R^{\frac{1}{M}} (15\beta)^{\frac{K+M}{M}} \beta^{\frac{K}{K+M}}}{n^{\frac{D_+(a,b,s,\varepsilon)}{D}}} \right)^z = o(1).$$

From the first condition in (25),

$$\sum_{v=N}^{2N-k-2} \left( \frac{\gamma_2^2 (15\beta)^{2\frac{K+M}{M}} R^{\frac{2}{M}}}{ns(p \wedge q)} \right)^{2N-v-k-2} \le 2.$$

When $v > 2N-k-2$, the power $2N-v-k-2 < 0$. Observe that $z \ge 2(v+k+2) - 4N = z - e(K_{12}) - e(K_{21}) - 2e(K_{22})$, we have the product of two summations upper bounded by

$$\sum_{v=2N-k-1}^{4N-k-2} \left( \frac{ns(p \wedge q)}{\gamma_2^2 (15\beta)^{2\frac{K+M}{M}} R^{\frac{2}{M}}} \times \frac{\gamma_2 (4N)^2 R^{\frac{1}{M}} (15\beta)^{\frac{K+M}{M}} \beta^{\frac{K}{K+M}}}{n^{\frac{D_+(a,b,s,\varepsilon)}{D}}} \right)^{v+k+2-2N}.$$

This is clearly $o(1)$ because $N = \Theta(\log n)$ and from the first and second condition (25), $n^{D_+(a,b,s,\varepsilon)/D}$ is the only term being $\log^{\omega(1)} n$.

From the third condition in (25),

$$\sum_{k \ge -1} \left( \frac{(15\beta)^{2(K+M)} 30R^4(4n+1)^2}{n} \right)^{k+2} \beta^{4K+1} (2\beta)^{4(K+M)} (4LM)^{4L} (4L)!$$

$$\le 2 \left( \frac{(15\beta)^{2(K+M)} 30R^4(4n+1)^2}{n} \right) \beta^{4K+1} (2\beta)^{4(K+M)} (4LM)^{4L} (4L)! \le 1.$$

Therefore, $V_{12}/\mu^2 = o(\frac{1}{|\mathcal{T}|\rho^{2N}})$. $\qquad\qquad\qquad\qquad\qquad\qquad\qquad\qquad \square$

## M.2   Proof of auxiliary Lemmas

**Lemma 27.** *For $j \ne \pi_*(i)$,*

$$P_L(v_L, z) \le n^{v_L} |\mathcal{T}| \beta^{4K} (4LM)^{4L} (4L)! (2\beta)^{4(K+M)} \left( (4N)^2 \beta^{\frac{K}{K+M}} \right)^z.$$

*Proof.* We define the unlabeled union graph sets corresponding to $\mathcal{U}(v_L, z)$ as $\widetilde{\mathcal{U}}(v_L, z)$. From the definition (75) and Claim 3,

$$P_L(v_L, z) \leq \sum_{\dot{U}_L \in \widetilde{\mathcal{U}}_L(v_L, z)} \frac{n^{v_L} w(\dot{U}_L)}{\mathrm{aut}(\dot{U}_L)} \leq (2K)^z n^{v_L} |\widetilde{\mathcal{U}}_L(v_L, z)|. \tag{86}$$

Recall that $\dot{U}_L$ consists of two disjoint trees, one rooted at $i$ and the other one rooted at $j$. We consider the branches of chandeliers without overlapping with each other.

Then, we specify two categories of branches. A branch is called an invader if it is rooted at $i$ (resp. $j$) but in $\dot{U}_L(j)$ (resp. $\dot{U}_L(i)$). A branch that is not an invader is called a residence. We observe that there are at most $L + \lfloor \frac{z}{K+M} \rfloor + 4$ effective non-isomorphic bulbs among residents (defined in Section L, the proof of Proposition 5) because of the followings: 1) If branches are perfectly matched and overlapped, there are at most $L$ pairs of them rooted at $i$ and another $L$ pairs rooted at $j$. We define effective non-isomorphic bulbs the same as in Lemma 25. Here we have $L$ effective non-isomorphic bulbs because $S_1 \cong S_2, T_1 \cong T_2$. 2) $\lfloor \frac{z}{K+M} \rfloor$ is the maximum number of fully 1-decorated branches in allowed $\dot{U}_L$, and 3) There are at most 4 invading branches, each of which can at most fully overlapping with one resident bulb, due to the fact that there cannot be two branches on the same chandelier passing through the same vertex.

The remaining is to bound $|\widetilde{U}_L(v_L, z)|$. We observe that there are at most $4(K + M - 1)$ edges from invading branches, which might be attaching to at most $4(K + M - 1)$ resident branches. This is because an invader rooted at $j$ may only have its bulb overlapping with $\dot{U}_L(i)$. In this case, one invader can overlap with multiple resident branches in $\dot{U}_L(i)$.

$$|\widetilde{U}_L(v_L, z)| \leq \binom{|\mathcal{J}|}{L + \lfloor \frac{z}{K+M} \rfloor + 4} \binom{2N}{z/2} (4N)^{\frac{z}{2}} (4LM)^{4L} (2\beta)^{4(K+M-1)} (4L)!, \tag{87}$$

where $\binom{|\mathcal{J}|}{L + \lfloor \frac{z}{K+M} \rfloor + 4}$ is the structures of all resident branches, $\binom{2N}{z/2}$ is the upper bound of choosing which edges to be not overlapped on bulbs assume starting from perfect overlapped bulbs, $(4N)^{\frac{z}{2}}$ is the bound for placing the remaining $z/2$ 1-decorated vertices, $(4LM)^{4L}$ is a trivial bound on how resident branches have their wires tangling with each other, $(\beta)^{4(K+M-1)}$ is the structure of invading edges, and lastly $2^{(K+M-1)}(4L)!$ bounds the different interactions between invading edges and the resident branches. To understand the quantity $2^{(K+M-1)}(4L)!$, this comes from the fact that each invading edge connected to the root can choose one out of at most $4L$ resident branch to attach, and that the following invading edges can choose to stay overlapping with the current resident branch or leave.

Plugging (87) back into (86) with basic binomial bounds and $|\mathcal{J}| \leq \beta^K$, we complete the proof. $\square$

**Claim 3.** *Assume that $j \neq \pi_*(i)$. For any $\dot{U}_L \in \mathcal{U}_L(v_L, z)$,*

$$w(\dot{U}_L) \leq \mathrm{aut}(\dot{U}_L)(2K)^z. \tag{88}$$

*Proof.* Denote all bulbs contained in $\dot{U}_L(i)$ (resp., $j$) and are attached to wires rooted at $i$ (resp., $j$) as $\mathcal{B}_1, \mathcal{B}_2, \ldots, \mathcal{B}_w$. Denote all bulbs contained in $\dot{U}_L(i)$ (resp., $j$) and are attached to wires rooted at $j$ (resp., $i$) as $T_1, T_2, \ldots, T_m$. For those branches rooted at $i$ (resp., $j$) but connect to $j$ (resp., $i$), although they can have their bulbs in $\dot{U}_L(j)$ (resp., $\dot{U}_L(i)$), they contribute to the weight of non-tree part $\dot{U}_N$ from definitions (66) and (68).

For an arbitrary bulb $\mathcal{B}_t$ in $\dot{U}_L(i) \cup \dot{U}_L(j)$, we discuss the following three cases. Without loss of generality, we assume that $\mathcal{B}_t$ is attached to a wire rooted at $i$.

Firstly, if there exists another bulb $\mathcal{B}_r$ such that $\mathcal{B}_r$ and $\mathcal{B}_t$ be two bulbs with wires rooted both at $i$ and overlapping with each other. Then, from Corollary 2, we have that $\sqrt{\mathrm{aut}(\mathcal{B}_r)\mathrm{aut}(\mathcal{B}_t)} \leq \mathrm{aut}(\mathcal{B}_r \cup \mathcal{B}_t)(2K)^{\frac{E(\mathcal{B}_r) \triangle E(\mathcal{B}_t)}{2}}$, because each bulb has size $K$ and the difference between two edge set is at most $K$. Secondly, if $\mathcal{B}_r$ is full 1-decorated, then it contributes to $\sqrt{\mathrm{aut}(\mathcal{B}_r)}$ to $w(\dot{U}_L)$ and $\mathrm{aut}(\mathcal{B}_t)$ to $\mathrm{aut}(\dot{U}_L)$. Thirdly, assume that there is another bulb $\mathcal{B}_t$ attached to a wire rooted at $j$ partly

overlapping with $\mathcal{B}_t$, then 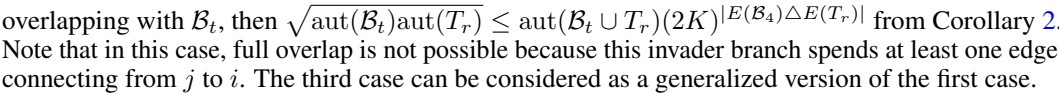 from Corollary 2. Note that in this case, full overlap is not possible because this invader branch spends at least one edge connecting from $j$ to $i$. The third case can be considered as a generalized version of the first case.

By a product over all overlapping bulbs, we have $w(\dot{U}_L) \leq \mathrm{aut}(\dot{U}_L)(2K)^z$ because the total number of edges in the difference sets of overlapping bulbs are upper bounded by $z$, the number of 1-decorated edges, and the automorphism number of $\dot{U}_L$ is greater than the product of automorphism number of all bulbs. $\qquad\square$

