# OpenReview forum: "Efficient Graph Matching for Correlated Stochastic Block Models"
_NeurIPS.cc/2024/Conference — NeurIPS 2024 poster_

### Official Review · Reviewer_PrRf · 2024-07-09

**Soundness:** 3
**Presentation:** 3
**Contribution:** 4
**Rating:** 6
**Confidence:** 2

**Summary:**

In this paper, the authors tackle two related problems: graph matching and community recovery of correlated balanced SBM under the logarithmic average degree regime. The authors extend proofs from the ER model for similar tasks and overcome the difficulties in the proofs raised by working with the SBM model.
The main two theorems are: (1) for correlation levels above a certain threshold, almost exact and exact graph matching are possible, and (2), again, for correlation levels above a certain threshold, community recovery is possible.

**Strengths:**

I liked the paper's presentation in general. The problem setting is well described, and the motivation is clear.
The major strenght is the theoretical result (Theorem 1, and Theorem 2 as an application of Thm1). It is important and is a nontrivial generalization of the ER case for 2-class SBMs.

**Weaknesses:**

The weakness I see is not of the paper itself but more of the paper-conference combination. The strengths of the paper, as stated above, are the two theoretical results. But the 9-page paper ONLY includes the statements of these results and some ideas about the proofs. I found it very very nice, well motivated, etc. But I cannot review a ~70-page paper in all detail like it deserves. Not for a conference. I can see this as a beatuiful journal paper, where a reviewer can spend some months to study the paper like it deserves.

**Questions:**

There are some verbs missing in several sentences, and other grammatical or spelling errors (for instance in Erdos). I suggest a profound check on the writing.
I think the other comment (about the suitability of this paper for a conference) is not something the authors could change in the rebuttal phase.

**Limitations:**

The authors claim to address the limitations in Section 7. However, this section is more of a "future work" section (as it is named). It's true that every possible line of future work is a limitation of the present paper, but it's not presented in this way, and no other limitations are discussed.

---

> ### Author Rebuttal · Authors · 2024-08-06
>
> Thank you for your review and comments!
>
> We concur with the reviewer that lengthy papers present challenges with the conference format and its restrictions. However, we argue that this is a widespread general issue, not specific to our paper -- indeed, there are numerous NeurIPS papers with lengthy appendices. Within the constraints, we certainly aimed to provide a good summary of the results and the proof ideas and techniques within the main text.
>
> We will certainly do a careful check over the whole paper for grammatical and spelling errors. We will also ask a native English-speaking colleague of ours to proofread our paper, specifically focusing on spelling and correct phrasing of sentences. (We (the authors) are non-native English speakers, and we appreciate your patience as we improve the writing of the paper.)

---

> > ### Comment · Reviewer_PrRf · 2024-08-13
> >
> > Thank you for the answer.
> > I really hope to see this paper published somewhere :)

---

### Official Review · Reviewer_9uZn · 2024-07-10

**Soundness:** 3
**Presentation:** 3
**Contribution:** 3
**Rating:** 6
**Confidence:** 4

**Summary:**

The text presents progress in solving learning conditions within correlated stochastic block models with two balanced communities. The main result is the creation of the first efficient algorithm for graph matching. This algorithm works well when the average degree is logarithmic in the number of vertices and can accurately match nearly all vertices with high probability if the edge correlation parameter $ s $ satisfies $ s^2 \geq \alpha + \epsilon$, with $\alpha $ being Otter's tree-counting constant. The algorithm is extended to achieve exact graph matching, solving an open problem posed by Rácz and Sridhar; 2021. This new algorithm generalizes recent work by Mao, Wu, Xu, and Yu (STOC 2023), which relies on centered subgraph counts of structures called chandeliers. A key challenge addressed is managing estimation errors since the latent community partition cannot be perfectly recovered from a single graph in certain parameter regimes. The findings also lead to an efficient algorithm for exact community recovery using multiple correlated graphs, even in cases where it is impossible with just a single graph.

**Strengths:**

The paper has several interesting points. First, the paper deals with an important problem in graph mining, namely, graph matching in correlated SBM with two balanced communities.
%
The proposed solution is based on probabilistic bounds based on well-established works applied to SBM with a logarithmic average degree.
%
The paper also discusses an example of community recovery on CSBMs in polytime.

**Weaknesses:**

The paper is highly theoretical and, as such, most limitations are associated with this. For instance, it does not incorporate simulations as a sanity check of the proofs which could be useful to illustrate not only the usefulness of the theorems for graph matching but also the assumptions.
A minor weakness in this regard is that although the introduction indicates the assumptions for which the framework is developed, e.g., the level of sparsity and logarithmic average degree, it could be useful to discuss the implications on highly dense graphs when the parameters are the complement of the SBM parameters p,q.

**Questions:**

How differentiated the SBM model has to be from an ER model to effectively achieve exact and almost exact matching if $s^2>\alpha$?

**Limitations:**

The discussion section indicates some of the limitations of the work including the lack of runtime analysis, and that the condition $s^2>\alpha$ is believed to be necessary but that is not proven in this paper.

---

> ### Author Rebuttal · Authors · 2024-08-06
>
> Thank you for your review and comments!
>
> Comment #1 (runtime analysis):
>
> In the literature on random graph matching, the quest for polynomial-time algorithms has been a major driving force behind the recent surge of papers on the topic. These culminated in the recent breakthrough works [33] and [35] on correlated Erdős--Rényi random graphs. Our work continues along this line of work, going beyond the correlated Erdős--Rényi model, and giving the first polynomial-time algorithm for graph matching on correlated SBMs. Unfortunately, the running time is a high order polynomial --- this is an issue that is present already in the prior work [35]. Thus, the implementation of the algorithm for simulations would only be possible on small graphs. As such, our contribution should be viewed as a conceptual, theoretical result.
>
> At the same time, we do believe that fast and implementable algorithms do exist under some weakened condition, such as $s^2>\alpha'$ for some $\alpha' > \alpha$. One recent empirical evidence towards this is the work of Muratori and Semerjian [42], who empirically studied a significantly accelerated version of the subtree counting method, using a restricted family of subtrees with a limited depth and where each node has at most $2$ or $3$ descendants. It is conjectured in [42] that for some $\alpha'>\alpha$, the proposed faster algorithm can achieve exact graph matching on correlated Erdős--Rényi graphs much more efficiently than the original algorithm. In future work, we plan to empirically study their algorithm on correlated SBMs. Furthermore, we plan to attempt to rigorously show the exact graph matching result on correlated Erdős--Rényi graphs and later extend it to correlated SBMs. This direction is exciting to explore, but it is beyond the scope of the current paper.
>
> Comment #2 (sparsity and other density regimes):
>
> In Section 2 (from line $89$ to $100$ in our manuscript), we have argued that the logarithmic average degree is the most interesting parameter regime to study. The logarithmic average degree regime is also called the "bottleneck regime" of random graphs and it is the bare minimum for the graph to be connected. If we go sparser to constant average degree regime, then it is information-theoretically impossible to recover the communities exactly. If the vertices have polynomially growing degrees, then community recovery is easy as long as $\liminf_{n \to \infty} |p_{n}/q_{n} - 1| > 0$ (see [39]).
>
> In Section 7 (from line $363$ to $372$ in our manuscript), we have also discussed that our result can be extended to a denser regime (where the average degree grows as a (small) polynomial of $n$) too. In the even denser regime, whenever $\liminf_{n \to \infty} |p_{n}/q_{n} - 1| > 0$ is satisfied, we can do community recovery (exactly and efficiently) by applying the work of Mossel, Neeman, and Sly [39], and then using the exact graph matching algorithm from Mao, Wu, Xu, and Yu [35] as a black box on the correlated Erdős--Rényi models under each community.
>
> Comment #3 (on differentiating SBM from ER):
>
> To apply our result in the logarithmic average degree regime where $p=a\frac{\log n}{n}$ and $q=b\frac{\log n}{n}$ (Theorem 1),  we only need that $a\neq b$. If $a = b$, then the correlated SBM degenerates to the correlated ER model, where the result by Mao, Wu, Xu, and Yu [35] already applies. In other words, we do not need any nontrivial differentiation.
>
> Comment #4 (on the necessity of the condition $s^2 > \alpha$):
>
> Indeed, we conjecture that the condition $s^{2} > \alpha$ is necessary for a polynomial-time algorithm for graph matching to exist, just as [35] conjectures the same for the correlated Erdős--Rényi model. Thus, this graph matching in these models is conjectured to exhibit an information-computation gap; that is, a region of the parameters where the problem is information-theoretically feasible, but it is not possible with polynomial-time algorithms. There are many interesting problems that are conjectured to exhibit such an information-computation gap, but as of yet there is no problem where this has been proven. Instead, people often resort to proving results on restricted classes of algorithms (such as within the low-degree polynomials framework), which can be seen as evidence of computational hardness (though not as a full proof). Related to our work, in the easier task of correlation detection in the case of correlated Erdős--Rényi random graphs, the recent work [16] proved a corresponding low-degree hardness result.

---

### Official Review · Reviewer_xbYm · 2024-07-13

**Soundness:** 3
**Presentation:** 3
**Contribution:** 3
**Rating:** 6
**Confidence:** 2

**Summary:**

The paper extends graph matching from correlated Erdos Reyni graphs to correlated SBM graphs.
To do so, they apply the recent breakthrough work of Mao, Wu, Xu, and Yu (STOC 2023) for ER graphs, which are based on counting a special kind of graph called chandeliers. A prerequisite for this method is to be able to center the graph. The primary challenge is that SBM graphs a graph cannot be easily centered (as the intra and inter-community edges have different means). To this end, they first apply a weak-community recovery approach to get a rough estimate of the communities in the different graphs. Then, they use this information to get a match.

As an application, they show efficient community recovery given multiple correlated SBM graphs in a regime where recovery is information-theoretically impossible given only one graph (This also shows that the approach is interesting, as they use a rough community recovery on individual graphs to obtain a matching, which in turn leads to a complete community recovery using the graphs collectively). This is the first efficient algorithm for this problem.

**Strengths:**

This technically involved paper extends work on graph matching for correlated ER graphs to (in my understanding, a restricted version of)  correlated SBM graphs.

The key technical difficulty seems to be that the results for the ER graphs require centralization of the adjacency matrix, which is not directly possible in SBM graphs. To solve this, the paper first obtains a rough estimate of the communities in the graph. Then, it shows that they are able to obtain a matching using the resultant centralization (which may not be completely correct).

**Weaknesses:**

1. The definition of correlated SBM seems a special case of the usual definition of correlated graphs in the ER model. Specifically, the authors assume that the non-edges of the first graph also remain non-edges in the second graph.

2. The paper does not contain any simulation results. Implementing this algorithm and applying it to SBM graphs should not be hard.

**Questions:**

It would be great if the authors could comment on the aforementioned weaknesses.

**Limitations:**

The authors have adequately addressed the limitations.

---

> ### Author Rebuttal · Authors · 2024-08-06
>
> Thank you for your review and comments!
>
> Comment #1:
>
> We would like to kindly make some clarifications regarding these comments.
>
> Firstly, the correlated ER model is actually a special case of the correlated SBM model (not the other way around). Specifically, when the in-community edge density $p$ is equal to the cross-community edge density $q$, then the resulting SBM is in fact an ER graph.
>
> Secondly, we do not assume that the non-edges of the first graph remain non-edges in the second graph. Since both graphs are subsampled from a mother graph, it is possible that some non-edges in the first graph are edges in the second graph. That said, it is true that the non-edges in the mother graph remain non-edges in both sub-sampled correlated graphs.
>
> Comment #2:
>
> In the literature on random graph matching, the quest for polynomial-time algorithms has been a major driving force behind the recent surge of papers on the topic. These culminated in the recent breakthrough works [33] and [35] on correlated Erdős--Rényi random graphs. Our work continues along this line of work, going beyond the correlated Erdős--Rényi model, and giving the first polynomial-time algorithm for graph matching on correlated SBMs. Unfortunately, the running time is a high order polynomial --- this is an issue that is present already in the prior work [35]. Thus, the implementation of the algorithm for simulations would only be possible on small graphs. As such, our contribution should be viewed as a conceptual, theoretical result.
>
> At the same time, we do believe that fast and implementable algorithms do exist under some weakened condition, such as $s^2>\alpha'$ for some $\alpha' > \alpha$. One recent empirical evidence towards this is the work of Muratori and Semerjian [42], who empirically studied a significantly accelerated version of the subtree counting method, using a restricted family of subtrees with a limited depth and where each node has at most $2$ or $3$ descendants. It is conjectured in [42] that for some $\alpha'>\alpha$, the proposed faster algorithm can achieve exact graph matching on correlated Erdős--Rényi graphs much more efficiently than the original algorithm. In future work, we plan to empirically study their algorithm on correlated SBMs. Furthermore, we plan to attempt to rigorously show the exact graph matching result on correlated Erdős--Rényi graphs and later extend it to correlated SBMs. This direction is exciting to explore, but it is beyond the scope of the current paper.

---

### Official Review · Reviewer_K457 · 2024-07-13

**Soundness:** 3
**Presentation:** 3
**Contribution:** 2
**Rating:** 5
**Confidence:** 3

**Summary:**

The author(s) consider graph matching and community recovery on correlated stochastic block models. In the stochastic block model (SBM) the algorithm's input is a graph G generated by first partitioning vertices into two equal size clusters and then adding every intra-cluster edge with probability p and every inter-cluster edge with probability q, independently. In the correlated version onG_ f irst samples a graph G from the SBM, and then takes two independent edge samplings of G, referred to as G_1 and G_2 -- these are the input of the algorithm (with randomly permuted vertices).  Thus, the correlated SBM setting generalizes the more basic correlated Erdos-Renyi setting, in which there is only one community (or, one can think of p=q). In the latter setting the objective is graph matching. In the more general setting of correlated SBMs the objective function is community recovery, but graph matching is a natural sub-objective along the way, as it is quite reasonable to match G_1 to G_2 first, and then use the improved edge density for community detection.

A strong recent result of Mao et al (STOC'23) established a tight threshold for reconstruction in the correlated Erdos-Renyi setting, and provided a polynomial time algorithm in the regime where  matching is possible. The present paper basically ports the techiques of Mao et al from the correlated Erdos-Renyi to the correlated SBM setting. Specifically, they develop technical tools for using Chandelier counts, which underly the result of Mao et al, in the SBM setting.

**Strengths:**

The correlated SBM model is a very natural setting to study, and some of the author(s)' results are tight.

**Weaknesses:**

There is a number of lower level technical challenges that stem from the need of extending an Erdos-Renyi result to the SBM setting, but the most conceptual one appears to be the fact that when using partial matching results to perform community recovery on the union of the two graphs, one needs to handle errors that come from the non-exact matching. This, while definitely an interesting technical challenge, does not seem sufficiently compelling as a conceptual one.

**Questions:**

None.

**Limitations:**

None.

---

> ### Author Rebuttal · Authors · 2024-08-06
>
> Thank you for your review and comments!
>
> We agree with the reviewer that there are several technical challenges to overcome, and a significant part of our contributions is indeed showing how to overcome these technical challenges.
>
> However, we also believe that there are compelling conceptual challenges in our work, the main one being how to center the edge-indicator random variables for the centered subgraph counts. Let us elaborate on why this is so.
>
> First of all, centering the edge-indicator random variables is crucial for the statistic to work, as discussed in Mao, Wu, Xu, Yu (2023) [ref. 35]. In the Erdős--R\'enyi setting it is trivial how to center the edge-indicator random variables, since it is a homogeneous model, where all edge probabilities are the same. However, for heterogeneous models, it is no longer clear how to do this. Moreover, for general models, this may not even be possible. Consider, for instance, the graph matching problem for correlated inhomogeneous random graphs, studied in the recent works [51] and [17]. In particular, the work [17], titled "Efficiently matching random inhomogeneous graphs via degree profiles" studied efficient algorithms for this task. However, they note that
>
> "in the inhomogeneous case it is also very difficult to obtain good estimates on these edge probabilities (for the obvious reason that the number of parameters is proportional to the number of observed variables)."
>
> In other words, this challenge is made explicit in the work [17]. The authors get around the challenge by considering a much simpler statistic: degree profiles. However, this comes at a significant cost: their matching algorithm only works when $s = 1 - O(1/\log^{2}(n))$. In other words, it only works for vanishing correlation; it cannot handle constant correlation. A major point of our work is that the matching algorithm works for constant correlation.
>
> More generally, it is not at all clear for which classes of graphs a "good enough" centering of the edge-indicator random variables can be computed. We view this as both a conceptual and a technical challenge. A main point of our work is to show that SBMs indeed fall within this class.

---

> > ### Comment · Reviewer_xbYm · 2024-08-11
> >
> > I thank the authors for their response, and shall keep my score unchanged. Best wishes.

---

> > ### Comment · Reviewer_K457 · 2024-08-12
> >
> > Thank you for the response.

---

### Official Review · Reviewer_Nbs5 · 2024-07-27

**Soundness:** 4
**Presentation:** 4
**Contribution:** 4
**Rating:** 7
**Confidence:** 4

**Summary:**

The paper under revision addresses the problem of graph matching and community recovery in correlated stochastic block models (SBM) with two balanced communities. In particular, it studies the regime where the vertices of the parent graph have a logarithmic average degree, considering the within-community edge probability to be $a\frac{\log n}{n}$ and between-community edge probability to be $b\frac{\log n}{n}$. The edge correlation parameter is equal to $s$.

Regarding graph matching, they proved in Theorem 1 the existence of a polynomial-time algorithm that can achieve both almost exact and exact matching. Almost exact matching is achieved when the square of the edge correlation is greater than Otter’s tree counting constant ($s^2 > \alpha$). To obtain exact matching, an additional condition is necessary, namely $s^2(a+b)/2 > 1$. This result extends known results on almost exact and exact matching for correlated Erd\H{o}s--Rényi graphs obtained in Mao et al. (2023). It corroborates the previous work by Racz & Sridhar (2021), which determined that exact matching in correlated SBM is only achieved when $s^2(a+b)/2 > 1$.

In the context of community detection, Theorem 2 of the manuscript states that exact community recovery in correlated SBM is achieved by a polynomial-time algorithm when \( s^2 > \alpha \), and it satisfies the information-theoretic threshold for community recovery in correlated SBM established by Racz & Sridhar (2021). The authors provide a brief description of the polynomial-time algorithm in Section 5 and a more detailed explanation in Appendix C. In this way, one of the main contributions of this work is to propose an efficient algorithm for graph matching for correlated SBM.

In terms of the algorithm itself, it is structured by first applying a community detection algorithm proposed by Mossel et al. (2015) to each of the graphs to ensure exact recovery of the communities. Then, an adaptation of the algorithm proposed by Mao et al. (2023) for correlated Erdős–Rényi graphs, based on rooted trees known as chandeliers, is applied.

The manuscript combines established ideas from previous works, specifically integrating the concepts from Racz & Sridhar (2021) and Mao et al. (2023) to develop a polynomial-time algorithm for graph matching in correlated SBM. The manuscript is well-structured and provides detailed explanations of the algorithms, results and their connections to related works.

References:

Mao, Cheng, Wu, Yihong, Xu, Jiaming, & Yu, Sophie H. 2023. Random graph matching
at Otter’s threshold via counting chandeliers. Pages 1345–1356 of: Proceedings of the
55th Annual ACM Symposium on Theory of Computing.

Mossel, Elchanan, Neeman, Joe, & Sly, Allan. 2015. Consistency thresholds for the planted
bisection model. Pages 69–75 of: Proceedings of the forty-seventh annual ACM
symposium on Theory of computing.

Racz, Miklos, & Sridhar, Anirudh. 2021. Correlated stochastic block models: Exact graph
matching with applications to recovering communities. Advances in Neural Information
Processing Systems, 34, 22259–22273.

**Strengths:**

The manuscript combines established ideas from previous works, specifically integrating the concepts from Racz & Sridhar (2021) and Mao et al. (2023) to develop a polynomial-time algorithm for graph matching in correlated SBM. The manuscript is well-structured and provides detailed explanations of the algorithms, results, and their connections to related works.

**Weaknesses:**

X

**Questions:**

X

**Limitations:**

X

---

> ### Author Rebuttal · Authors · 2024-08-05
>
> Thank you for your review!

---

### Decision · Program_Chairs · 2024-09-25

**Decision:**

Accept (poster)

**Comment:**

The problem of graph alignment -- ie. of matching the vertices of two random graphs with correlated edges -- has received significant attention in the literature. The present paper considers the case of correlated stochastic block models (cSBM) with two communities. The main result is to show how a certain estimator from previous work, which is based on on counting "signed chandeliers" in the graph, achieves good performance, and (most importantly) does so in polynomial time. The paper adds to recent information theoretic results for this model, and also to algorithmic results for the simpler Erd\H{o}s-Rényi case. At least in some cases, it reaches the natural information theoretic threshold below which alignment becomes (in some sense) impossible.

Reviewers' scores were 5, 6, 6 and 7. They all seem to agree that this is a technically challenging paper, which solves an interesting problem. Differences in scores seem to reflect differing views of the relevance of the paper. My own opinion is that this paper contains important ideas required to deal with the challenge brought about by the fact that edge probabilities take two values. Moreover, the problem considered is of broad interest, albeit the contributions here are mostly theoretical. Overall, I believe the paper should appear in NeurIPS.